# Transcription elongation can be sufficient, but is not necessary, to advance replication timing

Athanasios E Vouzas[1,2], Takayo Sasaki [2], Juan Carlos Rivera-Mulia [3], Jesse L Turner [1,2], Amber N Brown [1], Karen E Alexander[4], Laura Brueckner[5], Bas van Steensel [5] & David M Gilbert [2✉]

## Abstract

**DNA replication timing (RT) often correlates with transcription during cell fate transitions, yet notable exceptions indicate a complex relationship. Using a reductionist system in mouse embryonic stem cells, we manipulate transcriptional length and strength at a single locus upstream of the silent, late-replicating *Pleiotrophin* (*Ptn*) gene. Small reporter genes driven by two of four promoters advance RT, whereas all promoters advance RT when driving the 96-kb endogenous *Ptn* gene. Inducible transcription of *Ptn*, but not the reporter, triggers a rapid and reversible RT advance, providing a system to manipulate RT independent of differentiation. Strikingly, deletion of the *Ptn* promoter and enhancers abolishes transcription yet does not prevent the developmental RT switch to early replication during neural differentiation. These findings, supported by parallel genome-wide analyses during differentiation, demonstrate that transcriptional elongation can causally advance RT in a rate-dependent and context-specific manner, but that transcription is neither necessary nor sufficient for RT advancement. Our results provide a solid empirical base with which to re-evaluate decades of seemingly contradictory literature.**

**Keywords** Transcription Regulation; DNA Replication Timing; Epigenomic Remodeling During Differentiation; Cell Fate Transitions; Mouse Embryonic Stem Cells
**Subject Categories** Chromatin, Transcription & Genomics; DNA Replication, Recombination & Repair

## Introduction

Mammalian chromosomes replicate segmentally in a temporal order, called the replication timing (RT) program. RT is cell-type specific and is disrupted in diseased states (Rivera-Mulia et al, 2017, 2019b), and the RT program is critical for the faithful propagation of a cell's epigenomic state (Klein et al, 2021; Lande-Diner et al, 2009). In yeasts, RT is regulated by mechanisms that promote or antagonize steps in the conversion of DNA-bound MCM double hexamers into an active helicase (Hu and Stillman, 2023; Vouzas and Gilbert, 2023). Many of these mechanisms are conserved in mammalian cells, but the complexities of changing epigenomic states during cell fate transitions necessitate additional layers of upstream regulation. These additional layers regulate changes in the RT of sub-megabase replication domains (RDs) during cell fate transitions by altering the probability of initiating replication within multi-kilobase initiation zones (Sima et al, 2019; Rivera-Mulia et al, 2019a; Dileep et al, 2019; Vouzas and Gilbert, 2021; Wang et al, 2021a; Carrington et al, 2025). In mouse embryonic stem cells (mESCs), developmentally regulated early replicating RDs contain discrete early replication control elements (ERCEs) necessary to maintain domain-specific early RT, 3D architecture, chromatin compartment interactions, and transcription (Sima et al, 2019). These elements interact in 3D, are surrounded by acetylated histones and are bound by multiple transcription factors, which may function by creating a local sub-nuclear environment favorable for initiating replication (Sima et al, 2019; Turner et al, 2025). ERCEs are reminiscent of the Fkh1,2 transcription factors in budding yeast that interact and promote early replication independent of their transcription activity (Ostrow et al, 2017; Hoggard et al, 2021). RT can also be influenced by factors such as Rif1 that antagonizes initiation via mechanisms conserved with yeasts (Hiraga et al, 2017; Gnan et al, 2021; Peace et al, 2014; Heinz et al, 2018; Klein et al, 2021; Stow et al, 2022; Brueckner et al, 2020).

Transcriptional activation and silencing of genes is tightly coordinated with RT changes during cell fate transitions (Dileep et al, 2019; Hiratani et al, 2010; Rivera-Mulia et al, 2015) and early RT correlates with transcriptional activity in every multi-cellular organism investigated (Gilbert, 1986; Goldman et al, 1984; Hatton et al, 1988; Vouzas and Gilbert, 2021). However, the relationship between RT and transcription has been enigmatic. The correlation is largely driven by a majority of genes that are early replicating and expressed in all cell types (constitutively early housekeeping genes), while genes that are developmentally regulated show a much weaker correlation between RT and transcriptional activity (Rivera-Mulia et al, 2015). Indeed, some genes are induced when switching from early to late replicating (Rivera-Mulia et al, 2015; Hiratani

[1]Department of Biological Science, Florida State University, Tallahassee, FL 32306, USA. [2]San Diego Biomedical Research Institute, San Diego, CA 92121, USA. [3]Department of Biochemistry, Molecular Biology and Biophysics, University of Minnesota Medical School, Minneapolis, MN 55455, USA. [4]College of Medicine, Florida State University, Tallahassee, FL 32306, USA. [5]Division of Gene Regulation and Oncode Institute, Netherlands Cancer Institute, Amsterdam, Netherlands. ✉E-mail: gilbert@sdbri.org

et al, 2010). Moreover, mono-allelically expressed and asynchronously replicating genes show no relationship between their transcription and RT (Heskett et al, 2022).

Several groups have studied the effect of ectopic transcription on RT, with seemingly contradictory findings. In two studies targeting a strong acidic transcriptional activator to the promoter of late replicating genes in mESCs, both activated their transcription and advanced their RT (Brueckner et al, 2020; Therizols et al, 2014). In another case, targeting histone acetyltransferases to the mid S-phase replicating, human β-globin domain, inserted in the genome of transgenic mice, led to an advance in RT in the absence of transcriptional induction in both spleen lymphocytes and mouse embryonic fibroblasts isolated from these transgenic mice (Goren et al, 2008). Ectopic insertions of promoters have also had varied effects on RT. Insertion of an SV40-promoter driven vector in Chinese hamster cells was shown to advance RT of the insertion locus at one insertion site but not at two others (Gilbert and Cohen, 1990). Insertion of the β-actin and β-globin promoters in a mid-late replicating domain of chicken DT-40 cells advanced RT in a transcription-dependent manner (Brossas et al, 2020). Also, in DT-40 cells, transcription of a short transcript or low levels of transcription of a longer transcript were unable to advance RT, while high levels of transcription of a long transcript did advance RT (Blin et al, 2019; Hassan-Zadeh et al, 2012). The primary challenge with interpreting the seemingly contradictory results from the aforementioned collection of studies is the use of multiple methods to induce transcription and perturb RT in a variety of species, cell types and chromosomal sites.

Here, we take a systematic, reductionist, approach to identify elements of transcriptional activity that can advance RT of a developmentally regulated RD in which transcription of a single gene (Ptn) is known to be induced coincident with a switch to early RT. We find that, with short selectable marker gene insertions, promoters that induce a high transcription rate can partially advance RT. However, lower levels of transcription, that are nonetheless fully capable of conferring drug resistance, have no detectable effects on RT at the Ptn locus or in multiple vector and chromosomal contexts. Transcription driven by the same promoters integrated at the same site but driving the 96 kb Ptn gene can all advance RT to varying extents. Taking advantage of the artificial Tetracycline-inducible promoter (TRE), we demonstrate that the advance in RT by this promoter, when driving Ptn, is proportional to the rate of transcription, affected within hours of the onset of transcription and is reversible within a single cell cycle, providing direct evidence that the act of transcription elongation can be sufficient to advance RT. However, transcription of Ptn is not necessary to advance RT in the context of differentiation; deletion of the enhancers and promoter of Ptn eliminated all transcription induction within the Ptn replication domains, without preventing the RT advance during a cell fate transition. Finally, we performed a genome-wide survey of RT vs. total, domain-wide, changes in transcriptional elongation during differentiation. Although domain-wide upregulation of transcription was more often accompanied by advances rather than delays in RT, half of cases resulted in no change in RT. We also identified examples of RT changes in the absence of transcription, as well as late-replicating and RT-delayed domains with abundant expression and transcriptional induction of both large and small genes that was accompanied by no change in RT. Our results reconcile many

seemingly contradictory reports in the literature, demonstrating that: (1) transcriptional elongation can have a direct, rate-dependent effect on RT but not in all chromatin contexts; and (2) although RT advances are usually accompanied by transcription induction, transcription is not necessary for an RT advance and early replication is not necessary for transcription. Thus, causal relationships between transcription and RT are context-specific, and other factors such as ERCEs can regulate RT independent of transcription. Future studies should focus on the contexts and mechanisms by which transcriptional elongation can advance RT, whether or not and in what contexts early RT can promote transcription, and the nature of transcription-independent mechanisms that can robustly advance RT during cell fate changes.

## Results

### Transcription per se is not sufficient to advance RT

We reasoned that one explanation for the varied effects of transcription on RT in the literature could be the use of different promoters. To systematically study the effect of transcriptional activation driven by different promoters on RT, we inserted a set of reporter genes into the same location upstream of the mouse Pleiotrophin (Ptn) gene, which encodes a neuronal growth factor. We previously identified the replication domain containing Ptn as one that switches from late to early coincident with transcriptional induction during the differentiation of mESCs to mouse neural precursor cells (mNPCs) (Hiratani et al, 2004, 2008, 2010). Subsequently, we and others have shown that the domain can advance RT in response to transcriptional induction induced by targeting a strong transactivator to the Ptn promoter in the absence of differentiation (Therizols et al, 2014; Brueckner et al, 2020). Ptn is the only gene in the domain that undergoes transcriptional induction at the time of the RT switch during differentiation to mNPCs; three other nearby genes, the Dgki gene whose 3' end encroaches into the Ptn domain, an upstream non-coding RNA and the Chrm2 gene downstream of Ptn, all remain silent in both ESCs and mNPCs (Giorgetti et al, 2016; Rivera-Mulia et al, 2018). We constructed four reporter genes expressing the Hygromycin-Thymidine Kinase (HTK) gene fusion from a 2.0 kb transcript, differing only in the promoter used to drive expression (Fig. 1A): the mouse and human phosphoglycerate kinase 1 (mPGK and hPGK) promoters, the cytomegalovirus immediate-early enhancer/chicken β-actin (CAG) promoter and the inducible tetracycline response element (TRE) promoter. The CAG promoter is a widely used synthetic construct that contains a cytomegalovirus (CMV) enhancer element, promoter, first exon and first intron of the chicken β-actin gene and the splice acceptor of the rabbit β-globin gene. The TRE promoter contains seven repeats of the bacterial tetO operator separated by a spacer. The mESC cell line used in all these studies expresses a reverse tetracycline transactivator (M2rtTA), consisting of the *E. coli* tetracycline repressor DNA binding domain fused to the Herpes virus VP16 acidic transactivation domain. The coding sequence for this artificial transactivator is integrated in front of and expressed constitutively from the Rosa26 promoter (Quinodoz et al, 2021). Upon introduction of Doxycycline (Dox), a derivative of Tetracycline, rtTA binds to the tetO

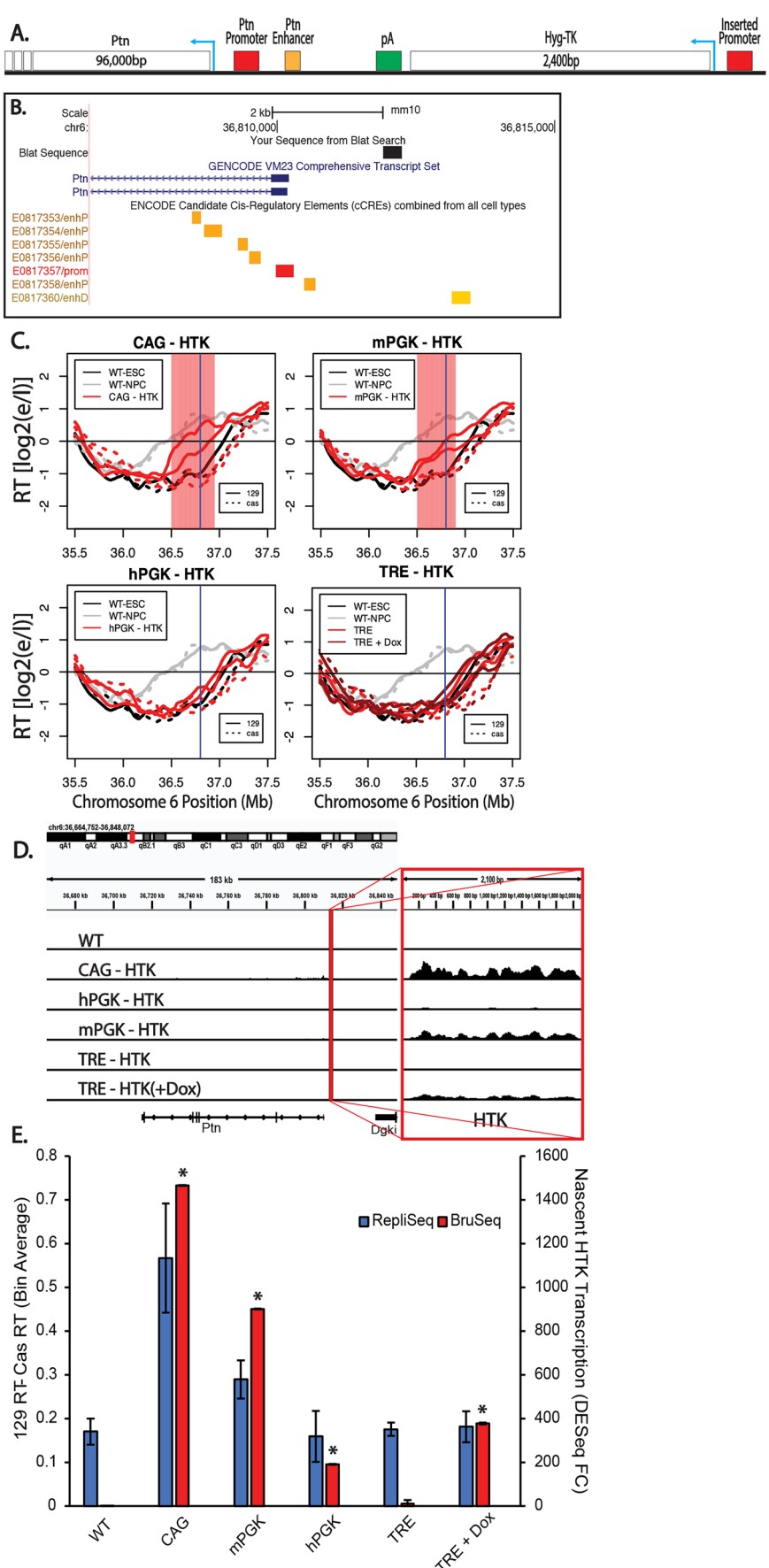

**Figure 1. Promoter-specific effects of reporter gene insertion on RT.**

(A) Schematic of the region directly upstream of the Ptn domain. All insertions were made ~1.7 kB upstream of the Ptn domain, leaving the natural Ptn promoter and enhancers intact. Arrows indicate the direction of the HTK and Ptn genes. The donor vectors used to make the insertions into the Ptn domain contained one of the four promoters used in this study driving the expression of a fusion positive/negative selectable marker consisting of the hygromycin resistance gene fused to the tyrosine kinase gene (HTK), followed by a transcription termination sequence (pA). Therefore, the inserted sequences were identical except for the promoter, allowing us to directly compare the effects of the four promoters on RT. (B) UCSC Genome Browser track indicating the locations of the exact genomic sequence (YourSeq) that was replaced by the insertions, as well as the Ptn promoter (red), proximal enhancers (orange), and distal enhancer (yellow) elements. All candidate *cis*-regulatory elements (ENCODE cCREs) remained intact after transgene replacement. (C) RT of the Ptn domain for the CAG-HTK (top left), mPGK-HTK (top right), TRE-HTK (bottom left) and hPGK-HTK (bottom right) cell lines. In each plot, the black and gray lines indicate the RT of WT ESCs and mNPCs, respectively. The red lines indicate the RT of the insertion cell line. In the TRE-Ptn plot, dark red lines and light red lines indicate the RT of TRE-Ptn cells with and without Dox, respectively. Solid and dashed lines indicate the RT of the *musculus* (mouse strain 129) and *castaneus* alleles, respectively. Vertical blue line indicates the site of the Promoter-HTK insertion. The red shading indicates 50 kb windows with statistically significant differences in RT between WT casteneus and modified 129 alleles, determined as described in Methods. (D) Genome browser view of the Ptn domain displaying nascent RNA (Bru-Seq) signal for each one of the four promoters. The left track displays the unparsed Bru-Seq signal in the Ptn domain. The vertical red line indicates the locus where the HTK insertion was made. The zoomed-in Bru-Seq signal for HTK is displayed on the right track. The dynamic range for both tracks is the same. Biological replicates are shown in Fig. EV1. (E) Bar graph of the relationship between nascent HTK transcription and the observed advance in the RT of the Ptn domain. HTK expression was calculated as the fold change vs WT, based on 4 biological replicates for mPGK, hPGK and CAG, and 2 biological replicates for WT, TRE, and TRE-Dox. The error bars for the HTK expression bar graphs represent standard error. The asterisks indicate a significant difference ($p < 0.001$, calculated using the Wald test) in the levels of HTK expression relative to HTK expression in the parental cell line as described in Methods. There are no asterisks for the RT data, as statistical significance was calculated for individual 50 kb windows as shown in (C). The RT advance of the Ptn domain is measured as the average difference between the RT of 50 kb bins of the *musculus* allele and the *castaneus* allele. As indicated, the *musculus* (129) allele replicates slightly earlier than the *castaneus* allele in WT; RT differences between the two alleles are normalized for these allelic differences. The error bars in the RT data show the range of the difference among the 50 kb bins between the two biological replicates. Source data are available online for this figure.

elements of the TRE promoter and induces transcription. For all experiments, unless otherwise stated, cells with the inserted TRE promoter were treated with 2 µg/ml of Dox for 24 h. These reporter cassettes were inserted, one at a time, via CRISPR mediated homologous recombination, into a site ~2.0 kb upstream of the Ptn gene transcription start site, avoiding all candidate *cis*-regulatory elements and replacing the genomic sequence marked as "YourSeq" in Fig. 1B (ENCODE Project Consortium et al, 2020). Insertion was made into the *musculus* allele of a *musculus* x *Castaneus* hybrid mESC line, greatly facilitating phasing of the two Ptn alleles due to the single-nucleotide polymorphism (SNP) density of approximately one SNP every 150 bp. In this way, we could distinguish the effects of insertion into one allele, while retaining the unmanipulated homologous allele in the same cells as an internal control, permitting the sensitivity to detect small RT changes (Rivera-Mulia et al, 2018; Sima et al, 2019).

To assess the effects of these insertions on RT and transcription rates, we performed Repli-Seq and Bru-Seq (Marchal et al, 2018; Paulsen et al, 2014). In Repli-seq, nascent DNA is labeled with BrdU, followed by flow cytometry to purify cells in early vs. late S phase based on their DNA content, immunoprecipitation of BrdU-substituted DNA and sequencing of the nascent DNA. Results are expressed as a log2 ratio of early to late S nascent DNA (log2E/L). BrU-seq labels total nascent RNA, which is then immunoprecipitated, sequenced and expressed as reads per million per kilobase (RPMK). The reporter cassettes harboring the CAG and mPGK promoters led to advances in the RT of the Ptn domain (Fig. 1C). In contrast, the reporter cassettes with hPGK or the inducible TRE promoter, with or without 2 µg/ml Doxycycline (Dox) for 24 h, were unable to advance the RT of the Ptn domain (Fig. 1C). Since all reporter cassettes transcribe HTK and sustain Hygromycin resistance, transcription of a reporter gene per sé is clearly not sufficient to advance RT. However, the CAG and mPGK promoters that were able to advance RT induced higher expression levels of HTK than the hPGK and TRE promoters (Fig. 1D). Note that Bru-Seq detects a small amount of transcriptional readthrough of the small transgene with the CAG and mPGK promoters, suggesting

that the poly-A termination site, used in many vector designs, is not fully effective at constraining high levels of transcription to the vector sequences (Figs. 1D and EV1A,B), albeit the level of read through Ptn expression is substantially lower than the levels of HTK expression (Fig. EV1C). Directly comparing the RT advance of the Ptn domain and the levels of transcription of HTK (Fig. 1E) revealed that, for the two promoters that did advance RT, the RT advance was greater when transcription rate was higher. There are several possible interpretations of these data. There may be small RT advances with the weaker promoters that are below the level of detection. There may be a transcription level threshold necessary to advance RT. Read-through transcription seen with CAG and mPGK, creating a longer transcript, may be necessary to elicit the RT advance (Fig. EV1D). There could also be transcription-independent differences between these promoters that influence their ability to advance RT. Regardless, our results reconcile previous contradictory reports in the literature (Blin et al, 2019; Brossas et al, 2020; Gilbert and Cohen, 1990; Goren et al, 2008; Therizols et al, 2014) by demonstrating that some selectable marker gene insertions advance RT while others do not.

## The inability of hPGK to advance RT is vector and position independent

The hPGK promoter is commonly used to drive transcription and clearly produces sufficient gene expression for drug resistance, yet neither of these promoters can advance RT at the Ptn locus. If its lack of effect on RT is position-independent, such a promoter can be a useful tool to manipulate genomes without the concern of altering RT and consequently inducing significant changes in surrounding chromatin context. To determine whether hPGK can drive gene expression without affecting RT in a vector and position-independent manner, we took advantage of two previously characterized F121-9 clonal cell lines (Brueckner et al, 2020) with a total of 107 single-copy/single-allele insertions of a PiggyBac (PB) vector expressing GFP from the hPGK promoter (Fig. 2A). By analyzing RT in these lines, we could evaluate the effect of a

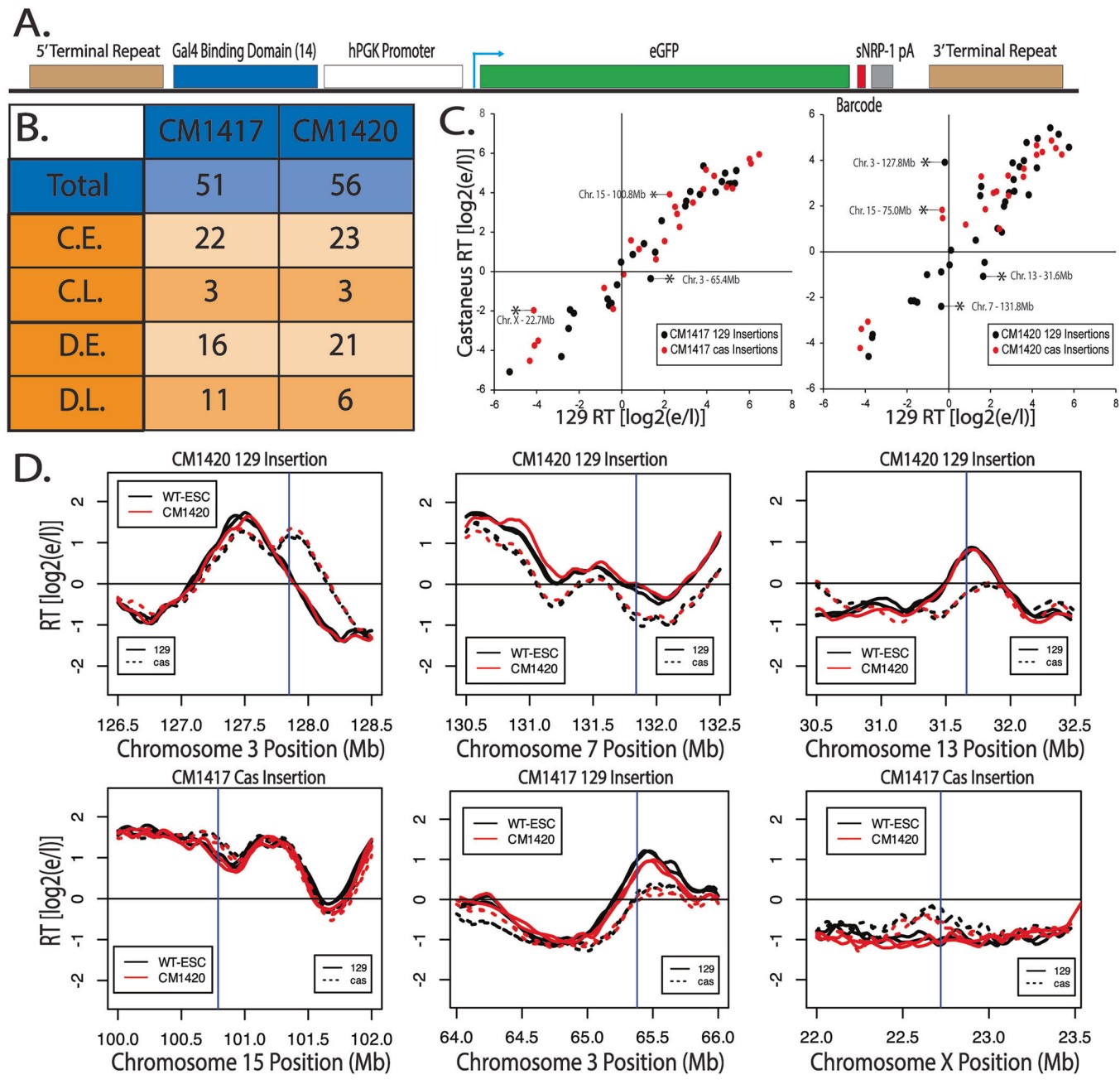

**Figure 2. The absence of RT advance with hPGK vectors is not vector or position dependent.**

(A) Schematic of the donor construct used for the Piggy Bac (PB) transposable element insertions, described in Brueckner et al (2020). The human PGK promoter is driving the expression of the enhanced Green Fluorescent Protein (eGFP) gene, along with a barcode specific to each insert. Transfection of this vector into F121-9 hybrid mESCs, along with the PB transposase, results in its insertion into multiple genomic sites. (B) Total number and RT distribution of PB insertion sites in two clones CM1417 and CM1420. Piggy Bac insertions are slightly enriched in constitutively (C.E.) and developmentally (D.E.) early replication domains, vs. constitutively (C.L.) and developmentally (D.L.) late domains. (C) Scatter plots of the RT of all ectopic insertion sites of the *musculus* (129) allele vs. *castaneus* allele for the CM1417 (left) and the CM1420 (right) clones. Black and red dots represent loci where the insertion was made in the 129 and cas allele, respectively. Asterisks indicate insertion sites where there is a significant difference between the replication timing of the 129 and the cas alleles, determined as described in Methods. Comparison of insertions to WT RT on the same alleles is shown in Fig. EV2. (D) RT profiles of six insertion sites, marked with asterisks in Fig. 2C, in which the two alleles replicate asynchronously, demonstrates that their differences in RT are species specific, not due to the PB insertion. Vertical blue lines indicate the site of the PiggyBac insertion. Source data are available online for this figure.

different hPGK-driven selectable marker construct on RT when integrated at many different chromosomal sites. Although there was a slight bias towards PB integration into early replicating domains, we detected integrations at sites replicating throughout S phase, including both constitutive and developmental late replicating regions (Fig. 2B). Results revealed that all single-copy/single-allele insertion sites showed a high correlation between the *musculus* and *castaneus* alleles both in the mutant CM lines (Fig. 2C) and the parental WT cell lines (Fig. EV2A), demonstrating preservation of RT, despite ectopic transcription in one allele. The few domains whose RT noticeably differed between alleles were at sites where there was a natural difference in the RT of the *musculus* vs *castaneous* genomes (Fig. 2D). Statistical analysis revealed a significant RT advance at only one of the insertion sites (red line in Fig. EV2B). The RT plot of the insertion site with the statistically significant RT advance is shown in Fig. EV2C (top right). A small number of other insertion sites had small but statistically non-significant RT changes (Fig. EV2C).

To assess the transcription levels at each insertion site we mined the mRNA-seq data from Brueckner et al. Since the inserted eGFP sequence is identical at all insertion sites this was challenging. However, dividing the total eGFP expression by the number of insertions in each clone reveals that the average eGFP expression is higher than 99% of all genes in both clones (Fig. EV2D). To assess expression levels across each insertion, we filtered for reads containing the entire 16 nucleotide barcode. It is important to note that coverage across any given 16 nucleotide segment of the genome may be low and therefore, this type of analysis can only be informative to assess the expression of barcodes relative to each other and not to other genes. Further, low or no expression of a barcode may be the result of low coverage, rather than a biological observation. Barcode-specific expression analysis was limited to 61/107 insertions for which the barcode has been identified. Calculating the RPM of each individual barcode reveals that barcodes inserted in early replicating loci tend to be more highly expressed than barcodes inserted in late replicating loci (Fig. EV2E). However, there are still a number of insertions in late replicating loci that are highly expressed. Interestingly, there was no correlation between the levels of transcription originating from a given insertion and changes in the RT of the insertion site (Fig. EV2E). We conclude that, with rare exceptions, transcription from the hPGK promoter fails to advance RT at many ectopic sites in mESCs.

## Transcription of the Ptn gene enhances the ability of promoters to advance RT

A previous study in chicken DT40 cells provided evidence suggesting that longer transcripts are more effective at advancing RT but there was not a direct comparison of transcription and RT at the same genomic location, in the same cell lines (Blin et al, 2019). To directly investigate the influence of gene size on RT, we took advantage of the negative selection marker of our reporter genes (Fig. 1) to delete the sequences between each of our inserted promoters and the +1 TSS for the 96 kb Ptn gene, thereby bringing the Ptn gene under the transcriptional control of each promoter (Fig. 3A). Under these conditions, the TRE promoter (in the presence of Dox) and the hPGK promoter gained the ability to advance RT (Fig. 3B), while the CAG promoter led to a more

substantial advance in RT. Surprisingly, the mPGK promoter was very weak in this context (Fig. 3B and Fig. EV3), and advanced RT to a lesser extent. Quantification of the levels of Ptn transcription, and the RT advance of the Ptn domain induced by each promoter, revealed a positive correlation between RT advance and transcription rates (Fig. 3C,D, Fig. EV3). Altogether, results using the same set of promoters in the same location transcribing units of different length support a role for elongation of transcription in advancing RT.

## Dox-Induced RT advance is rapid, reversible and correlates to transcription levels

The ability of the TRE to advance RT in a Dox-dependent manner when driving the expression of the 96 kb Ptn gene provided the opportunity to directly investigate the effect of transcription rate on RT, modulating only a single variable at a time: either the dosage or the time of rtTA DNA binding activity induced by Dox. We treated TRE-Ptn cells with various concentrations of Dox for 24 h (approximately 1.5 cell cycles), followed by Repli-Seq and Bru-Seq. Under these conditions, increasing concentrations of Dox lead to increased rates of Ptn transcription correlating with advances in RT, until the point at which both plateau (Figs. 4A–C and EV3A). Notably, the effects of increasing concentrations of Dox on transcription were restricted to the Ptn gene (Fig. EV4B,C).

To measure kinetics of the transcription-induced RT switch, we treated TRE-Ptn cells with 2 µg/ml of Dox for 0, 3, 6, 12, and 24 h post treatment, followed by a 2-h BrdU (Repli-Seq) or 30-minute BrU (Bru-Seq) label in the continued presence of Dox. Interestingly, 3 h of Dox treatment was sufficient to induce maximal Ptn transcription rates, and led to a substantial advance in RT, while 6 h of treatment was sufficient to induce an RT advance similar to 12 and 24 h of treatment (Fig. 5A,B). Taking into account the 2-h BrdU label following the 6-h induction, that S phase is approximately 8 h, and that the average doubling time is approximately 14 h, we conclude that the RT shift occurs within the first cell cycle of transcriptional induction. We further investigated whether the observed RT advance is sustained upon Dox removal, perhaps as an autonomous epigenetic memory. We treated TRE-Ptn cells with 2 µg/ml of Dox for 24 h and then removed the Dox for a further 24 h. Within 24 h of the removal of Dox, transcription of the Ptn gene was eliminated and the RT of the Ptn locus returned to its original timing (Figs. 5C,D and EV5). Quantification of the levels of Ptn expression and changes in RT show a positive correlation of the increase in Ptn transcription with RT advance during the time course, and a complete elimination of both the Ptn transcription and the RT advance upon removal of Dox (Figs. 5E and EV5). We conclude that the TRE promoter-induced RT shift is rapid and reversible. Given that the same promoter at the same concentrations of Dox do not affect RT in the context of a reporter gene (Fig. 1), our results strongly suggest that it is elongation of transcription, and not initiation, that is eliciting the RT changes.

## Transcription is not necessary to advance the Ptn domain RT during differentiation

The results we have shown with TRE-driven short and long transcripts demonstrate directly that, when isolated as the only

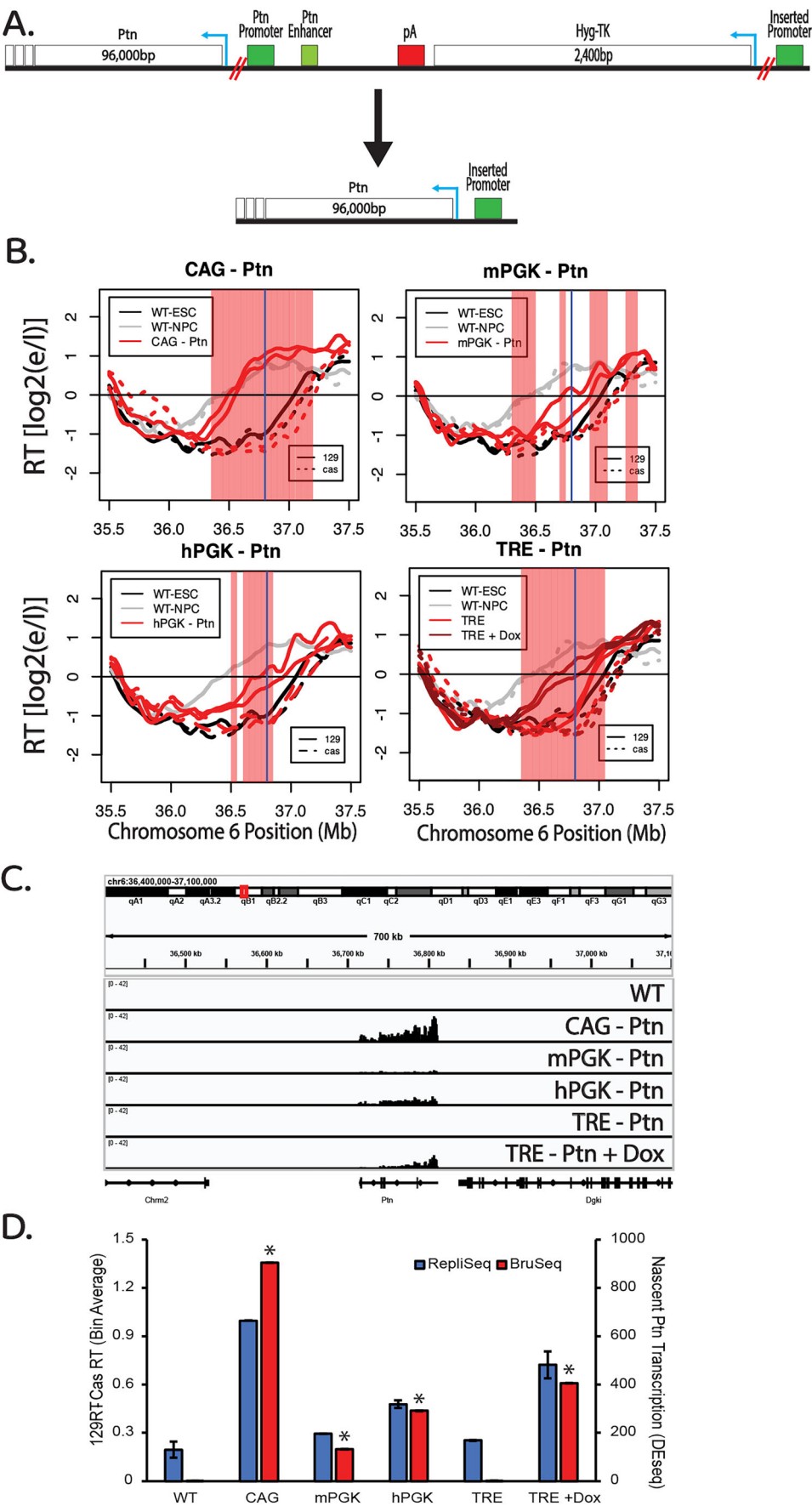

◄ **Figure 3. Transcription of a long transcript enhances the ability of promoters to advance RT.**

(A) Schematic of the region directly upstream of the Ptn gene. Deletions of the HTK selectable markers and the endogenous promoter and enhancers of the Ptn gene were made using CRISPR-cas9 targeting the loci marked by the double red lines. Blue arrows indicate the direction of the HTK and Ptn genes. (B) RT of the Ptn domain for the CAG-Ptn (top left), mPGK-Ptn (top right), TRE-Ptn (bottom left) and hPGK-Ptn (bottom right) cell lines, presented as in Fig. 1C. (C) Genome browser view of nascent RNA signal of the Ptn gene for each one of the four promoters. Biological replicates are shown in Fig. EV2. (D) Bar graph of the relationship between nascent Ptn transcription and the observed advance in the RT of the Ptn domain, presented as in Fig. 1E. Ptn expression was calculated as the fold change vs WT, based on 2 biological replicates for WT, mPGK, and CAG, and 3 biological replicates for hPGK, TRE, and TRE-Dox. The error bars for the Ptn expression bar graphs represent standard error. The asterisks indicate a significant difference ($p < 0.001$, calculated using the Wald test) in the levels of Ptn expression relative to Ptn expression in the parental cell line. The error bars in the RT data show the range of the difference among the 50 kb bins between the two biological replicates. Source data are available online for this figure.

variable, transcription can be sufficient to advance RT. The fact that a small TRE reporter gene cannot advance RT with the same promoter at the same location eliminates the possibility that binding of the rtTA transactivator on its own is sufficient to advance RT but, rather, the effect on RT depends upon transcriptional elongation. This raises the question as to whether the natural RT advance that accompanies Ptn induction during differentiation requires Ptn transcription, or whether other mechanisms, such as ERCEs, can advance RT independent of transcription (Turner et al, 2025; Sima et al, 2019). We identified the Ptn domain as an RT-switching domain over 20 years ago, and showed that its RT is advanced coincident with movement away from the nuclear periphery and activation of transcription (Hiratani et al, 2010, 2004). The coincidence was such that we were not able to delineate the order in which these correlated events occurred over the course of in vitro differentiation, even at one-day intervals, consistent with a direct link between transcription and RT. To directly address the causality between transcription and RT during differentiation, we deleted both the promoter and predicted enhancers for Ptn transcription (Fig. 6A), either on the *musculus* (129) allele alone or on both *musculus* and *castaneus* alleles, and differentiated these cells to neural precursor cells (mNPCs). Results of three independent experiments (Figs. 6B,C and EV6) demonstrated that the RT advance during differentiation still occurred despite elimination of all detectable transcription throughout the domain, detected either as total nascent RNA by BrU-seq (Fig. 6C) or as steady-state RNA by rRNA-depleted total RNA-seq (Fig. EV6). Ptn was the only RNA detected throughout this domain during NPC differentiation, consistent with prior reports (Giorgetti et al, 2016; Rivera-Mulia et al, 2018). The heterozygote configuration allowed us to conclude that both alleles advance similarly, despite the differential presence of Ptn gene induction. Thus, Ptn transcription is sufficient to advance RT in mESCs, but not necessary to advance RT during differentiation. The switch to early replication under these conditions must depend upon non-transcriptional mechanisms, such as ERCEs that may be activated specifically in neural precursor cells (Turner et al, 2025; Sima et al, 2019). Consistent with the nature of ERCEs, there are many sites of chromatin opening in the Ptn locus after differentiation that are not located within the deletion site (Fig. 6D).

## Genome-wide analyses confirm that transcription is not necessary and not always sufficient for an RT advance during a cell fate transition

While the studies above resolve discrepancies of prior ectopic manipulations, it is not clear how they relate to transcriptional

changes genome-wide during cellular differentiation. Our own prior studies identified late-replicating and expressed genes and even genes that are upregulated when becoming later replication (Hiratani et al, 2008, 2010; Rivera-Mulia et al, 2015). However, these studies measured steady-state levels of poly-adenylated mRNA. Indeed, we are not aware of any studies integrating changes in RT with total nascent RNA synthesis rates during a cell fate change. Moreover, most prior studies have examined the behavior of individual genes. Our findings that transcriptional elongation can drive an RT advance suggests that it may be the total domain-wide transcription that is important. Indeed, using steady state mRNA microarray datasets, correlations of transcription to early RT were stronger when integrated over larger genomic windows (Hiratani et al, 2008; MacAlpine et al, 2004). We thus re-investigated the necessity and sufficiency of transcription to advance RT during differentiation by performing genome-wide analyses using the Repli-seq and matching total nascent transcription (BrU-seq) datasets from the experiment shown in Fig. 6. Given that we had two replicates of two clones that each consisted of both *castaneus* and *musculus* genomes, we effectively had 8 replicate experiments each for mESCs and mNPCs, permitting rigorous statistical analyses.

We first asked whether we could find examples in the native genome where transcription is not necessary for an RT advance. To assess replication domain-wide changes in transcription, we analyzed total transcription per RT "switching region", defined as contiguous 50 kb windows that showed significant RT changes in mNPC as compared to mESC. We identified 766 such switching regions (Fig. 7A), containing zero (130), one (193) or more than one (443) gene(s) (Fig. EV7A). Of these, 375 were advanced in RT. As expected, 307 of those showed a significant ($p$-value $< 0.01$) change in transcription (Fig. 7B; circles), 302 of which increased. However, we also found 68 RT-advancing regions whose transcription was not significantly changed (Fig. 7B; triangles), many of which had a transcription level below the limit of detection ("noise threshold" defined in Fig. EV7B), as well as 5 that showed a decrease in overall transcription. Figure 7C shows an example of an RT-advancing locus with no annotated genes and three additional examples of regions with RT advance but no transcription induction are plotted in Fig. EV7C. Since the locus shown in Fig. 7C is neighboring another locus with both transcriptional induction and RT advance, we asked whether this gene-less domain might reside in the same sub-nuclear micro-environment as the active domain by examining their interaction frequencies in Hi-C datasets (Bonev et al, 2017). We found no significant contact between these domains (Fig. EV7D). These examples demonstrate that our finding that transcription is not necessary to advance RT,

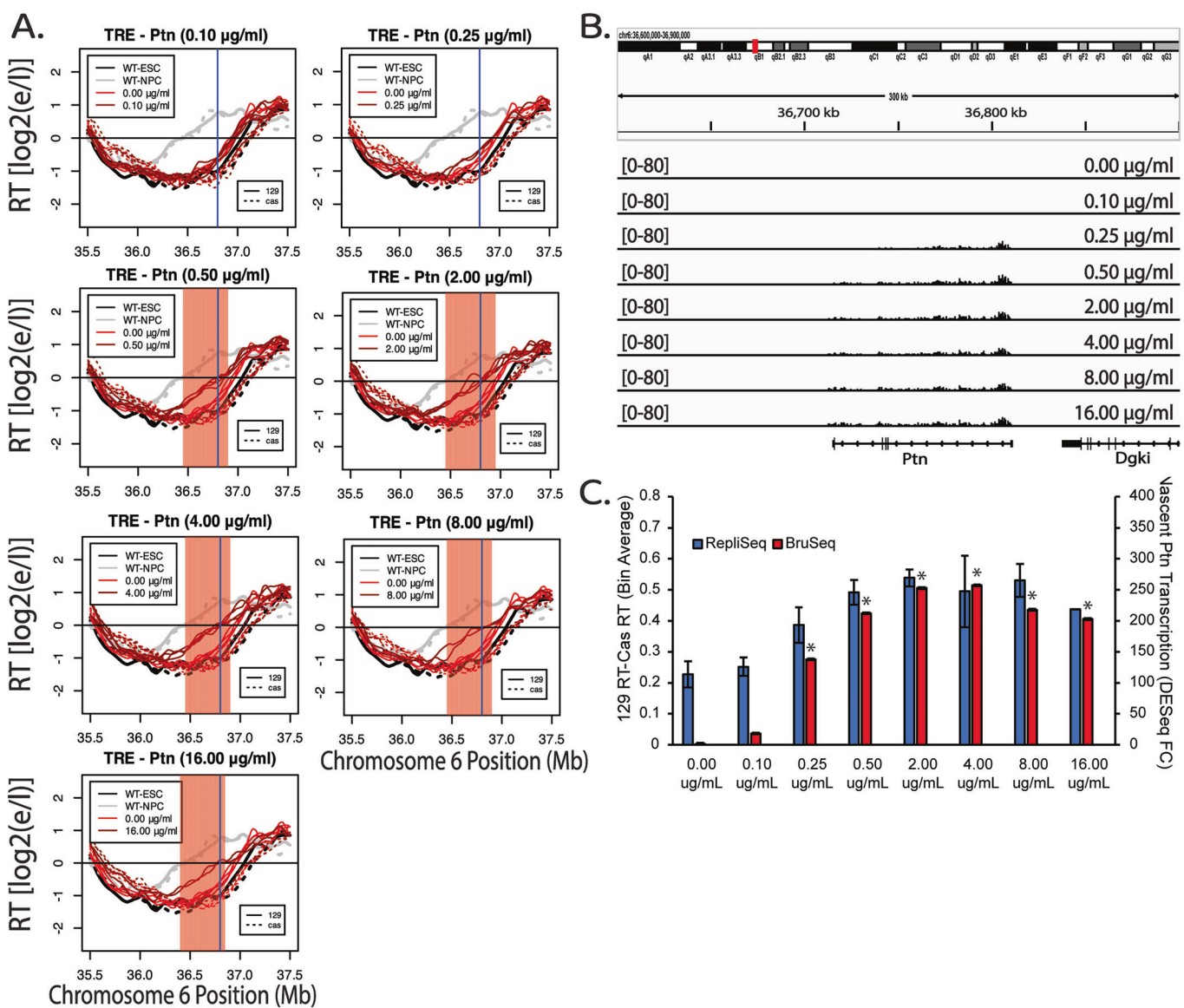

**Figure 4. Dosage-dependent advance of RT in response to transcriptional induction of the Ptn gene.**

(A) RT plots of the Ptn domain when TRE-Ptn cells are treated with increasing concentrations of Dox for 24 h, presented as in Fig. 1C. (B) Genome browser view of nascent RNA signal of the Ptn gene at each Dox concentration. Biological replicates are shown in Fig. EV3. (C) Bar graph of the relationship between nascent Ptn transcription and the observed advance in the RT of the Ptn domain, presented as in Fig. 1E. Ptn expression was calculated as the fold change vs WT, based on 6 biological replicates for 0 μg/mL, 0.1 μg/mL, 0.25 μg/mL, 0.5 μg/mL, and 2.0 μg/mL, 4 biological replicates for 4.0 μg/mL, and 2 biological replicates for 8.0 μg/mL, and 16.0 μg/mL. The error bars for the Ptn expression bar graphs represent standard error. The asterisks indicate a significant difference ($p < 0.001$, calculated using the Wald test) in the levels of Ptn expression relative to Ptn expression in the parental cell line. The error bars in the RT data show the standard deviation across 5 biological replicates for 0.0 μg/mL, and 4.0 μg/mL, and 6 biological replicates for 0.1 μg/mL, 0.25 μg/mL, 0.5 μg/mL, and 2.0 μg/mL, or the range of the difference among the 50 kb bins between 2 biological replicates, for 8.0 μg/mL, and 16.0 μg/mL. Source data are available online for this figure.

demonstrated via genetic manipulation of the Ptn locus (Fig. 6), can be extrapolated to endogenous, unmanipulated, loci.

Next, we asked whether there are cases where significant domain-wide transcriptional induction was not sufficient for an advance in RT. For this analysis, we could not define domains as regions of altered RT because we are interested in domains that do not change. Instead, we used our previously defined list of 3981 replication domains (RDs), defined as regions switching replication timing in at least one of 13 mouse cell types (Pope et al, 2014). We

found 697 RDs that show significant domain-wide transcriptional upregulation with transcription above noise in mNPCs, 247 of which were unchanged in RT and 103 that were delayed in RT (Fig. EV7F). Figure 7D shows the relative transcription rates in mESCs vs. mNPCs for RDs with significant transcriptional upregulation whose RT is either delayed or unchanged (350 domains), while Fig. 7E plots the domain-wide transcription vs. RT change for all RDs with significant transcription in either mESCs or mNPCs, or both (1673 domains). Despite a clear positive

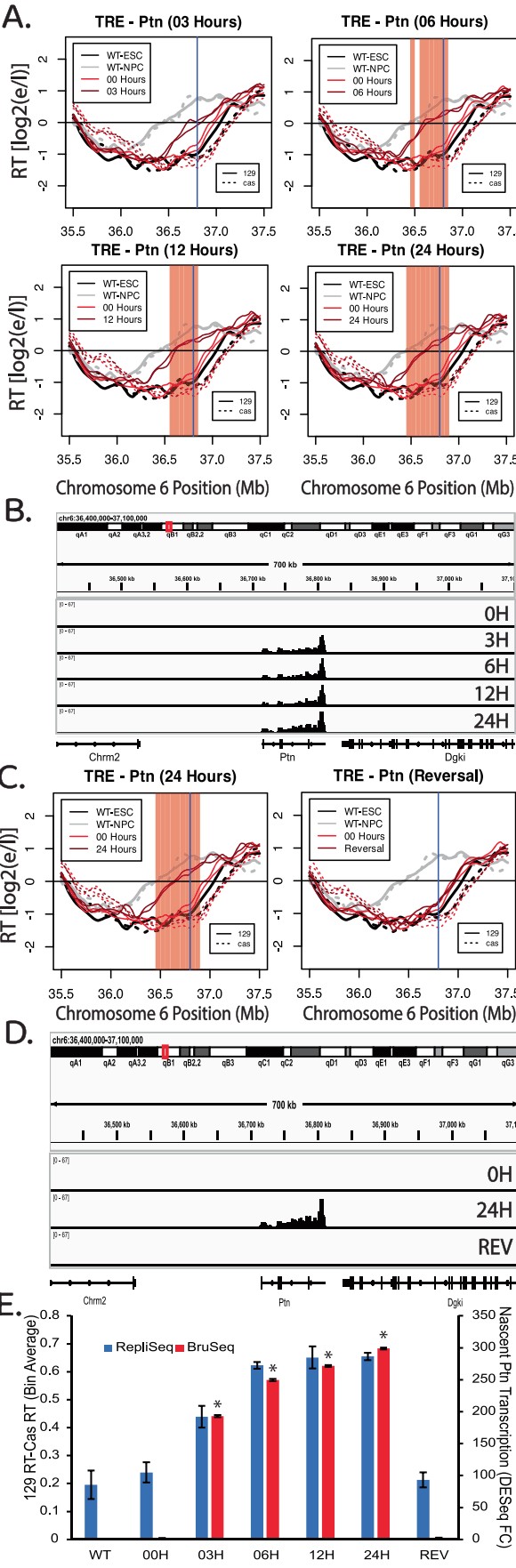

**Figure 5. The RT advance induced by the TRE-mediated transcription of the Ptn gene is rapid and reversible.**

(**A**) RT plots of the Ptn domain at different time points after the addition of Dox. BrdU (2 h) or BrU (30 min) was added at the end of each Dox incubation time period in the continued presence of Dox. In each plot, the black and gray lines indicate the RT of WT ESCs and NPCs, respectively. The red lines indicate the RT of the insert cell line with (dark red) or without (light red) Dox. Solid and dashed lines indicate the RT of the *musculus* and *castaneus* alleles, respectively. Vertical blue line indicates the +1 of the Ptn gene. The red shading indicates 50 kb windows with statistically significant differences in RT between WT 129 and the modified 129, as presented in Fig. 1C. (**B**) Genome browser view of the Ptn domain displaying nascent RNA signal of the Ptn gene at each time point. Biological replicates are shown in Fig. EV4. (**C**) RT plots of the Ptn domain after 24 h of Dox treatment and subsequent removal of Dox for an additional 24 h. In each plot, the black and gray lines indicate the RT of WT ESCs and NPCs, respectively. The red lines indicate the RT of the insert cell line with (dark red) or without (light red) Dox. Solid and dashed lines indicate the RT of the *musculus* and *castaneus* alleles, respectively. Vertical blue line indicates the +1 of the Ptn gene. The red shading indicates 50 kb windows with statistically significant differences in RT between WT 129 and the modified 129. (**D**) Genome browser view of the Ptn domain displaying nascent RNA signal of the Ptn gene at each condition in Fig. 5C. (**E**) Bar graph of the relationship between nascent Ptn transcription and the observed advance in the RT of the Ptn domain as presented in Fig. 1E. Ptn expression was calculated as the fold change vs WT, based on 2 biological replicates for 24 h, and 3 biological replicates for 0 h, 3 h, 6 h, 12 h, and Rev. The error bars for the Ptn expression bar graphs represent standard error. The asterisks indicate a significant difference (*p* < 0.001, calculated using the Wald test) in the levels of Ptn expression relative to Ptn expression in the parental cell line. The error bars in the RT data show the range of the difference among the 50 kb bins between the two biological replicates. Source data are available online for this figure.

correlation between transcriptional induction and RT advance (Fig. 7E), half of RDs that significantly upregulate transcription do not advance RT. Figure 7F,G shows examples of four RDs that were amongst those that experienced high transcriptional induction from undetectable levels in mESCs. Since small advances in very late replicating regions go undetected in log2(E/L) data, we co-plotted these four examples with our published high-resolution Repli-seq analysis (Fig. 7F) (Zhao et al, 2020). Two of these regions (left two in Fig. 7F) experienced small advances in RT, demonstrating that the numbers of domains experiencing transcriptional induction with no RT advance in our genome-wide analysis is an over-estimate. Nonetheless, given that over 350 RDs fell into this category, it is clear that domain-wide transcriptional induction is not always sufficient to advance RT. This is true even for quite long genes. For example, the 264,407 bp Mmp16 gene (chr4:17.8 Mb) is strongly induced in mNPCs, yet becomes slightly later replicating and is flanked by a gene-less domain that switches from initiating in mid-S TTR to being extremely late replicating.

To further investigate the sufficiency of transcription to advance RT, we re-investigated a set of 204 200 kb genomic segments previously identified as containing genes that are upregulated during the mESC to mNPC transition, with delayed or unchanging RT (Hiratani et al, 2010). Using our more statistically rigorous set of RT and nascent RNA measurements, and comparing to high-resolution Repli-seq, all but 14 of these regions were eliminated as either advanced in RT or having insignificant regional transcriptional changes (Fig. EV8A). Of those, most were TTRs, with groups of genes contributing to the overall regional transcription changes (Fig. EV8B). However, we did identify one region (Chr14:49.7-50.1) with one small (< 5 kb) upregulated gene with no RT advance flanked by a gene-less domain switching from early to late replication. The non-significance of transcriptional changes for most of these regions is likely because the prior study took a very different approach—using single gene transcription analyses, microarray vs. sequencing, and dividing the genome into sequential genomic segments instead of RDs—coupled with the considerably higher number of replicate experiments used here.

Altogether, we conclude that transcriptional upregulation and even transcriptional induction from the silent state is not always sufficient to detectably advance RT. Of course, we cannot rule out that our bulk cell population assays may miss a small population of cells that express these genes highly and advance RT, although it seems unlikely that could account for every case. Note that our

study was focused on regional transcription changes vs. advances in RT. It was also abundantly evident that transcription does not require and is not sufficient for early replication; many genes are expressed or upregulated while either remaining very late replicating or even becoming later replicating (Fig. EV9). Altogether, we have demonstrated that, despite the strong correlation of early replication with transcription, transcription is not necessary and not always sufficient to advance RT and early RT is not necessary for transcription.

## Discussion

The RT program is essential for maintenance of the global epigenome, strongly correlated with other structural and functional features of chromosomes and is regulated during development. Therefore, it is likely that changes in RT during differentiation are an integral part of organismal development, but we have little understanding of how they are executed. One longstanding hypothesis has been that transcription is sufficient for early replication and potentially drives RT changes during development (Goldman et al, 1984; Hatton et al, 1988; Therizols et al, 2014). If true, then such changes, affecting large replication domains and in some cases multiple genes, could alter domain-wide epigenome architecture to assemble a chromatin environment more favorable for transcription throughout the domain (Klein et al, 2021). As attractive as this model is, it is challenged by both manipulative studies and genomics data that provide numerous contradictory examples of late replicating and expressed genes discussed in the introduction. Here, we have taken a reductionist approach to systematically alter single variables that could reconcile these contradictions. At this single locus, we find that transcriptional elongation can be sufficient to advance RT depending upon the length and strength of the resulting transcript; a small reporter gene can be transcribed with no detectable effect on RT but when driven by promoters that drive high levels of transcription, can advance RT. A large transcription unit can advance RT even at low rates of transcription and the RT shift is positively correlated with transcription rate. However, we also show that transcription is entirely dispensable for a developmental RT shift, demonstrating that non-transcriptional mechanisms can be equally effective at driving an RT advance during differentiation. In a genome-wide survey, we find numerous examples of abundant transcription in

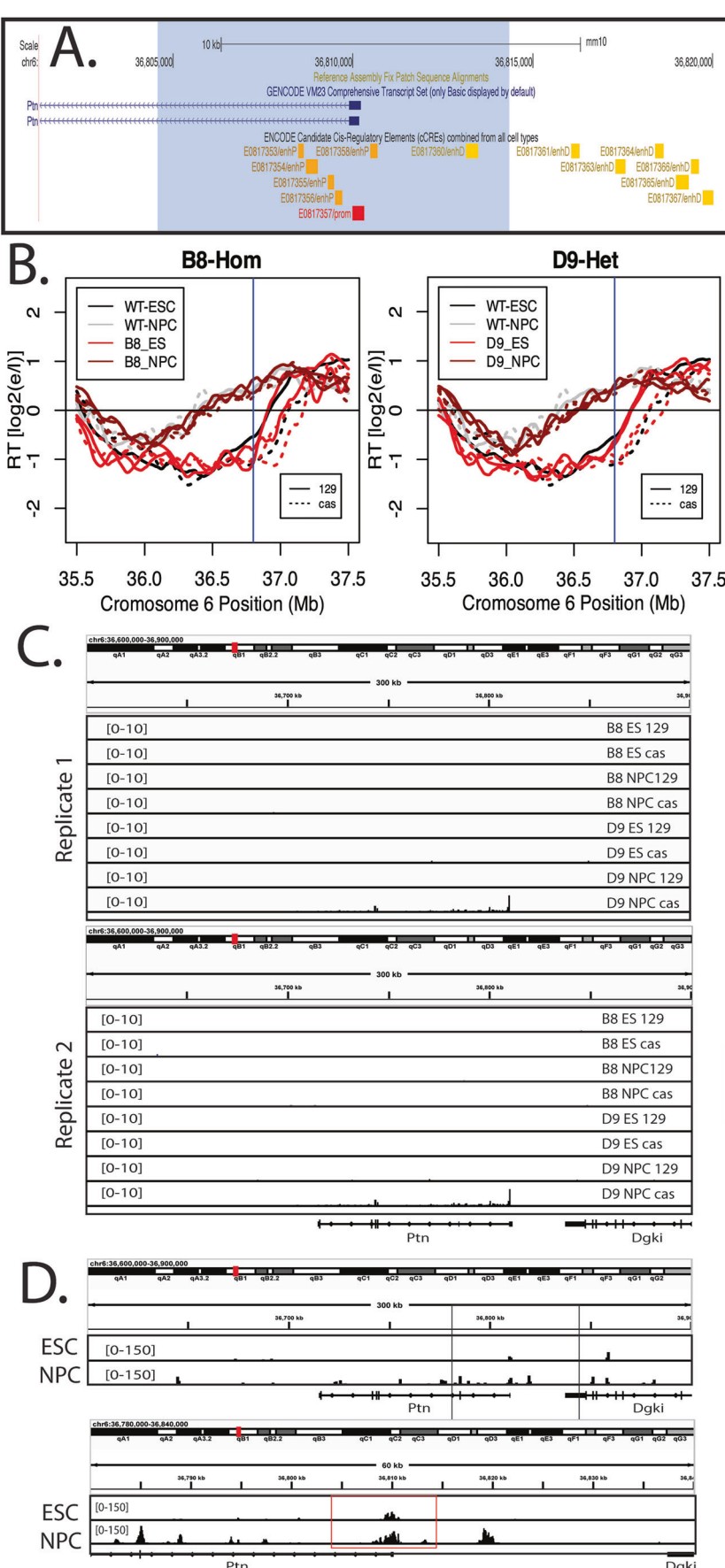

**Figure 6.  Transcription is not necessary to advance the Ptn domain RT during differentiation.**

(A) UCSD Genome Browser track indicating the part of the Ptn domain that was deleted. The deleted sequence is highlighted by the blue shade. Candidate promoter, proximal enhancers and distal enhancers are indicated by the red, orange and yellow boxes. (B) RT plots of clones B8 and D9, in which the Ptn promoter deletion is homozygous and Heterozygous, respectively. Dark gray and light gray indicate the RT profiles of WT ESCs and NPCs, respectively. Light red and dark red indicate the RT profiles of ESC and NPC deletion clones, respectively. Solid and dashed lines indicate the RT of the musculus and castaneous alleles, respectively. Vertical line indicates the site of the deletion. (C) Browser track of BrU-seq for WT NPCs and the two deletion clones. (D) Browser track of ATAC-seq changes upon differentiation of mESCs to mNPCs relative to the promoter/enhancer deletion (Nazim et al, 2024). The top track shows the entire Ptn domain, with the region delineated with vertical lines expanded in the bottom track, which shows the location of the Ptn promoter/enhancer deletion in a red box. A prominent ATAC-seq site appears upstream of the deletion after differentiation, as well as several sites within the body of the gene, that could be transcription-dependent and one site approximately 70 kb downstream of the gene body. Source data are available online for this figure.

late-replicating domains, RT advances without transcriptional induction, and transcriptional upregulation that is not sufficient to advance RT, demonstrating that multiple, context-dependent, mechanisms can influence relationships between transcription and RT at any given locus. Together, these experimental results provide an empirical rationale to reconcile data in the literature relating transcription and RT.

## Length and strength explain much of the literature with ectopic transcription

Our results performing systematic manipulations at the Ptn gene replication domain provide a solid empirical base with which to re-evaluate past studies of RT and transcription using ectopic vector insertions (see Introduction). Small vectors may or may not advance RT, while long transcripts are more effective at doing so. Ectopic inserts that do not induce transcription (Goren et al, 2008) may advance RT through alternative mechanisms. Our results extend observations in chicken DT40 cells (Blin et al, 2019). While this study was focused on the effects of RT on fragile site expression, and the design of the experiments did not directly compare transcription length, strength and RT at the same locus, their results were consistent with those reported here. An apparent contradiction with our results was that Blin et al did not find any significant effect of transcriptional induction by the TRE promoter even when driving a long transcript. There are several differences in our approaches that could account for this discrepancy. First, Blin et al detected a small advance in RT prior to induction and did not titrate their induction to determine whether they had achieved maximum transcription thereafter, thus it is possible that they had leaky transcription prior to induction and/or did not fully induce the TRE promoter. Second, steady state transcription was measured rather than nascent transcription rates as we have done. Third, RT relative to the wild-type locus was measured in different cell lines; our direct allelic comparison in a hybrid-genome cell line offers much improved sensitivity to detect RT changes. Fourth, RT and transcription were assayed by single locus PCR quantification, which in our hands can be misleading. Here, we have measured whole-domain RT and total nascent transcription, with and without induction, at different times and concentrations of inducer, and relative to an internal wild-type control allele. Lastly, our genome-wide survey demonstrates that effects of transcriptional induction on RT are domain-dependent, hence it is possible that the locus studied by Blin et al does not respond to transcriptional induction. Altogether, our results can account for all of the discrepancies in the literature regarding ectopic manipulations of genomic sites to study the relationship of RT to transcription, by

demonstrating that transcription is not always sufficient, and transcription is not always necessary to elicit an RT advance.

## Causality is linked to transcriptional elongation

While our results do not unveil detailed mechanisms by which transcription can advance RT, they do focus our attention on the elongation step of transcription. Most compellingly, we show that an artificial trans-activator, as the only variable, can be targeted to a small DNA sequence to induce transcription of a small gene with no effect on RT, while the same targeting event to the same small DNA sequence, when inducing a larger gene, substantially advances RT. It is therefore unlikely that the effects on RT are indirectly related to Tet trans-activator binding and its downstream effects on local chromatin or recruitment of the transcription initiation machinery per sé. Rather, the RT advances are more likely a consequence of transcriptional elongation through a larger segment of chromatin. Further, the fact that the RT advance reaches a dose-dependent steady state within a few hours, and is rapidly reversible, suggests a direct dependence upon transcription itself. We also observed that short transcripts capable of advancing RT can drive detectable, albeit low level, read-through transcription beyond the vector polyA sequence, again implicating the extent of transcriptional elongation. One hypothesis worth investigating is that more extensive transcription may remodel large segments of chromatin, possibly by creating domains of DNA supercoiling (Naughton et al, 2013) or reducing interactions with the nuclear lamina. In fact, small hPGK-driven reporter genes were found to have only moderate and localized (~ 20 kb) effects on interactions of flanking chromatin with the lamina while longer genes had more extensive effects (Brueckner et al, 2020). Another possibility is that longer transcripts tend to be spliced and since proximity to speckles, the sub-nuclear storage compartments for splicing machinery, is highly correlated with early replication, splicing may increase proximity of domains towards the speckles (Bhat et al, 2024; Chen et al, 2018).

The rapid nature of the RT change after induction of transcription suggests that RT changes can occur after the functional loading of inactive Mcm helicases onto chromatin in telophase/early G1 (Okuno et al, 2001; Dimitrova et al, 2002, 1999), and possibly after the early G1 phase Timing Decision Point (Dimitrova and Gilbert, 1999) or even after S phase begins. Transcription of large genomic segments could move the irreversibly loaded Mcm helicases (Kuipers et al, 2011) in such a way that creates a high-efficiency initiation zone (Sasaki et al, 2006). What is needed to test this hypothesis are effective means to map the positions of Mcm double hexamers during the cell cycle (Li et al, 2023). Regardless, our results provide a system for

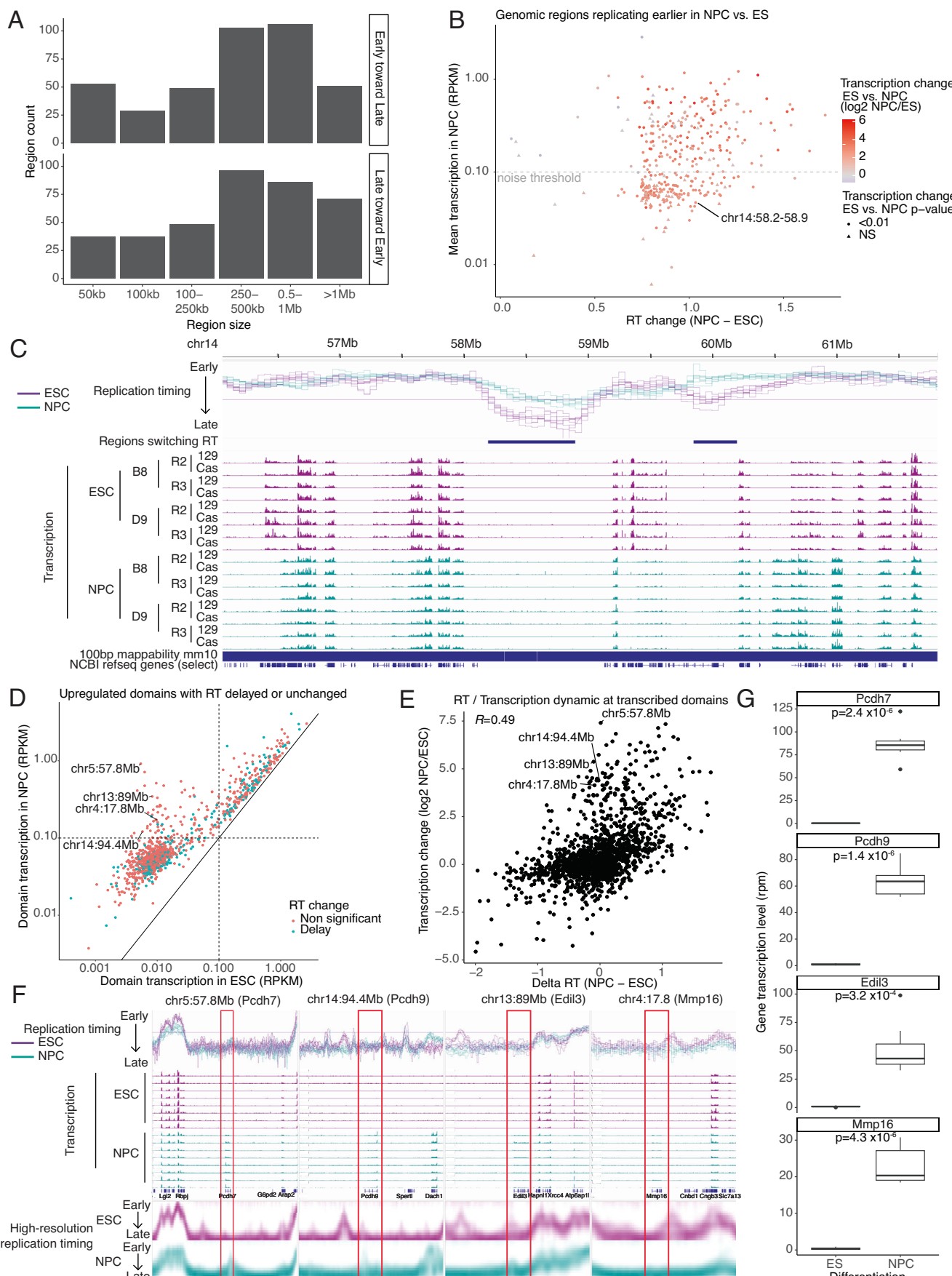

◄ **Figure 7. Transcription is not necessary and not always sufficient for an RT advance during the ESC to NPC transition.**

(A) Size distribution of the regions with significant RT changes. (B) Dotplot of regions with significant (t-test) RT advance in ESC to NPC transition. Each dot represents one region, defined by stitching together consecutive 50 kb windows of significant RT advance. X-axis represents the advance in RT with higher value corresponding to higher RT advance. Y-axis represents the average transcription coverage in the region in NPCs (read per kb per million reads, RPKM). Color-scale represents the log2 transcription fold change between ESCs and NPCs in the region and the symbol represents the p-value assessing the significance of the transcription change (t-test). The Y-axis contains a "noise threshold" for transcription in NPCs, determined as described in Fig. EV7B. (C) RT and transcription tracks for region chr14:58.2-58.9 that is indicated in (B). This region has no annotated genes using the mm10 RefGene list from UCSC. Tracks represent, from top to bottom: Genomic coordinates, RT (log2 ratio Early/Late), regions with significant RT change, transcription coverage (rpm, with a maximum scale set to 10 rpm) in ESC and NPC datasets, mm10 mappability for 100 bp reads, NCBI RefSeq genes. Data from ESC are in purple and data from NPC are in teal. (D) Total replication domain (RD) transcription vs. RT change, for the 350 domains that showed significant transcription upregulation and with RT either delayed or unchanged. Each dot represents one RD, defined as in (Pope et al, 2014). X-axis represents the average transcription coverage in the region in ESC (read per kb per million reads, RPKM). Y-axis represents the average transcription coverage in the region in NPC (read per kb per million reads, RPKM). (E) Total domain transcription change vs. RT change for the 1673 domains that showed significant transcription in either mESCs, mNPCs or both. Each dot represents one RD. X-axis represents the change in RT in the whole domain, with higher value corresponding to higher RT advance. Y-axis represents the log2 transcription fold change between mESCs and mNPCs in the whole domain. (F) RT and transcription tracks for the loci around the domains indicated in (D) and (E). Tracks represent, from top to bottom: RT (log2 ratio Early/Late), transcription coverage (rpm, with a maximum scale set to 10 rpm) in ESC and NPC datasets, NCBI RefSeq genes and high-resolution Repli-seq. Data from ESC are in purple and data from NPC are in teal. (G) Boxplots of gene transcription coverage in rpm for genes in regions in (F). Boxplots represent the median and interquartile range (IQR); whiskers mark 1.5x the IQR; data beyond 1.5x the IQR are plotted as individual points. Statistics by t test. Number of biological replicates (n) = 8. Source data are available online for this figure.

inducible RT shifts to investigate mechanisms linking RT and transcription.

## Transcription is not necessary to advance RT during a cell fate change

By deletion of the regulatory elements inducing the only gene in the Ptn replication domain, we directly demonstrate that transcription is not necessary for a differentiation-induced RT advance. Some other mechanism, perhaps developmentally-activated ERCEs present in the Ptn domain (Sima et al, 2019; Turner et al, 2025), can elicit the RT switch in the absence of transcription. This is not surprising, in light of recent results demonstrating that ERCEs can maintain early RT in the absence of transcription (Turner et al, 2025). In fact, new sites of open chromatin, consistent with the properties of ERCEs, do appear outside of the deleted Ptn transcription control elements after differentiation (Fig. 6D). The necessity and sufficiency of these sites to advance RT independent of transcription will be important to follow up. Indeed, transcription and ERCEs could form a powerful positive feedback loop to drive cell fate changes. Either transcription or ERCEs could initiate an RT switch, followed by, respectively, binding of core TFs to activate latent ERCEs or a higher level of transcriptional activation following an ERCE-driven RT shift (Turner et al, 2025). Together, this could serve to lock in epigenetic states in a step-wise fashion, or to provide flexible alternatives during a cell fate transition. Experimental approaches that can uncouple transcription induction from core TF binding during a natural cell fate transition will be needed to investigate the separate roles of these different regulatory mechanisms.

## Genome-wide studies reveal context dependence of the RT-transcription liaison

Genomics results surveying natural relationships between RT and transcription in different tissues and differentiation systems have uncovered numerous examples of large late replicating and expressed genes (Heskett et al, 2022; Rivera-Mulia et al, 2015; Hiratani et al, 2010). Most of these datasets score only steady-state RNA and do not take into account the effects of neighboring genes

within the same domains. Indeed, even with steady-state RNA measurements, when regional transcription has been taken into account the correlation between RT and transcription increases (Hiratani et al, 2008; MacAlpine et al, 2004). Domain-wide analysis is a difficult task, because the units of RT regulation can only be deciphered by tracking the boundaries of RT changes across multiple cell types (Pope et al, 2014). For these reasons we re-evaluated the genome-wide relationship between RT and transcription during differentiation, examining nascent RNA, summing the total transcription within the units of defined RT changes, and using 8 replicate assays each in mESCs and mNPCs. Moreover, since standard log2(E/L) RT measurements would miss any partial changes in RT that remain in the first or second half of S phase, we also examined key examples of a lack of RT change using published high-resolution 16 fraction Repli-seq datasets (Zhao et al, 2020).

Using this statistically rigorous and biologically stringent approach, we found that, despite the expected strong correlation between early replication and transcription: (1) early replication does not require transcription; (2) robust transcription can occur within very late replicating regions; (3) transcriptional induction of both small and large genes is not always accompanied by advances in RT. Looking at many individual examples, the overarching principle is that RDs have widely different numbers of genes of varied regulation, flanked by RDs that have varied RT responses to differentiation. Together, these observations suggest that the relationship between RT and transcription, while positively correlated, is highly context dependent, making it impractical to apply simple rules of engagement that apply at any given domain.

## Reporter genes that do not affect RT provide useful tools

Our work also identified promoters that can drive ectopic gene expression in either a constitutive or inducible fashion, and function well as selectable markers, without affecting RT. Since RT is so closely related to sub-nuclear position and chromatin compartment, relationships that are also poorly understood, these markers may prove useful in applications where integration is desired without affecting RT, for example, to study large-scale chromosome structure and function (Wang et al, 2021b; 4D Nucleome Consortium et al, 2024; Ryba et al, 2010).

# Methods

### Reagents and tools table

| Reagent/Resource | Reference or Source | Identifier or Catalog Number |
|---|---|---|
| **Experimental models** | | |
| F121-9 | Rivera-Mulia, 2018 | |
| F121:R26puroR-M2rtTA | Quinodoz, 2021 | |
| CM1417 | Brueckner, 2020 | |
| CM1420 | Brueckner, 2020 | |
| **Recombinant DNA** | | |
| pX458 | Addgene | Plasmid #48138 |
| **Antibodies** | | |
| Anti-BrdU | BD | 555627 |
| **Oligonucleotides and other sequence-based reagents** | | |
| gRNA cloning oligos | This study | Tables 1, 3 |
| PCR Primers | This study | Table 2 |
| **Chemicals, Enzymes and other reagents** | | |
| P3 Primary Cell 4D Nucleofector X Kit | Lonza | V4XP3024 |
| XFect Transfection Reagent | Takara | 631318 |
| BrdU | Sigma-Aldrich | B5002 |
| NEBNext Ultra Library DNA Prep Kit for Illumina | New England Biolabs | E7370 |
| BrU | Sigma-Aldrich | 850187 |
| Direct-Zol RNA Miniprep Plus Kit | Zymo | R2070 |
| NEBNext® rRNA Depletion Kit v1 | New England Biolabs | E6310 |
| NEBNext Ultra II RNA First Strand Synthesis Module | New England Biolabs | E7771 |
| NEBNext Ultra II Directional Second Strand Synthesis Kit (NEB) | New England Biolabs | E7550 |
| NEBNext Ultra II DNA Library Prep Kit for Illumina | New England Biolabs | E7645 |
| Hygromycin B | InvivoGen | ant-hm-5 |
| Ganciclovir | Sigma-Aldrich | G2536-100MG |
| Doxyxycline hyclate | Sigma-Aldrich | D9891-1G |
| Q5 High-Fidelity DNA Polymerase | New England Biolabs | M0491S |
| **Software** | | |
| Bowtie2 | Langmead and Salzberg, 2012 | |
| SNP-split | Krueger and Andrews, 2016 | |
| Samtools | Li et al, 2009 | |
| Bedtools | Quinlan and Hall, 2010 | |
| R | https://www.r-project.org/ | |
| Tophat | https://ccb.jhu.edu/software/tophat/index.shtml | |

| Reagent/Resource | Reference or Source | Identifier or Catalog Number |
|---|---|---|
| STAR | Dobin, 2013 | |
| DESeq2 | Love, 2014 | |
| featureCounts | Liao, 2014 | |
| EdgeR | Robinson, 2010 | |
| TrimGalore | https://www.bioinformatics.babraham.ac.uk/projects/trim_galore/ | |
| IGV | Robinson, 2010 | |
| RepliPrint | Ryba, 2011 | |
| **Other** | | |

## Cell culture

F121:R26puroR-M2rtTA mESCs (Quinodoz et al, 2021), expressing the tetracycline transactivator from the Rosa22 locus, and all mutant cell lines derived from the parental cell line, were cultured, maintained, and passaged according to the 4DNucleome SOP (https://data.4dnucleome.org/protocols/cb03c0c6-4ba6-4bbe-9210-c430ee4fdb2c/) for F121:R26puroR-M2rtTA unless otherwise stated. For Dox-induction, doses ranged from 0 to 16 µg/ml of Dox for 24 h. Mild toxicity and changes in genome-wide transcription were observed at 8 µg/ml and more so at 16 µg/ml; higher doses of Dox led to significant rates of cell death making it challenging to collect enough material for RT and transcription assays. ESCs were differentiated into NPCs using the 4DNucleome differentiation protocol (https://data.4dnucleome.org/protocols/fa439403-03b2-4ef4-be5f-3c76fb2e8e1a/).

## Ectopic insertions

F121:R26puroR-M2rtTA mESCs were nucleofected with a donor plasmid containing one of the four promoters used, a hygromycin resistance/thymidine kinase fusion gene and a transcription termination sequence, flanked by homology arms targeting sequences directly upstream of the Ptn promoter on chromosome 6. The donor plasmid was co-transfected with two modified Px458 cas9 plasmids containing two guide RNAs (Table 1) targeting sequences upstream of the Ptn promoter, and a plasmid expressing mCherry used to assess the success of the nucleofection. Nucleofected cells were cultured for two days and then treated with 350 µg/mL of Hygromycin. Colonies of hygromycin-resistant clones were isolated and plated into 96-well plates 7–10 days after the addition of Hygromycin. Colonies were expanded into multiple 96-well plates and screened by PCR with a set of primers (Table 2), one targeting the genomic sequence flanking the homology arms and the other targeting the inserted sequence. To screen the colonies, DNA was isolated by removing the cell culture media, lysing the cells overnight at 60 °C in 150 µL of lysis buffer (10 mM Tris pH 7.5, 10 mM EDTA, 10 mM NaCl, 0.5% sarcosyl, and 1 mg/mL fresh proteinase K). To precipitate the DNA, 150 µL of ice-cold precipitation buffer (1.5 mL 5 M NaCl in 100 mL of 100% ethanol) was added to the wells, followed by a 30 min incubation at room temperature, and a spin at 3000 rpm for 5 min, at 4 °C.

**Table 1.  gRNAs for ectopic insertion of HTK expression cassettes and deletion of HTK**

| Target | Guide sequence | Forward primer | Reverse primer | PAM Sequence Start | PAM Sequence End |
|---|---|---|---|---|---|
| Ptn_5_Insertion_Guide | TCAGTCCTCTGGTGAATAGAC | CACCgtcattcaccagaggactga | AAACtcagtcctctggtgaatagac | chr6:36,811,898 | chr6:36,811,900 |
| Ptn_3_Insertion_Guide | AGTTCTCCCAAAATCTAAAT | CACCgAGTTCTCCCAAAATCTAAAT | AAACATTTAGATTTTGGGAGAACTc | chr6:36,812,276 | chr6:36,812,278 |
| Ptn_5_Deletion_Guide | GAGAAGAAGCAGGCTGTGCG | CACCGAGAAGAAGCAGGCTGTGCG | AAACCGCACAGCCTGCTTCTTCTC | chr6:36,810,222 | chr6:36,810,225 |
| Ptn_3_Deletion_Guide | TCAGTCGCCCTGTTGACAAT | CACCgATTGTCAACAGGGCGACTGA | AAACTCAGTCGCCCTGTTGACAATc | Inserted Sequence | |

**Table 2.  PCR primers to confi rm ectopic insertion of HTK expression cassettes and deletion of HTK**

| Primer name | Primer sequence |
|---|---|
| Insertion Screening Primer Fwd | ATATCCACGCCCTCCTACATCG |
| Insertion Screening Primer Rev | AGGGAGATGGGTTAAGCGGA |
| Insertion Screening WT Primer Fwd | TATCCCGTCAGTCCTCTGGTGA |
| Insertion Screening WT Primer Rev | AGGGAGATGGGTTAAGCGGA |
| Insertion Screening CAG Primer Fwd | CAGGCTATGCACATCCCTGT |
| Insertion Screening CAG Primer Rev | GGGCTACTTGCCAATACGGT |
| Deletion Screening Primer Fwd | GCATCCCAACCAAGCAAAGG |

**Table 3.  gRNAs used to create Ptn promoter deletion cell lines**

| Guide name | Guide sequence | Guide coordinate |
|---|---|---|
| Fig 6_Ptn_Del_Guide_1 | tgagtggagctactacagat | chr6:36804675 |
| Fig 6_Ptn_Del_Guide_2 | tgagaagtttatttggccta | chr6:36814363 |

Precipitated DNA was washed twice with 200 µL of ice-cold 70% EtOH, air-dried for 10 min and resuspended in 40–200 µl of $H_2O$, depending on the size of the original colony. PCR reactions were conducted using Q5 high-fidelity DNA polymerase in 12.5 µL reactions, using 1 µL of extracted DNA from each colony, for 35 cycles. Annealing temperatures used corresponded to each individual primer set. Extension time for all reactions was set at 30 s. A different set of screening primers was used for the insertion of the CAG promoter due to difficulties with the PCR amplification of the CAG promoter. Clones were also screened with a wild type set of primers, one targeting the genomic sequence flanking the homology arms and the other targeting the sequence between the two guide RNAs. A positive on both screens indicated that the insertion was made on a single copy of chromosome 6.

## Deletions

Cell lines with ectopic insertions were nucleofected with two modified Px458 cas9 plasmids containing two guide RNAs (Table 1), one targeting a sequence between the inserted promoter and the inserted hygromycin resistance gene and the other targeting a sequence directly upstream of the +1 of the Ptn gene. Two days after the nucleofection, cells were treated with 100 µg/ml of Ganciclovir. Colonies of Ganciclovir-resistant clones were isolated and transferred to 96-well plates 10–14 days post treatment. Clones were expanded into multiple 96-well plates and screened with a set of primers (Table 2), one targeting the inserted sequence upstream of the guide RNA targeted sequence and the other targeting a genomic sequence downstream of the +1 of the Ptn gene. DNA extraction and screening conditions were similar to the insertions. Ptn promoter deletion cell lines (clone B8 and D9) were made as previously described (Sima et al, 2019) using gRNAs in Table 3.

## E/L Repli-Seq

E/L Repli-Seq was performed as previously described (Marchal et al, 2018). Briefly, cells were labeled with 100 µM BrdU for 2 h and

then fixed in 75% cold ethanol. Fixed cells were FACS sorted into early-S and late-S fractions based on DNA content. Genomic DNA, from 40,000 cells per fraction, was isolated, sheared using Covaris ME220, and adapter-ligated with NEBNext Ultra Library DNA Prep Kit. These adapter-ligated DNA fragments were enriched for nascent BrdU-labeled DNA via immunoprecipitation with anti-BrdU antibody, PCR amplified, indexed, and sequenced on HiSeq2500 or NovaSeq6000 aiming to obtain ≥30 million clusters of ≥50 bp reads per library.

## E/L Repli-Seq analysis

Raw sequencing reads were aligned to an mm10 reference genome with bowtie2 and subsequently musculus and castaneous specific reads were isolated using SNPsplit (Krueger and Andrews, 2016). Aligned and parsed reads were then quality filtered and depleted of PCR duplicate reads. To calculate the statistical significance of RT differences across samples, processed early and late fraction reads were binned into 50 kb windows, genome-wide and the log2 ratio of early to late fraction was calculated for each window, which was then quantile normalized to the RT of WT F121 cells. Statistical significance of RT changes for all windows in each sample, relative to WT, were calculated using a Monte Carlo algorithm as part of RepliPrint (Ryba et al, 2011), with a p-value of 0.01 used as the cut-off for windows with statistically significant differences. To generate RT plots, reads were binned into 5 kb windows, and the early-to-late log2 ratio was quantile normalized to the RT of WT F121 cells, scaled and smoothed in 300 kb bins. It is important to note that scaling, which was applied on the data to generate RT plots but not for calculating statistical significance, alters the dynamic range of the data (~−3 – ~3) from the original dynamic range (~−7 – 7). This means that RT values in RT line plots are not comparable to the RT values in the rest of the plots.

## Bru-Seq

Bru-Seq libraries were prepared according to previously published protocols (Paulsen et al, 2014). Cells were labeled for 30 min with Bromouridine (Sigma-Aldrich, 850187). BrU-labeled cells were collected and total RNA was isolated from the cells using the Direct-zol RNA Miniprep Plus kit (Zymo, R2070). BrU-labeled RNA was isolated using a Bromodeoxyuridine antibody (Sigma-Aldrich, B5002) conjugated to goat anti-mouse IgG Dynabeads (Thermo-Fisher, 11033). Nascent BrU-labeled RNA was fragmented at 85 °C for 10 min and reverse transcribed into cDNA using NEBNext Ultra II RNA First Strand Synthesis Module (NEB, E7771) and the NEBNext Ultra II Directional RNA Second Strand Synthesis Module (NEB, E7550). Libraries were prepared from cDNA using the NEBNext Ultra II Library Prep kit (NEB, E7645), indexed and sequenced on NovaSeq6000 to obtain ≥80 million of 100 bp single end reads.

## Bru-Seq analysis

Raw sequencing reads were aligned to the mouse ribosomal DNA with bowtie2, keeping only reads that did not align to the ribosomal DNA. Non-ribosomal reads were subsequently sequenced to the mm10 genome, using tophat aligner and the following flags {--min-

isoform-fraction 0 --max-multihits 1 --no-coverage-search --library-type fr-unstranded --bowtie-n --segment-mismatches 1 --segment-length 25}. The Bru-Seq data for the cell lines with the HTK insertions were aligned to a custom genome in which the Mitochondrial DNA sequence was replaced with the HTK sequence. Allele specific reads were isolated using SNPsplit. Bru-Seq tracks for visualization were generated using the STAR RNA-seq aligner (Dobin et al, 2013). Counts tables were generated using featureCounts (Liao et al, 2014). Differential expression analysis, including the calculation of statistically significant differences in expression, was conducted using the R package DESeq2 (Love et al, 2014). DESeq2 uses the Wald test to calculate statistical significance. Statistical significance was calculated relative to HTK expression in the parental cell line, which is expected to be zero, since the parental line does not have an HTK insertion.

## RNA-seq

From 300 ng total RNA extracted using Direct-zol RNA MiniPrep (Zymo research R2050), ribosomal RNA was depleted using NEBNext® rRNA Depletion Kit v1 (NEB E6310). The entire resulting RNA was used for 1st strand synthesis by NEBNext Ultra II RNA First Strand Synthesis Module (NEB, E7771), following "Directional RNA-seq Workflows". Second strand was synthesized using NEBNext Ultra II Directional Second Strand Synthesis Kit (NEB E7550) then cDNA was made into sequencing library using NEBNext Ultra II DNA Library Prep Kit for Illumina (NEB E7645).

## RNA-seq analysis of NPC data

RNA-seq reads were aligned to an mm10 genome with masked 129/Cas SNPs, using the STAR RNA-seq aligner (Dobin et al, 2013). Reads aligned to mm10 were subsequently parsed into 129 and Cas specific reads using SNPsplit. Browser tracks were generated using the inputAlignmentsFromBAM function of the STAR RNA-seq aligner (Dobin et al, 2013).

## CM Insertion RNA-seq data analysis

Barcodes were matched to 61/107 insertions in the CM1417 and CM1420 clones. To assess the abundance of these barcodes in the RNA-seq datasets, reads containing the exact 16 nucleotide barcode were extracted from the raw FastQ files and counted. Barcode counts were normalized by the total number of sequencing reads of each replicate (RPM). Separately, to assess the average levels of GFP expression per insertion in each clone, reads were mapped onto a custom reference genome containing the sequence of the GFP gene in chromosome M, using the STAR RNA-seq Aligner (Dobin et al, 2013). Count tables for each clone were prepared using feature-Counts (Liao et al, 2014). The RPKM of the average counts across two replicates was calculated using the EdgeR R package (Robinson et al, 2010). The RPKM of GFP for each clone was divided by the number of insertions present in that clone. Subsequently, the log2 of the RPKM for each gene was calculated.

## Genome-wide ESC-NPC transition repli-seq analysis

Repli-seq sequencing data have been analyzed by trimming adapters (TrimGalore), mapped on mm10 genome with masked

Cas/129 variants (Bowtie2), quality filtered and duplicate removed (samtools), parsed for Cas and 129 genomes (SNPsplit). RT has been produced by computing the log2 ratio of the normalized coverage on 5 kb windows (zoom on Ptn locus, Fig. 6) and 50 kb windows (RT/transcription genome-wide integration, Fig. 7) of early and late fractions and scaled and smoothed (https://github.com/ClaireMarchal/ScalingSmoothing). Region with significant RT changes have been identified by calling windows with significant RT changes between ESC and NPC (q-value threshold of 0.001) using SwitchRT (https://github.com/ClaireMarchal/SwitchRT) and stitching windows with significant RT changes allowing 300 kb gap (bedtools).

### Genome-wide ESC-NPC transition BrU-seq analysis

BrU-seq data have been analyzed by removing reads mapping on ribosomal DNA (mapping on NCBI BK000964.3 mouse rDNA using bowtie and keeping unmapped reads), mapping non-rDNA reads on mm10 genome with masked Cas/129 variants associated to mm10 reference genes (Tophat2), parsed for Cas and 129 genomes (SNPsplit). Normalized strand-specific gene coverage has been computed using bedtools, awk and R. PCA has been run on gene transcription using R.

### Genome-wide ESC-NPC transcription and RT integration

Data have been integrated and plotted using R (ggplot2) and IGV. Importantly, for both repli-seq and BrU-seq analysis, we observed that, while the cells were diploid for all autosomes, D9 clone was diploid but homozygote for the whole chr9. Thus, we kept all autosomes except chr9 for the data integration. To assess transcription changes at regions with significant advance of RT in NPC compared to ESC, global transcription of the region advancing RT is computed for each replicate and normalized per million reads per kb (RPKM). Transcription change in each region is assessed using a paired t-test. Noise threshold for transcription is set at 0.1 RPKM, as determined by plotting transcription at RT advancing regions randomly shuffled within intergenic regions. To assess RT changes at regions with change in transcription, public mouse replications domains have been used (replicationdomain.com). For each domain and each replicate, global RT and global transcription (RPKM) is computed and change of RT and transcription are assessed using paired t-test.

## Data availability

The datasets and computer code produced in this study are available in the following databases: Repli-Seq data: Gene Expression Omnibus GSE310795. Reviewer token: alglgcoovhsptox. Bru-Seq data: Gene Expression Omnibus GSE310676. Reviewer token: exgnoicabpatxgj. RNA-Seq data: Gene Expression Omnibus GSE312661.

The source data of this paper are collected in the following database record: biostudies:S-SCDT-10_1038-S44319-026-00735-2.

## Peer review information

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

## Acknowledgements

We thank B. Washburn and C. Pye for their valuable input in the design of the constructs and the guides used in these experiments. We thank C. Marchal and In siliChrom for their valuable input in the analysis of BruSeq and RepliSeq datasets. We also thank C. Vied for her valuable input with sequencing our libraries. We thank A. Chow and M. Guttman for sharing cell line F121:R26puroR-M2rtTA. Finally, we thank B. Chadwick for the modified px458 plasmid used to clone CRISPr Cas9 guide RNA sequences. This work was funded by NIH grant GM083337 to DMG and NIH Common Fund "4D Nucleome" Program grant U54DK107965 (BvS and DMG). The Oncode Institute is partly supported by KWF Dutch Cancer Society.

## Author contributions

**Athanasios E Vouzas**: Conceptualization; Data curation; Formal analysis; Validation; Investigation; Visualization; Methodology; Writing—original draft; Writing—review and editing. **Takayo Sasaki**: Data curation; Supervision; Project administration; Writing—review and editing. **Juan Carlos Rivera-Mulia**: Data curation. **Jesse L Turner**: Data curation. **Amber N Brown**: Data curation. **Karen E Alexander**: Data curation. **Laura Brueckner**: Data curation. **Bas van Steensel**: Data curation; Supervision; Writing—review and editing. **David M Gilbert**: Conceptualization; Resources; Supervision; Funding acquisition; Validation; Investigation; Visualization; Writing—original draft; Project administration; Writing—review and editing.

Source data underlying figure panels in this paper may have individual authorship assigned. Where available, figure panel/source data authorship is listed in the following database record: biostudies:S-SCDT-10_1038-S44319-026-00735-2.

## Disclosure and competing interests statement

The authors declare no competing interests.

# Expanded View Figures

**Figure EV1.  Promoter-specific effects of reporter gene insertion on RT.**

(**A**) Genome Browser tracks of genome-parsed Bru-Seq biological replicates for the cell lines with the Promoter-HTK insertions (vector sequences not included), indicating that when transcription is observed to read through the vector gene (most prominently with the CAG promoter, but barely above noise with the mPGK promoter), those reads originate exclusively from the 129 allele containing the ectopic gene insertion. (**B**) Genome Browser tracks of Bru-Seq biological replicates for the cell lines with the Promoter-HTK insertions, showing only the inserted allele for each biological replicate (this data cannot be parsed because the insert resides only on the 129 allele). Top tracks display the read-through expression of the Ptn gene, while the zoomed-in, bottom tracks display the expression of the HTK gene. (**C**) Bar graph comparing HTK and Ptn read-through expression in the Promoter-HTK cell lines. The error bars represent standard error. (**D**) Bar graph of the levels of read through Ptn expression in relation to changes in the RT of the Ptn domain. The error bars for the Ptn expression bar graphs represent standard error and the error bars for the RepliSeq data show the range of the difference among the 50 kb bins between the two replicate experiments.

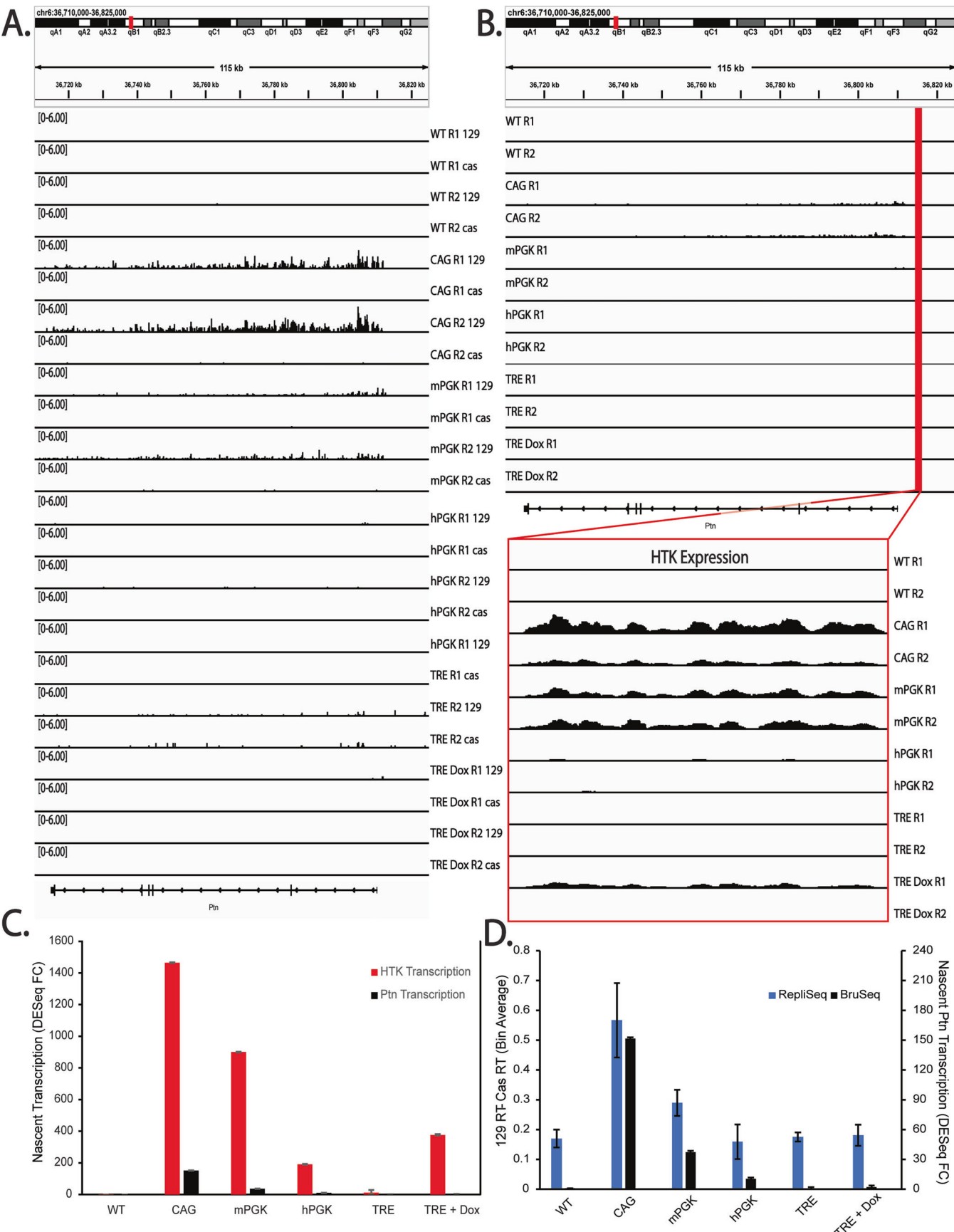

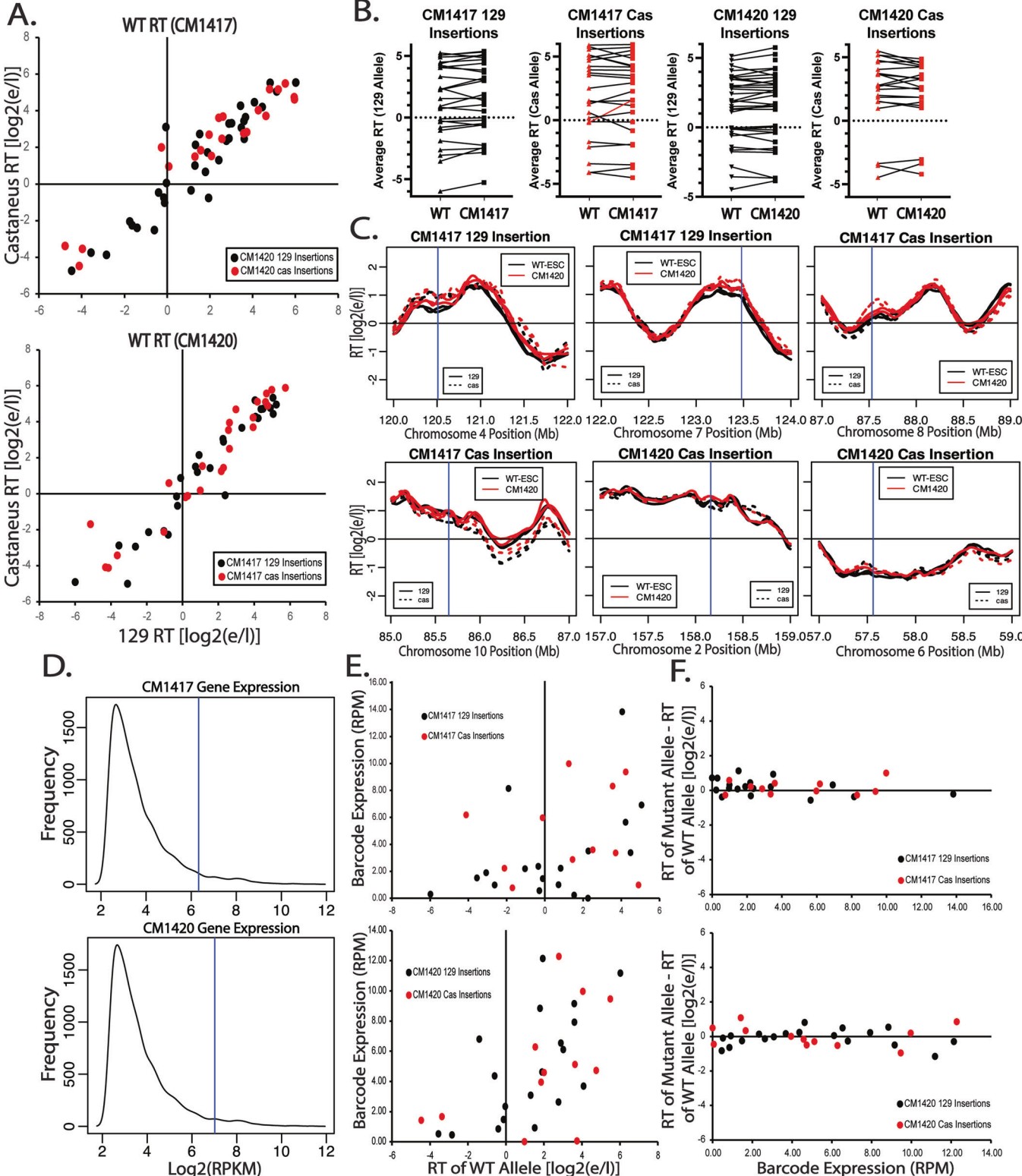

◀ **Figure EV2. Detailed analysis of PiggyBac vector insertions.**

(A) Scatter plots of the RT (129 vs Cas) of the WT parental alleles of all CM1417 (left) and CM1420 (right) ectopic insertion sites. Black and red dots represent loci where the eventual insertion was made in the 129 and Cas allele, respectively. (B) RT of the CM1417 129 insertions (left), CM1417 Cas insertions (middle-left), CM1420 129 Insertions (middle-right) and CM1420 Cas Insertions (right) before (WT, triangles) and after (CM1417 or CM1420, squares) the PB insertion. There were no statistically significant RT advances, determined as described in Methods, with the exception of the site inserted at Chromosome 8, 87.5 Mb, of the Cas allele of CM1417. The locus with the statistically significant RT change is marked with a red line in the CM1417 plot. (C) RT plots of six insertion sites where the largest advances in RT of the insertion allele were observed, including the single statistically significant RT change at the Cas Chromosome 8 allele (top right). (D) Distribution of all expressed genes in CM1417 (top) and CM1420 (bottom) clones. The x-axis represents the log2(RPKM) expression level of each gene and the y-axis represents the number of genes that are expressed at each level. RPKM was calculated from the average read count for each gene across two biological replicates. Genes with extremely low or no expression (RPKM < 5) were excluded from the plot. Average GFP expression, calculated by dividing the GFP RPKM of each clone by the number of insertions in that clone, is marked with vertical blue lines. (E) Position effect of each insertion locus on the expression level of each insertion. To differentiate between expression levels at each insert, only reads containing the 16-nucleotide barcode were extracted from the unprocessed FastQ and were used to calculate the RPM for each barcode. (F) Effect of transcription of individual insertions on the RT of the insertion site. The Y-axis range represents the dynamic range of RT in each clone.

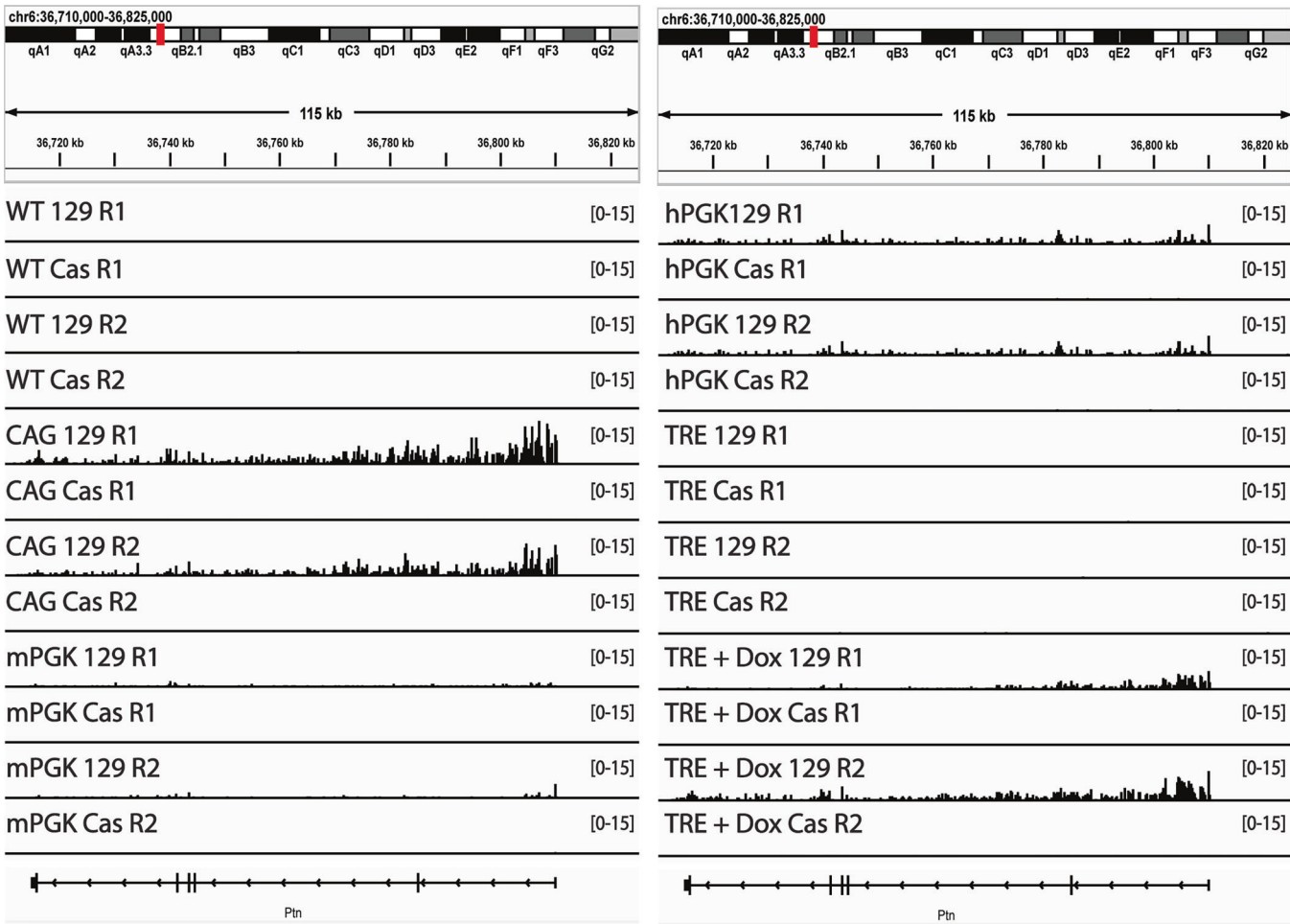

**Figure EV3. Transcription of a long transcript enhances the ability of promoters to advance RT.**

Genome Browser track of parsed Bru-Seq biological replicates for the cell lines in which the inserted promoters drive Ptn transcription.

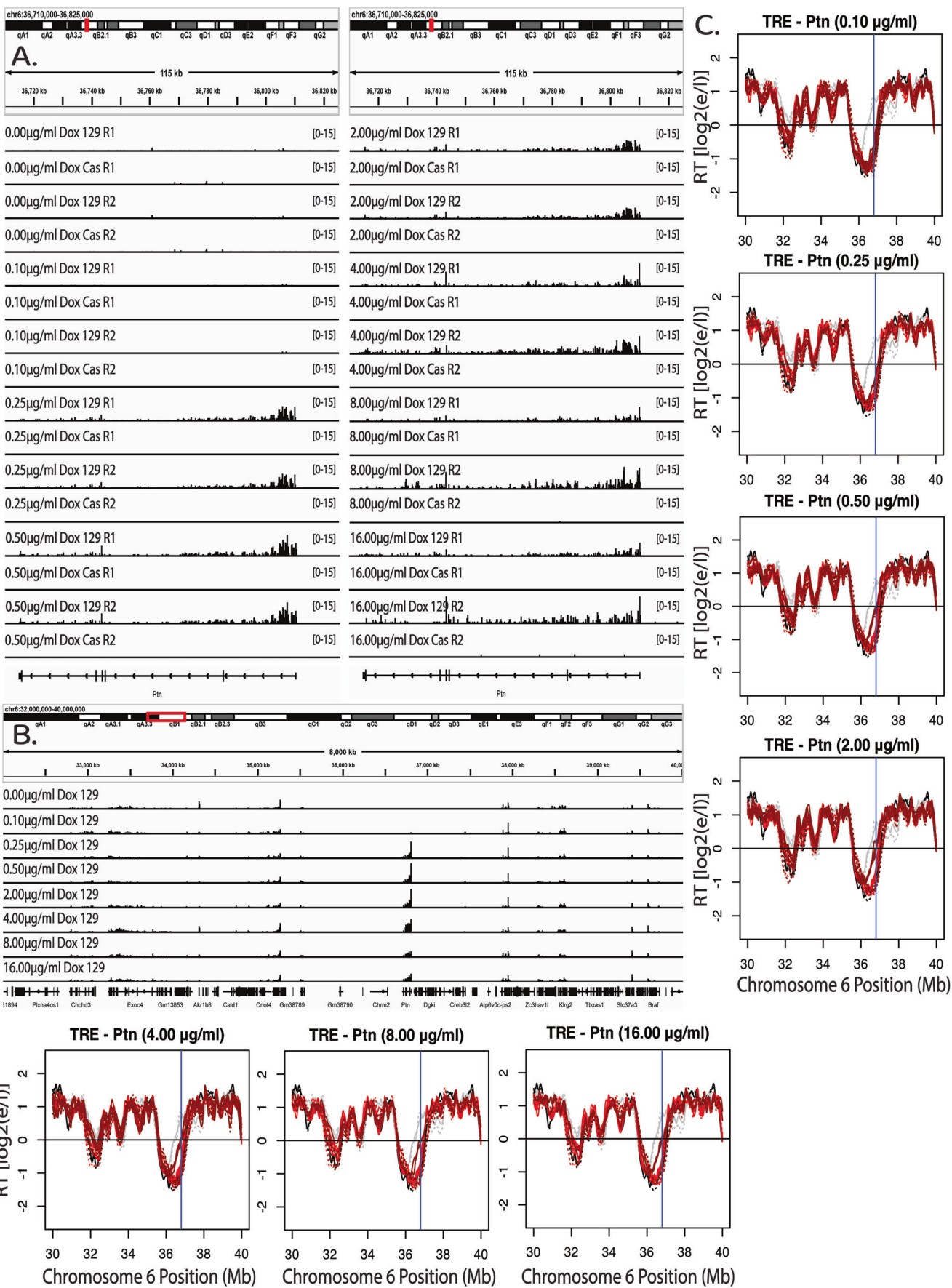

◀ **Figure EV4. Dosage-dependent advance of RT in response to transcriptional induction of the Ptn gene.**

(A) Genome Browser track of parsed Bru-Seq biological replicates displaying Ptn expression at different concentrations of Dox in the TRE-Ptn cell lines. (B) Ten Mb Zoom out of the same Genome Browser tracks in (A), demonstrating that the Ptn gene is the only gene whose transcription is affected by Dox induction for 5 Mb upstream or downstream. (C) Ten Mb Zoom out of RT across the same regions as in (B), demonstrating that the Ptn domain is the only domain whose RT is affected by Dox induction for 5 Mb upstream or downstream.

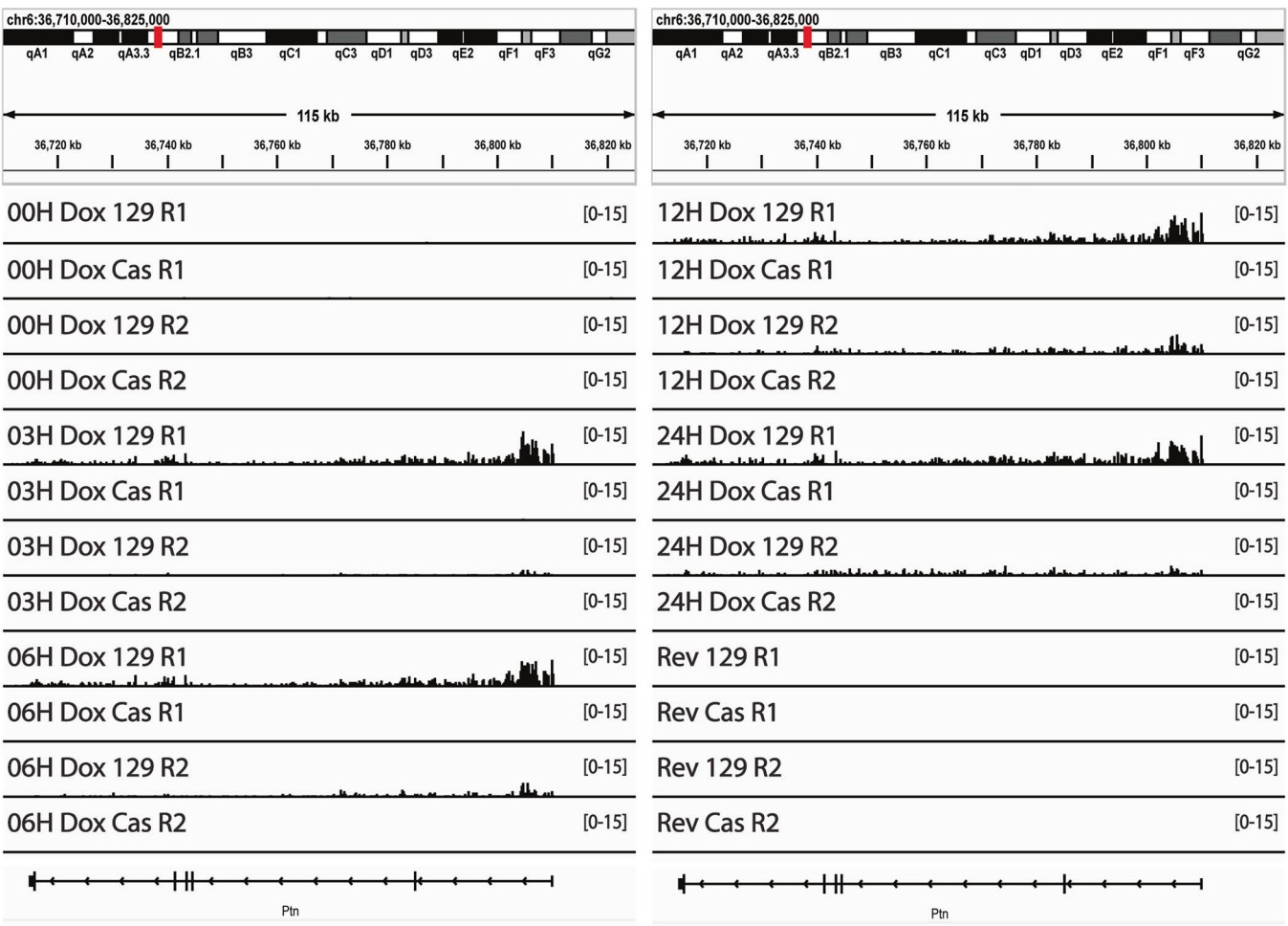

**Figure EV5.** Genome Browser track of parsed Bru-Seq biological replicates displaying Ptn expression at different time points after the addition of Dox in the TRE-Ptn cell lines.

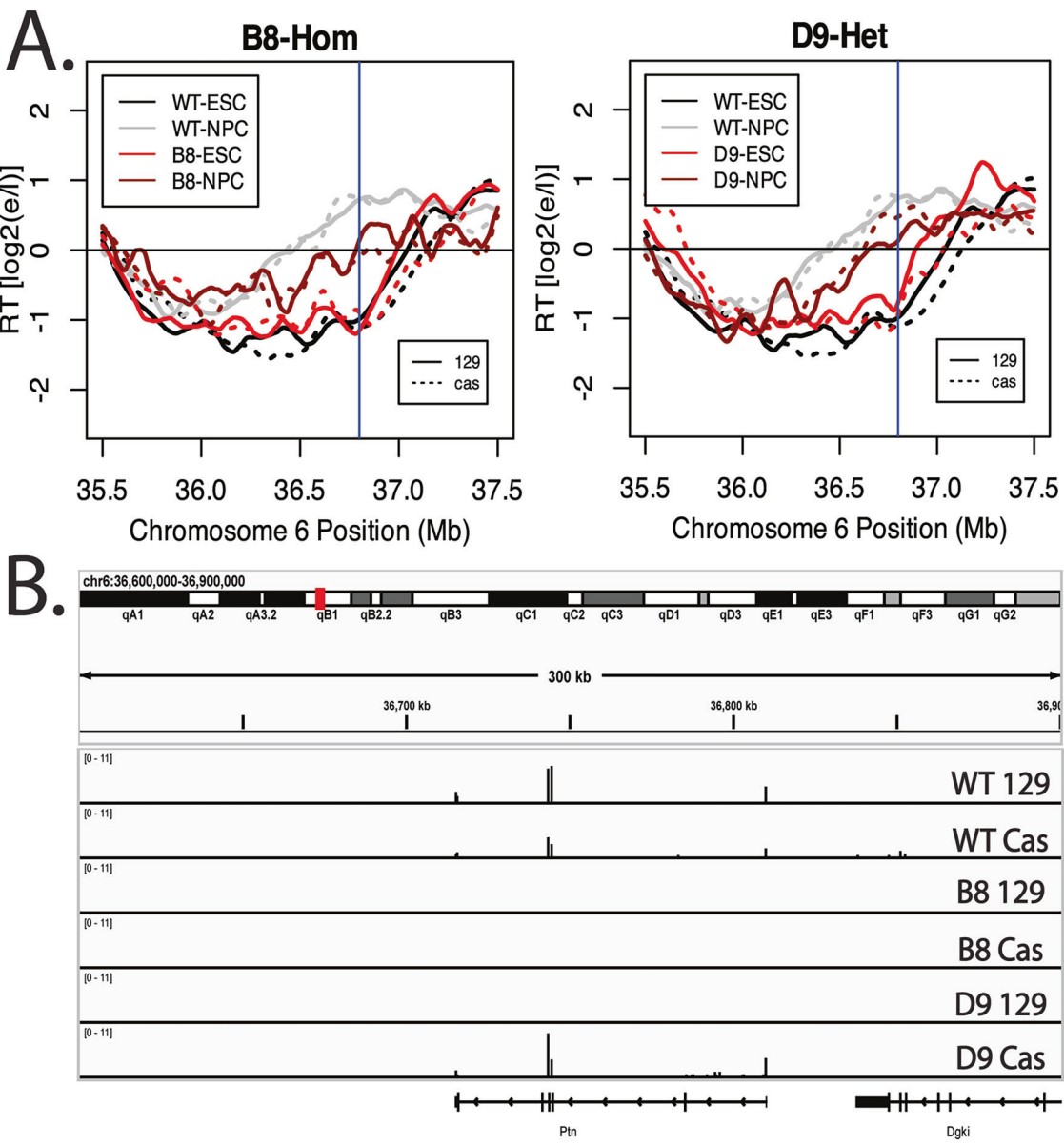

**Figure EV6. Deletion of the Ptn promoter and elimination of detectable Ptn trancripts does not significantly affect the RT advance of the Ptn domain during NPC differentiation.**

(A) RT plots of clones B8 and D9, in which the Ptn promoter deletion is homozygous and Heterozygous, respectively. Dark gray and light gray indicate the RT profiles of WT ESCs and NPCs, respectively. Light red and dark red indicate the RT profiles of ESC and NPC deletion clones, respectively. Solid and dashed lines indicate the RT of the musculus and castaneous alleles, respectively. Vertical line indicates the site of the deletion. (B) Browser track of total rRNA-depleted RNA-seq for WT NPCs and the two deletion clones. RNA-seq was performed in lieu of Bru-Seq because cell death during the differentiation process makes it difficult to collect enough cells to perform Bru-Seq.

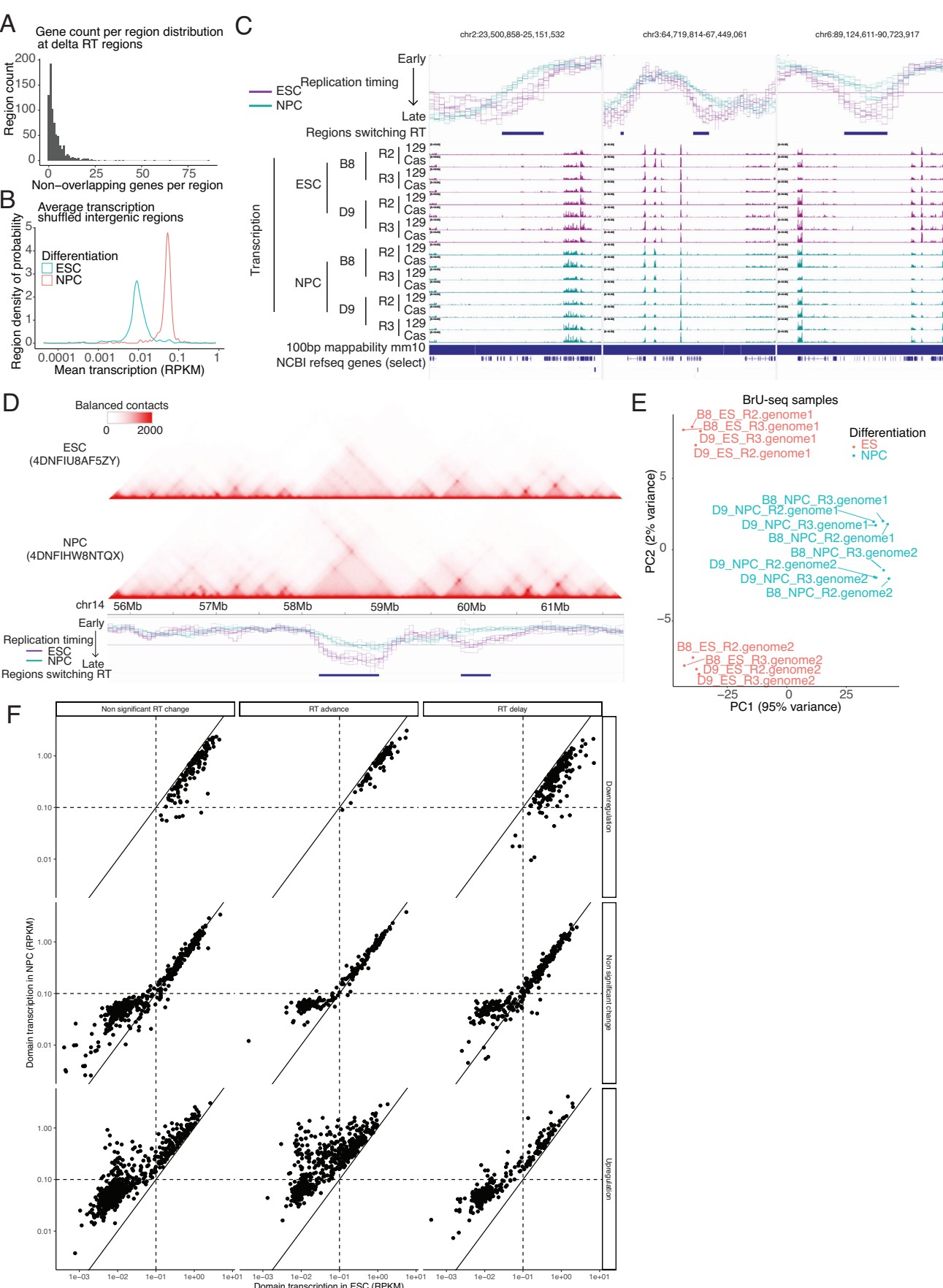

**Figure EV7. Transcription is not necessary and not always sufficient for an RT advance during the mESC to mNPC transition.**

(A) Histogram presenting the gene count distribution at regions with significant RT changes. More than half of switching regions harbor multiple annotated genes whose transcriptional elongation was summed for subsequent analyses. (B) Determination of the transcriptional noise threshold (shown in Fig. 7B,D) by shuffling regions with significant RT advance on the genome, constrained to the intergenic regions only (see Methods). Threshold is set at 0.1 RPKM. (C) RT and transcription tracks for three regions with a significant advance in RT with no significant transcription detected during the mESC to mNPC transition. Tracks represent, from top to bottom: Genomic coordinates, RT (log2 ratio Early/Late), regions with significant RT change, transcription coverage (rpm, with a maximum scale set to 10 rpm) in mESC and mNPC datasets, mm10 mappability for 100 bp reads, NCBI RefSeq genes. Data from mESC in purple and data from mNPC are in teal. (D) 3D chromatin contacts in ESC (first track) and NPC (second track) at the locus presented in Fig. 7C. RT profile and regions with significant RT changes are plotted below, identical to Fig. 7C, for comparison. (E) PCA plot of the BrU-seq gene count demonstrates clear separation of differentiated and undifferentiated cells. (F) Dotplot of replication domains. Each dot represents one domain. X-axis represents the average transcription coverage in the region in ESC (read per kb per million reads, RPKM). Y-axis represents the average transcription coverage in the region in NPC (read per kb per million reads, RPKM).

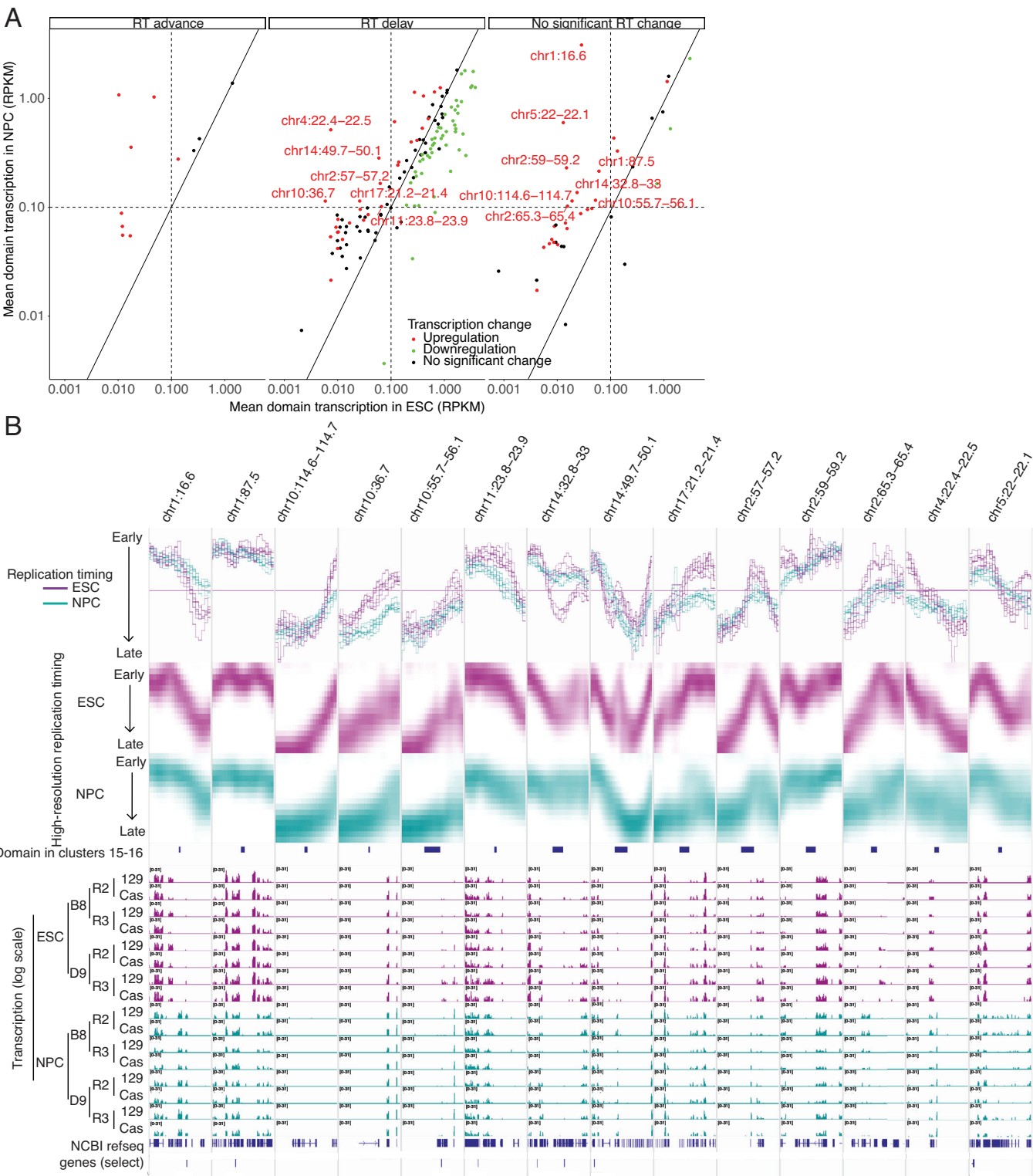

**Figure EV8.** **Revisiting a prior study identifying genomic regions that induce transcription with no RT advance.**

(**A**) Dotplot of previously identified 200 kb genomic regions containing induced genes that did not advance RT (Hiratani et al, 2010), here re-investigated using Repli-seq and BrU-seq data generated in this manuscript. The X-axis represents the average transcription coverage in the domain in mESCs (read per kb per million reads, RPKM), The Y-axis represents the average transcription coverage in the domain in mNPCs (read per kb per million reads, RPKM). Dot color represents transcription changes of the whole 200 kb window with significant up-regulation (red), significant down-regulation (green), or no significant change (black). Data are separated into three plots by replication timing change within the whole domain (advance, delay or not significant change). (**B**) RT and transcription tracks for the loci around the domains highlighted in (**A**). Tracks represent, from top to bottom: RT (log2 ratio Early/Late), high resolution Repli-seq from WT mESC and for mNPC (GSE137764), original domain localization, transcription coverage (log scale) in mESC and mNPC datasets, NCBI RefSeq genes. Data from mESC are colored in purple and data from mNPC are colored in teal.

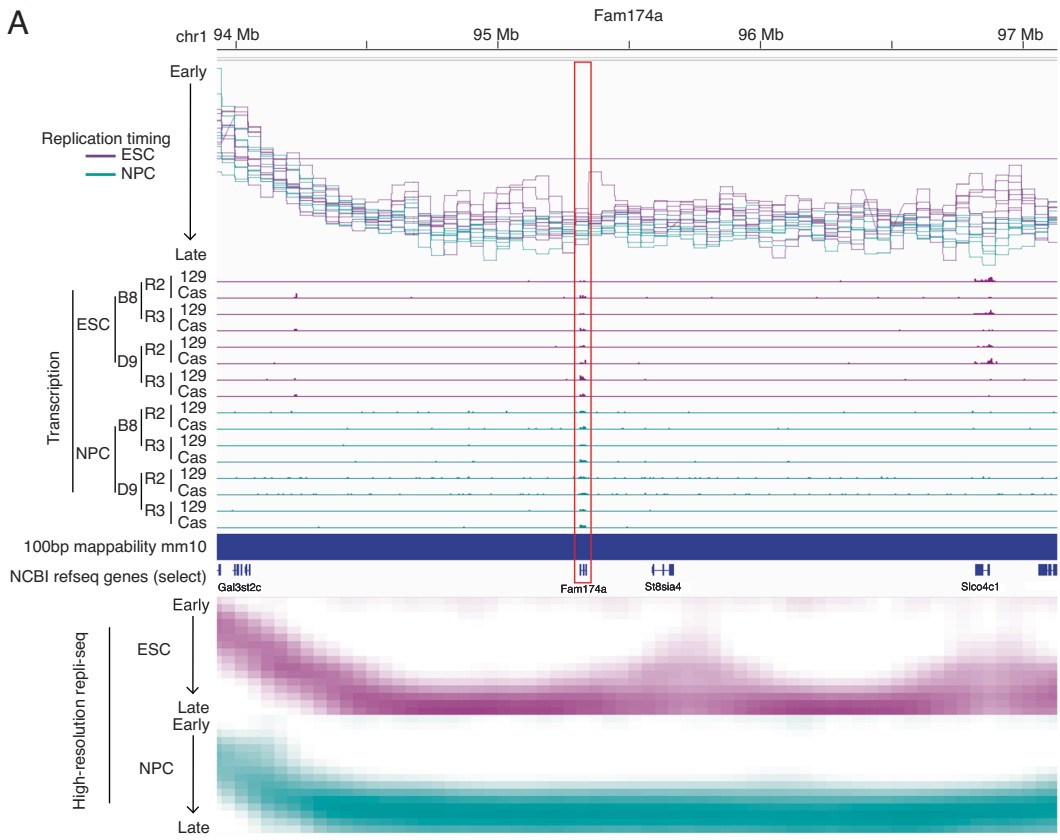

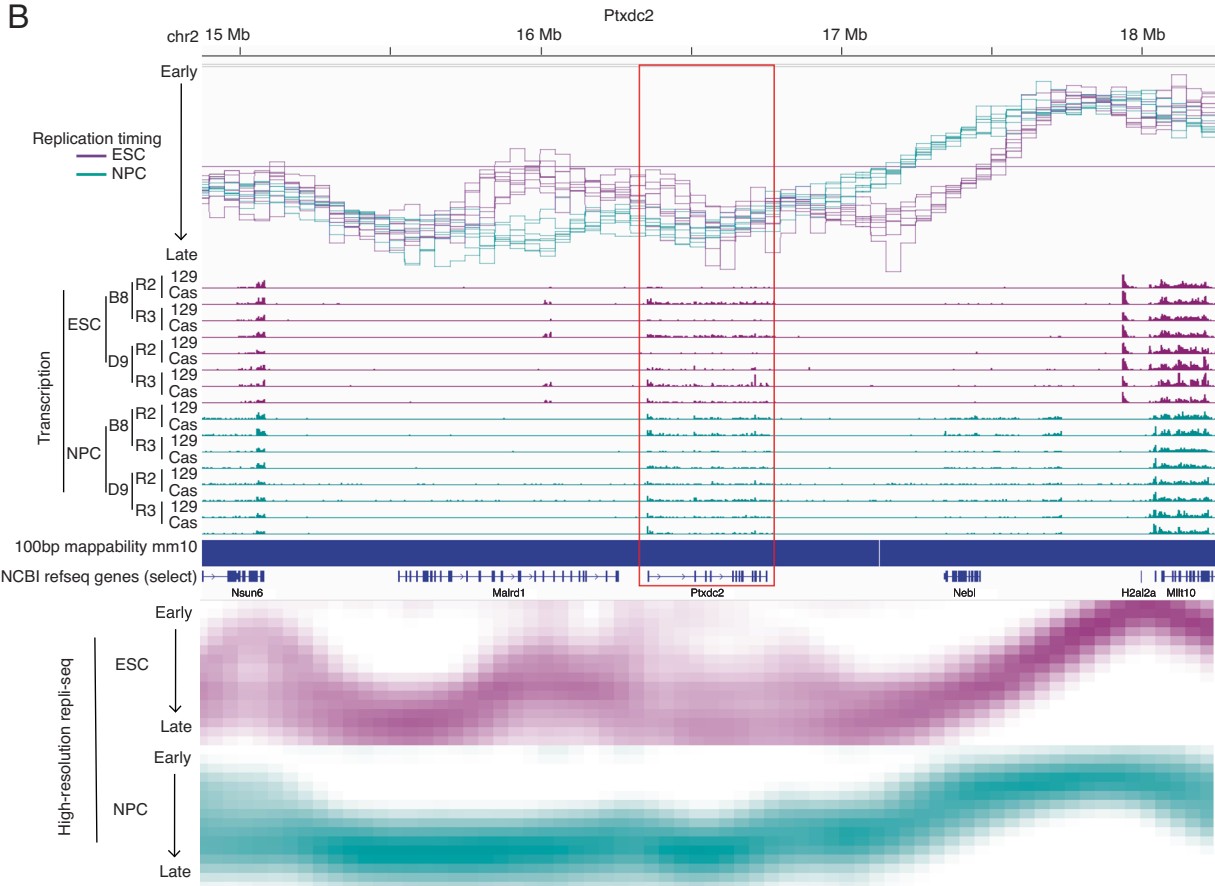

◄ **Figure EV9. Early replication is not necessary for transcription.**

(A, B) RT and transcription tracks for the loci around the late-replicating and transcribed genes Fam174a (**A**) and Ptxdc2 (**B**). Tracks represent, from top to bottom: Genomic coordinates, RT (log2 ratio Early/Late), transcription coverage (rpm, with a maximum scale set to 10 rpm) in ESC and NPC, mm10 mappability for 100 bp reads, NCBI RefSeq genes. Data from ESC are colored in purple and data from NPC are colored in teal, high resolution Repli-seq from WT mESC and for mNPC (GSE137764).

