## [Peer Review File · EMBO Reports]

Transcription elongation can be sufficient, but is not necessary, to advance replication timing

Athanasios Vouzas, Takayo Sasaki, Juan Carlos Rivera-Mulia, Jesse Turner, Amber Brown, Karen Alexander, Laura Brueckner, Bas van Steensel, and David Gilbert

Corresponding author(s): David Gilbert (gilbert@sdbri.org)

Review Timeline:

Submission Date:	6th May 25
Editorial Decision:	12th May 25
Revision Received:	8th Dec 25
Editorial Decision:	7th Jan 26
Revision Received:	26th Jan 26
Accepted:	10th Feb 26

Editor: Esther Schnapp

**Transaction Report: This manuscript was transferred to
EMBO reports following peer review at Review Commons.**

Review
COMMONS

Review #1

1. Evidence, reproducibility and clarity:

Evidence, reproducibility and clarity (Required)

The study investigates the relationship between replication timing (RT) and transcription. While there is evidence that transcription can influence RT, the underlying mechanisms remain unclear. To address this, the authors examined a single genomic locus that undergoes transcriptional activation during differentiation. They engineered the Pln locus by inserting promoters of varying strengths to modulate transcription levels and assessed the impact on replication timing using Repli-seq.

Key Findings:

- Figure 1C and 1D: The data show that higher transcription levels correlate with an advanced RT, suggesting that transcriptional activity influences replication timing.
- Figure 2: To determine whether transcription alone is sufficient to alter RT, the authors inserted an hPGK reporter at different genomic locations. However, given the findings in Figure 1, which suggest that this is not the primary mechanism,
- Figure 3: The authors removed the marker to examine whether the observed effects were due to the promoter-driven Pln locus, which has significantly larger than the marker.
- Figure 4: The study explores the effect of increased doxycycline (Dox) treatment at the TRE (tetracycline response element), further supporting the role of transcription in RT modulation.
- Figure 5: The findings demonstrate that Dox-induced RT advancement occurs rapidly, is reversible, and correlates with transcription levels, reinforcing the hypothesis that transcription plays a direct role in influencing replication timing.
- Figure 6. Shows that during differentiation transcription of Pln is not required for RT advancement.

Overall, the study presents a compelling link between transcription and replication timing, though some experimental choices warrant further clarification. I have no major comments.

Minor Comments:

Overall, the results are convincing, and the study appears to be well-conducted. In Figure 2, the authors use the hPGK promoter. However, it is unclear why they did not use

the constructs from the previous experiments. Given that the hPGK promoter did not advance RT in Figure 1, the results in Figure 2 may not be entirely unexpected.

Additionally, the study does not formally exclude the possibility that Pln protein expression itself influences RT. In Figure 1, readthrough transcription at the Pln locus could potentially drive protein expression. It would be useful to know whether the authors address this point in the discussion.

Regarding the mechanism, if transcription across longer genomic regions contributes to RT changes, transcription-induced could DNA supercoiling play a role. For instance, could negative supercoiling generated by active transcription influence replication timing?

It remains puzzling why Pln transcription does not contribute to replication timing during differentiation. Is there any evidence of chromatin opening during this process? For example, are ATAC-seq profiles available that could provide insights into chromatin accessibility changes during differentiation?

2. Significance:

Significance (Required)

This is a compelling basic single-locus study that systematically compares replication timing (RT) and transcription dynamics while measuring several key parameters of transcription.

My relevant expertise lies in transcriptional regulation and understanding how noncoding transcription influences local chromatin and gene expression.

3. How much time do you estimate the authors will need to complete the suggested revisions:

Estimated time to Complete Revisions (Required)

(Decision Recommendation)

Less than 1 month

4. Review Commons values the work of reviewers and encourages them to get credit for their work. Select 'Yes' below to register your reviewing activity at Web of Science Reviewer Recognition Service (formerly Publons); note that the content of your review will not be visible on Web of Science.

No

Review #2

1. Evidence, reproducibility and clarity:

Evidence, reproducibility and clarity (Required)

In the manuscript entitled: "Transcription can be sufficient, but is not necessary, to advance replication timing", the authors use as they state a "reductionist approach" to address a long-standing question in the replication field on what level the process of transcription within a replication domain can alter the underlying replication timing of this domain. The authors use an elegant hybrid mouse embryonic stem cell line to discriminate the two allelic copies and focus on a specific replication domain harboring the neuronal Ptn gene that is only expressed upon differentiation. The authors first introduce four different promoters in the locus upstream of Ptn gene that drive expression of small transgenes. Only the promoters with highest transcriptional induction could advance RT. If the promoters are placed in such a way that they drive expression of the 96kb Ptn gene, then also some the weaker promoters can drive RT advancement, suggesting that it is a combination of transcriptional strength and size of the transcribed domain important for RT changes. Using a DOX-inducible promoter, the authors show that this happens very fast (3-6h after transcription induction) and is reversible as removal of DOX leads to slower RT again. Finally, deleting the promoter of Ptn gene and driving cells into differentiation still advances RT, allowing the authors to conclude that "transcription can be sufficient but not necessary to advance replication timing."

****Major comments:****

Overall, this is a well designed study that includes all necessary controls to support the author's conclusions. I think it is a very interesting system that the authors developed. The weakness of the manuscript is that there is no mechanistic explanation how such RT changes are achieved on a molecular basis. But I'm confident that the system could be indeed used to further dissect the mechanistic basis for the transcription dependence of RT advancements.

Therefore, I support publication of this manuscript if a few comments below can be addressed.

1. Figure 4 shows a titration of different DOX concentrations and provides clear evidence that the degree of RT advancement tracks well with the level of transcription. As the doses of DOX are quite high in this experiment, have the authors checked on a global scale to what extent transcription might be deregulated in neighbouring genes or genome-wide?

2. One general aspect is that the whole study is only focused on the one single Ptn replication domain. Could the authors extend this rather narrow view a bit and also show RT data in the neighbouring domains. This would be particularly important for the DOX titration experiment that has the potential to induce transcriptional deregulation (see comment above).

3. Figure 5 shows that the full capacity to advance RT upon DOX induction of the Ptn gene is achieved after 3h to 6h of DOX induction, so substantially less than a full cell cycle in mESCs (12h). This result suggests that origin licensing/MCM loading cannot be the critical mechanism to drive the RT change because only a small fraction of the cells has undergone M/G1-phase where origins are starting to get loaded. As a large fraction of mESCs (60-70%) are S-phase cells in an asynchronous population, the mechanism is likely taking place directly in S-phase. Could the authors try to synchronize cells in G1/S using double-thymidine block, then induce DOX for 3h before allowing cells to reenter S-phase and then check replication timing of the domain? This can be compared to an alternative experiment where transcription is only induced for 3h upon release into S-phase. This could provide more mechanistic insights as to whether transcription is sufficient to drive RT changes in G1 versus S-phase cells.

****Minor comments:****

- Figure 1B and Figure 6A. Quality of the genome browser snapshots could be improved and certain cryptic labelling such as "only Basic displayed by default" could be removed

- The genome browser tracks appear a bit small across the figures and could be visually improved.

- In figure 1E we see an advancement in RT in Ptn gene caused by nearby enhanced Hyg-TK gene expression induced by mPGK promoter. However, in figure 3D we see mPGK promoter has reduced ability to advance RT of Ptn gene. It would be nice to address this discrepancy in the results.

2. Significance:

Significance (Required)

In my point of view, this is an important study that unifies a large amount of literature into a conceptual framework that will be interesting to a broad audience working on the intertwined fields of gene regulation, transcription and DNA replication, as well as cell fate switching and development.

3. How much time do you estimate the authors will need to complete the suggested revisions:

Estimated time to Complete Revisions (Required)

(Decision Recommendation)

Between 1 and 3 months

4. Review Commons values the work of reviewers and encourages them to get credit for their work. Select 'Yes' below to register your reviewing activity at Web of Science Reviewer Recognition Service (formerly Publons); note that the content of your review will not be visible on Web of Science.

Yes

Review #3

1. Evidence, reproducibility and clarity:

Evidence, reproducibility and clarity (Required)

In their manuscript, "Transcription can be sufficient, but is not necessary, to advance replication timing," Vouzas et al. take a systematic and reductionist approach to investigate a late-replicating domain on chromosome VI. Here, they examine the effect of transcribing a single gene locus, Pleiotrophin, on replication timing. When inserting or manipulating promoters or transcript lengths using CRISPR-Cas9, replication timing was altered in mESCs as judged by a combination of Repli-Seq, Bru-Seq, and RNA-Seq. Importantly, they found that transcription can be sufficient to advance replication timing depending on the length and strength of the expression of an ectopically transcribed gene. Taken together, the manuscript presents a compelling argument that transcription can advance replication timing but is not necessary for it.

****Major comments****

- A schematic or conceptual model summarising the major findings of transcription-dependent and independent mechanisms of RT advancement should be included in the discussion to add to the conceptual framework
- Vouzas et al. spend a substantial part of the manuscript to delve into the requirements to advance RT and even use a Doxycycline-based titration for temporal advancement of RT. Yet, all conclusions come from the use of hybrid-genome mouse embryonic stem cells (mESCs). Therefore, it remains speculative if and whether findings can be generalized to other cell types or organisms. The authors could include another organism/ cell type to strengthen the relevance of their findings to a broader audience, particular as they identified promoters that drive ectopic gene expression without affecting RT. Showcasing this in other model organisms would be of great interest.
- OPTIONAL: as with the previous point, the authors went to great depth and length to show how ectopic manipulations affect RT changes on a single locus using genome-wide methods. In addition, the manuscript would benefit from the inclusion of other loci, particularly as transcription of the Ptn locus wasn't needed during differentiation to advance RT at all.
- The same point of Ptn not needing to be transcribed to advance RT of the respective domain, albeit being a very interesting observation, disturbs the flow of the manuscript, as the whole case was built around transcription and this particular locus-containing domain. Maybe one can adapt the storytelling to fit better within the overall framework.

****Minor comments****

- While citations are thorough, some references (e.g., "need to add Wang, Klein, Mol. Cell 2021") are incomplete.
- The text corresponding to Figure 1C could use more explanation for readers not familiar with the depiction of Repli-Seq data.
- Figure 1C needs labelling of the x-axes.
- Statistical analyses should be used consistently throughout the manuscript and explained in more detail, i.e. significance levels, tests, instead of "Significant differences....calculated using x".

****Referees cross-commenting****

Comment on Reviewer#1's review, comment mentioning ATAC-Seq: Another way to look at this could be to investigate for origin usage changes (BrdU-Seq or GLOE-Seq) of

chromosome 6 during differentiation.

Comment on Reviewer#2's review, major comment 3: I do agree with their statement that origin loading cannot be the driver of RT change, as MCM2-7 double hexamer loading is strictly uncoupled from origin firing. Hence, any mechanism responsible for RT advance must happen at the G1/S phase transition or during S-phase, most likely due to the regulated activity of DDK/CDK or the limitation and preferred recruitment of firing factors to early origins. This could be tested through overexpression of said factors.

2. Significance:

Significance (Required)

General: This manuscript presents a compelling study investigating the relationship between transcription and replication timing (RT) using a reductionist approach. The authors systematically manipulated transcriptional activity at the Ptn locus to dissect the elements of transcription that influence RT. The study's strengths lie in its rigorous experimental design, clear results, and the reconciliation of seemingly contradictory findings in the existing literature. However, some aspects could be improved, particularly in exploring the mechanistic details of transcription-independent RT regulation at the investigated domain, the generalisability of the findings to other cells/organisms, and enhancing the presentation of certain data (explanation of e.g. Figure 1c, dense figure arrangement, lack of a summary figure illustrating key findings (e.g., correlation between transcription rate, readthrough effects, and RT advancement)).

Advance: The manuscript directly addresses and reconciles contradictory findings in the literature regarding the effect of ectopic transcription on RT. Previous studies have reported varying effects, with some showing that transcription advances RT (Brueckner et al., 2020; Therizols et al., 2014), while others have shown no effect or only partial effects depending on the insertion site (Gilbert & Cohen, 1990; Goren et al., 2008). The current study conceptually advances the field by systematically testing different promoters and transcript lengths at a single locus (mechanistic insight), demonstrating that the length and strength of transcription, as well as promoter context, influence RT. This presents a unifying concept on how RT can be influenced. The authors also present a tunable system (technical advance) that allows rapid and reversible alterations of RT, which will certainly be useful for future studies and the field.

Audience: The primary audience will be specialised researchers in the fields of replication timing, epigenetics, and gene regulation. This study may be of interest beyond

the specific field of replication timing, such as cancer biology, developmental biology, particularly if a more broader applicability of its tools and concepts can be shown.

Expertise: origin licensing, origin activation, MCM2-7, yeast and human cell lines

3. How much time do you estimate the authors will need to complete the suggested revisions:

Estimated time to Complete Revisions (Required)

(Decision Recommendation)

Between 1 and 3 months

Yes

Revision Plan

Manuscript number: RC202502929

Corresponding author(s): David M. Gilbert

[The “revision plan” should delineate the revisions that authors intend to carry out in response to the points raised by the referees. It also provides the authors with the opportunity to explain their view of the paper and of the referee reports.]

The document is important for the editors of affiliate journals when they make a first decision on the transferred manuscript. It will also be useful to readers of the reprint and help them to obtain a balanced view of the paper.

*If you wish to submit a full revision, please use our "Full Revision" template. **It is important to use the appropriate template to clearly inform the editors of your intentions.**]*

1. General Statements [optional]

This section is optional. Insert here any general statements you wish to make about the goal of the study or about the reviews.

We are pleased with the quality of the reviews and the review process. We anticipate this work to be recognized as an authoritative reconciliation of decades of literature describing seemingly contradictory observations that can now be consolidated into factual statements regarding the relationship of transcription to replication timing.

In the following Revision plan, for part 2 we include a complete point by point response, with our responses in **red font** and the changes already made in the attached provisional revised manuscript in **bold red font**. We then provide the new experiments and analyses that are near completion and expected to be completed “soon” (within a few weeks) in **bold blue font**. For part 3 and 4, we break out those parts done and not planning to be done.

Revision Plan

2. Description of the planned revisions

Insert here a point-by-point reply that explains what revisions, additional experimentations and analyses are planned to address the points raised by the referees.

Reviewer #1 (Evidence, reproducibility and clarity (Required)):

The study investigates the relationship between replication timing (RT) and transcription. While there is evidence that transcription can influence RT, the underlying mechanisms remain unclear. To address this, the authors examined a single genomic locus that undergoes transcriptional activation during differentiation. They engineered the PIn locus by inserting promoters of varying strengths to modulate transcription levels and assessed the impact on replication timing using Repli-seq.

Key Findings:

- Figure 1C and 1D: The data show that higher transcription levels correlate with an advanced RT, suggesting that transcriptional activity influences replication timing.
- Figure 2: To determine whether transcription alone is sufficient to alter RT, the authors inserted an hPGK reporter at different genomic locations. However, given the findings in Figure 1, which suggest that this is not the primary mechanism,
- Figure 3: The authors removed the marker to examine whether the observed effects were due to the promoter-driven PIn locus, which has significantly larger than the marker.
- Figure 4: The study explores the effect of increased doxycycline (Dox) treatment at the TRE (tetracycline response element), further supporting the role of transcription in RT modulation.
- Figure 5: The findings demonstrate that Dox-induced RT advancement occurs rapidly, is reversible, and correlates with transcription levels, reinforcing the hypothesis that transcription plays a direct role in influencing replication timing.
- Figure 6. Shows that during differentiation transcription of PIn is not required for RT advancement.

Overall, the study presents a compelling link between transcription and replication timing, though some experimental choices warrant further clarification. **I have no major comments.**

Minor Comments:

Overall, the results are convincing, and the study appears to be well-conducted. In Figure 2, the authors use the hPGK promoter. However, it is unclear why they did not use the constructs from the previous experiments. Given that the hPGK promoter did not advance RT in Figure 1, the results in Figure 2 may not be entirely unexpected.

We took advantage of previously published cell lines using a PiggyBac Vector designed to pepper the reporter gene at random sites throughout the genome; the point of the experiment was to acquire supporting evidence for the hypothesis that any vector with its selectable marker driven by the hPGK promoter will not advance RT no matter where it is inserted. Since there are reports concluding that transcription per se is sufficient to advance RT, it was important to

Revision Plan

confirm that there was nothing unique about the particular vector or locus into which we inserted our panel of vectors.

ACTION DONE: We have now added the following sentence to the results describing this experiment: “By analyzing RT in these lines, we could evaluate the effect of a different hPGK vector on RT when integrated at many different chromosomal sites. “

Additionally, the study does not formally exclude the possibility that PIn protein expression itself influences RT. In Figure 1, readthrough transcription at the PIn locus could potentially drive protein expression. It would be useful to know whether the authors address this point in the discussion.

NOT DONE FOR NEED OF CLARIFICATION: It is unclear why a secreted neural growth factor would have a direct effect on replication timing in embryonic stem cells and, in particular, only *in cis* (remember there is a control allele that is unaffected). We would be happy to address this in the Discussion if we understood the reviewers’ hypothesis. **We cannot respond to this comment without understanding the hypothesis being tested as we do not know how a secreted protein could affect the RT of one allele without affecting the other.**

Regarding the mechanism, if transcription across longer genomic regions contributes to RT changes, transcription-induced could DNA supercoiling play a role. For instance, could negative supercoiling generated by active transcription influence replication timing?

Yes, many mechanisms are possible.

ACTION DONE: We have added the following sentence to the discussion, referencing a seminal paper on that topic by Nick Gilbert: “ For example, long transcripts could remodel a large segment of chromatin, possibly by creating domains of DNA supercoiling (Naughton et. al., 2013).”

It remains puzzling why PIn transcription does not contribute to replication timing during differentiation. Is there any evidence of chromatin opening during this process? For example, are ATAC-seq profiles available that could provide insights into chromatin accessibility changes during differentiation?

We thank the reviewer for asking this as we should have mentioned something very important here. Lack of necessity for transcription implies that independent mechanisms are functioning to elicit the RT switch. In other work (Turner et. al., bioRxiv, provisionally accepted to EMBO J.), we have shown that specific cis elements (ERCs) can function to maintain early replication in the absence of transcription.

ACTION DONE: We now explicitly state in the Discussion: “This is not surprising, given that ERCs can maintain early RT in the absence of transcription (Turner, bioRxiv).”

ACTION TO BE DONE SOON: We will provide a new Figure 6D showing ATAC-seq changes upon differentiation of mESCs to mNPCs and their location relative to the promoter/enhance deletion. As you can see, there is an ATAC-seq site that appears during differentiation, upstream of the deletion. We will hypothesize in the revised

Revision Plan

manuscript that these are the elements that drive the RT switch and that future studies need to investigate that hypothesis. We have also added the following sentences to the discussion after the sentence above, stating: “In fact, new sites of open chromatin, consistent with ERCEs appear outside of the deleted Ptn transcription control elements after differentiation (soon to be revised Figure 6D). The necessity and sufficiency of these sites to advance RT independent of transcription will be important to follow up.”

Figure for referee with unpublished data has been removed upon request by the authors.

We also provide the figure below for the reviewer only. These are preliminary data that are part of a separate project in the lab so they are not ready for publication, but are directly relevant to the reviewer’s question. This figure shows preliminary evidence for a region upstream of the Ptn promoter/enhancer deletion described in Figure 6 that, when deleted, DOES have an effect on the RT switch during differentiation. This deletion overlaps an ATAC-seq site we will show in the figure above.

Figure for referee with unpublished data has been removed upon request by the authors.

Reviewer #1 (Significance (Required)):

This is a compelling basic single-locus study that systematically compares replication timing (RT) and transcription dynamics while measuring several key parameters of transcription.

My relevant expertise lies in transcriptional regulation and understanding how noncoding transcription influences local chromatin and gene expression.

Reviewer #2 (Evidence, reproducibility and clarity (Required)):

In the manuscript entitled: Transcription can be sufficient, but is not necessary, to advance replication timing”, the authors use as they state a “reductionist approach” to address a long-standing question in the replication field on what level the process of transcription within a replication domain can alter the underlying replication timing of this domain. The authors use an elegant hybrid mouse embryonic stem cell line to discriminate the two allelic copies and focus

Revision Plan

on a specific replication domain harboring the neuronal Ptn gene that is only expressed upon differentiation. The authors first introduce four different promoters in the locus upstream of Ptn gene that drive expression of small transgenes. Only the promoters with highest transcriptional induction could advance RT. If the promoters are placed in such a way that they drive expression of the 96kb Ptn gene, then also some the weaker promoters can drive RT advancement, suggesting that it is a combination of transcriptional strength and size of the transcribed domain important for RT changes. Using a DOX-inducible promoter, the authors show that this happens very fast (3-6h after transcription induction) and is reversible as removal of DOX leads to slower RT again. Finally, deleting the promoter of Ptn gene and driving cells into differentiation still advances RT, allowing the authors to conclude that "transcription can be sufficient but not necessary to advance replication timing."

Major comments:

Overall, this is a well designed study that includes all necessary controls to support the author's conclusions. I think it is a very interesting system that the authors developed. The weakness of the manuscript is that there is no mechanistic explanation how such RT changes are achieved on a molecular basis. But I'm confident that the system could be indeed used to further dissect the mechanistic basis for the transcription dependence of RT advancements.

Therefore, I support publication of this manuscript if a few comments below can be addressed.

1) Figure 4 shows a titration of different DOX concentrations and provides clear evidence that the degree of RT advancement tracks well with the level of transcription. As the doses of DOX are quite high in this experiment, have the authors checked on a global scale to what extent transcription might be deregulated in neighbouring genes or genome-wide?

The DOX concentration that we use for all experiments other than the titration is 2 µg/ml, which is quite standard. The high concentrations (up to 16µg/ml) are only used in the titration experiments shown in Figure 4 to demonstrate that we have reached a plateau. In fact, we stated in Materials and Methods that high doses of Dox led to cell toxicity. Looking at the transcription datasets, there are no significant changes in transcription below 8µg/ml, a few dozen significant changes at 8 and more such changes at 16µg/ml of DOX. The tables of genome wide RT and transcription are provided in the manuscript for anyone wishing to investigate the effects of Dox on cellular physiology but at the concentration used in all other experiments (2µg/ml) there are no effects on transcription.

ACTION DONE: We have now modified the statement in the Materials and Methods to read: " Mild toxicity and changes in genome-wide transcription were observed at 8µg/ml and more so at 16µg/ml".

2) One general aspect is that the whole study is only focused on the one single Ptn replication domain. Could the authors extend this rather narrow view a bit and also show RT data in the neighbouring domains. This would be particularly important for the DOX titration experiment that has the potential to induce transcriptional deregulation (see comment above).

Revision Plan

ACTION DONE: We have now added to revised Supplemental Figure 4 a zoom out of 10 Mb surrounding the Ptn gene showing no detectable effects on RT at any of the titration concentrations.

ACTION TO BE DONE SOON: To address the generalization of the findings (length and strength matter), we have repeated the ESC to NPC differentiation and performed both Repli-seq and BrU-seq to evaluate RT changes relative to total genomic nascent transcriptional changes. The sequencing reads for this experiment are in our analyst's hands so we expect this to be ready within a few weeks. We will provide a new Figure 7 comparing genome-wide changes in RT vs. transcription to determine the significance of length and strength of transcription induction to RT advances and the necessity of transcriptional induction for RT advances. We and other laboratories have performed many integrative analyses of RNA-microarray/RNA-seq data vs. RT changes, but not total genomic nascent transcription and not with a focus on the effect of length and strength of transcription. For example, outcomes that would be consistent with our reductionist findings at the Ptn locus would be if we find domains that are advanced for RT with no induction of transcription (transcription not necessary) and little to no regions showing significant induction of transcription without RT advances.

3) Figure 5 shows that the full capacity to advance RT upon DOX induction of the Ptn gene is achieved after 3h to 6h of DOX induction, so substantially less than a full cell cycle in mESCs (12h). This result suggests that origin licensing/MCM loading cannot be the critical mechanism to drive the RT change because only a small fraction of the cells has undergone M/G1-phase where origins are starting to get loaded. As a large fraction of mESCs (60-70%) are S-phase cells in an asynchronous population, the mechanism is likely taking place directly in S-phase. Could the authors try to synchronize cells in G1/S using double-thymidine block, then induce DOX for 3h before allowing cells to reenter S-phase and then check replication timing of the domain? This can be compared to an alternative experiment where transcription is only induced for 3h upon release into S-phase. This could provide more mechanistic insights as to whether transcription is sufficient to drive RT changes in G1 versus S-phase cells.

We agree that the timing of induction is such that it is very likely that alterations in RT can occur during S phase. The reviewer proposes a reasonable experiment that could be done, but it would require a long delay of this publication to develop and validate those synchronization protocols and we do not have personnel at this time to carry out the experiment. This would be a great initiating experiment for someone to pursue the mechanisms by which transcription can advance RT.

ACTION DONE: We have added the following sentence to the Discussion section on mechanisms: The rapid nature of the RT change after induction of transcription suggests that RT changes can occur after the functional loading of inactive MCM helicases onto chromatin in telophase/early G1 (Dimitrova, JCB, 1999; Okuno, EMBO J. 2001; Dimitrova, J. Cell Sci, 2002), and possibly after S phase begins.

Minor comments:

- Figure 1B and Figure 6A. Quality of the genome browser snapshots could be improved and certain cryptic labelling such as "only Basic displayed by default" could be removed

Revision Plan

ACTION DONE: We have modified these figures.

- The genome browser tracks appear a bit small across the figures and could be visually improved.

ACTION DONE: We have modified the genome browser tracks to improve their presentation

- In figure 1E we see an advancement in RT in Ptn gene caused by nearby enhanced Hyg-TK gene expression induced by mPGK promoter. However, in figure 3D we see mPGK promoter has reduced ability to advance RT of Ptn gene. It would be nice to address this discrepancy in the results.

The reviewer's point is well taken. We are not sure of the answer. You can see that the transcription is very low in both cases, while the RT shift is greater in one replicate vs. the other.

ACTION DONE: We have, rather unsatisfactorily, added the following sentence to the results section describing Figure 3. "We do not know why the mPGK promoter was so poor at driving transcription in this context."

Reviewer #2 (Significance (Required)):

In my point of view, this is an important study that unifies a large amount of literature into a conceptual framework that will be interesting to a broad audience working on the intertwined fields of gene regulation, transcription and DNA replication, as well as cell fate switching and development.

Reviewer #3 (Evidence, reproducibility and clarity (Required)):

In their manuscript, "Transcription can be sufficient, but is not necessary, to advance replication timing," Vouzas et al. take a systematic and reductionist approach to investigate a late-replicating domain on chromosome VI. Here, they examine the effect of transcribing a single gene locus, Pleiotrophin, on replication timing. When inserting or manipulating promoters or transcript lengths using CRISPR-Cas9, replication timing was altered in mESCs as judged by a combination of Repli-Seq, Bru-Seq, and RNA-Seq. Importantly, they found that transcription can be sufficient to advance replication timing depending on the length and strength of the expression of an ectopically transcribed gene. Taken together, the manuscript presents a compelling argument that transcription can advance replication timing but is not necessary for it.

Major comments

- A schematic or conceptual model summarising the major findings of transcription-dependent and independent mechanisms of RT advancement should be included in the discussion to add to the conceptual framework

Revision Plan

NOT DONE: We discussed this at length between the two senior authors and the first author and we do not feel ready to draw a summary model. We do not know what is advancing RT when transcription is induced or not induced, and we are not comfortable choosing one possible model of many. We hope that the added speculations on mechanism in the Discussion will sufficiently convey the future research that we feel needs to be done.

ACTIONS DONE: In addition to the speculation on mechanism that already was in our Discussion section, we have added: On mechanisms of rapid induction of RT change, we have added to the Discussion: “The rapid nature of the RT change after induction of transcription suggests that RT changes can occur after the functional loading of inactive MCM helicases onto chromatin in telophase/early G1 (Dimitrova, JCB, 1999; Okuno, EMBO J. 2001; Dimitrova, J. Cell Sci, 2002), and possibly after S phase begins.” And “For example, long transcripts could remodel a large segment of chromatin, possibly by creating domains of DNA supercoiling (Naughton et. al., 2013, PMID 23416946). “On mechanisms of RT advance in the absence of transcription, we have added the following to the Discussion: “This is not surprising, given that ERCEs can maintain early RT in the absence of transcription (Turner, bioRxiv). In fact, chromatin features with the properties of ERCEs do appear outside of the deleted Ptn transcription control elements after differentiation (soon to be revised Figure 6C). The necessity and sufficiency of these new chromatin features to advance RT independent of transcription will be important to follow up.”

- Vouzas et al. spend a substantial part of the manuscript to delve into the requirements to advance RT and even use a Doxycycline-based titration for temporal advancement of RT. Yet, all conclusions come from the use of hybrid-genome mouse embryonic stem cells (mESCs). Therefore, it remains speculative if and whether findings can be generalized to other cell types or organisms. The authors could include another organism/ cell type to strengthen the relevance of their findings to a broader audience, particular as they identified promoters that drive ectopic gene expression without affecting RT. Showcasing this in other model organisms would be of great interest.

NOT DONE: To set this system up in another cell type or species would take a very long time. We also do not have personnel to carry that approach.

ACTION TO BE DONE SOON: As an alternative approach that partially addresses this reviewer’s concern, we will provide a new Figure 7 with an analysis of RT changes vs. transcriptional changes when mESCs are differentiated to neural precursor cells. As described above in response to Reviewer #2’s criticism #2, we have repeated the ESC to NPC differentiation and performed both Repli-seq and BrU-seq to evaluate RT changes relative to total genomic nascent transcriptional changes. The sequencing reads for this experiment are in our analyst’s hands so we expect this to be ready within a few weeks. We will compare genome-wide changes in RT vs. transcription to determine the significance of length and strength of transcription induction to RT advances and the necessity of transcriptional induction for RT advances. We and other laboratories have performed many integrative analyses of RNA-microarray/RNA-seq data vs. RT changes, but not total genomic nascent transcription and not with a focus on the effect of length and strength of transcription. For example, outcomes that would be consistent with our

Revision Plan

reductionist findings at the Ptn locus would be if we find domains that are advanced for RT with no induction of transcription (transcription not necessary) and little to no regions showing significant induction of transcription without RT advances.

- OPTIONAL: as with the previous point, the authors went to great depth and length to show how ectopic manipulations affect RT changes on a single locus using genome-wide methods. In addition, the manuscript would benefit from the inclusion of other loci, particularly as transcription of the Ptn locus wasn't needed during differentiation to advance RT at all.

NOT DONE: This rigorous reductionist approach is laborious and to set it up at one gene at a time at additional loci would be a huge effort taking quite a long time.

ACTION TO BE DONE SOON: (same as response above) As an alternative approach that partially addresses this reviewer's concern, we will provide a new Figure 7 with an analysis of RT changes vs. transcriptional changes when mESCs are differentiated to neural precursor cells. As described above in response to Reviewer #2's criticism #2, we have repeated the ESC to NPC differentiation and performed both Repli-seq and BrU-seq to evaluate RT changes relative to total genomic nascent transcriptional changes. The sequencing reads for this experiment are in our analyst's hands so we expect this to be ready within a few weeks. We will compare genome-wide changes in RT vs. transcription to determine the significance of length and strength of transcription induction to RT advances and the necessity of transcriptional induction for RT advances. We and other laboratories have performed many integrative analyses of RNA-microarray/RNA-seq data vs. RT changes, but not total genomic nascent transcription and not with a focus on the effect of length and strength of transcription. For example, outcomes that would be consistent with our reductionist findings at the Ptn locus would be if we find domains that are advanced for RT with no induction of transcription (transcription not necessary) and little to no regions showing significant induction of transcription without RT advances.

- The same point of Ptn not needing to be transcribed to advance RT of the respective domain, albeit being a very interesting observation, disturbs the flow of the manuscript, as the whole case was built around transcription and this particular locus-containing domain. Maybe one can adapt the storytelling to fit better within the overall framework.

We would argue that demonstrating induction of Ptn, the only gene in this domain, is sufficient to induce early RT is a logical segway to asking whether, in the natural situation, induction is correlated with advance in RT. Our results show that transcription is sufficient but not necessary, which is expected if there are other mechanisms that regulate RT.

ACTION DONE: To make this transition more smooth, we have added the following sentence to the beginning of the results section describing Figure 6: “ This raises the question as to whether the natural RT advance that accompanies Ptn induction during differentiation requires Ptn transcription, or whether other mechanisms, such as ERCEs (Sima / Turner) can advance RT independent of transcription. “

ACTION TO BE DONE SOON: To finish the work flow in a way that ties length and strength and sufficiency but not necessity in to the theme of natural cellular

Revision Plan

differentiation, we will provide a new Figure 7 with an analysis of RT changes vs. transcriptional changes when mESCs are differentiated to neural precursor cells, as described above.

Minor comments

- While citations are thorough, some references (e.g., "need to add Wang, Klein, Mol. Cell 2021") are incomplete.

ACTION TO BE DONE SOON: We apologize that some references seemed to not be incorporated into the reference manager Mendely. Since we are still planning to add one more figure soon and we will need to add some references for the datasets that will be shown in future Figure 6D, after that draft is ready, we will comb the manuscript for any references that were not entered and correct them.

- The text corresponding to Figure 1C could use more explanation for readers not familiar with the depiction of Repli-Seq data.

ACTION DONE: "Repli-seq labels nascent DNA with BrdU, followed by flow cytometry to purify cells in early vs. late S phase based on their DNA content, then BrdU-substituted DNA from each of these fractions is immunoprecipitated, sequenced and expressed as a log2 ration of early to late synthesized DNA (log2E/L). BrU-seq labels total nascent RNA, which is then immunoprecipitated an expressed as reads per million per kilobase (RPMK)."

- Figure 1C needs labelling of the x-axes.

ACTION DONE: We have now labeled the X axes.

- Statistical analyses should be used consistently throughout the manuscript and explained in more detail, i.e. significance levels, tests, instead of "Significant differences....calculated using x".

We used the same analysis for all the Repliseq data and the same analysis for all the Bruseq data. We agree that we did not present this consistently in the figure legends and methods.

ACTION DONE: To correct the confusion we have clarified the statistical methods in the methods section and referred to methods in the figure legends as follows:

The methods description of statistical significance for RT now reads: "Statistical significance of RT changes for all windows in each sample, relative to WT, were calculated using RepliPrint (Ryba et al., 2011), with a p-value of 0.01 used as the cut-off for windows with statistically significant differences."

The methods description of statistical significance for transcription now reads: "Differential expression analysis, including the calculation of statistically significant

Revision Plan

differences in expression, was conducted using the R package DESeq2. In Figure 1, statistical significance was calculated relative to HTK expression in the parental cell line, which is expected to be zero, since the parental line does not have an HTK insertion. In all other Figures significance was calculated relative to Ptn expression in the parental line, which is expected to be zero, since the parental line does not express Ptn.”

The legend to Figure 1C now reads: The red shading indicates 50kb windows with statistically significant differences in RT between WT casteneus and modified 129 alleles, determined as described in Methods.

The legend to Figure 1E now reads: “The asterisks indicate a significant difference in the levels of HTK expression relative to HTK expression in the parental cell line as described in Methods. There are no asterisks for the RT data, as statistical significance was calculated for individual 50kb windows as shown in panel (C).”

Each time significance is measured in the subsequent legends, it is followed by the phrase “, determined as described in Methods” or “presented as in Figure 1C” or “presented as in Figure 1E” as appropriate.

****Referees cross-commenting****

Comment on Reviewer#1's review, comment mentioning ATAC-Seq: Another way to look at this could be to investigate for origin usage changes (BrdU-Seq or GLOE-Seq) of chromosome 6 during differentiation.

NOT DONE: Unfortunately we could not find any studies comparing origin mapping in mESCs and mNPCs.

Comment on Reviewer#2's review, major comment 3: I do agree with their statement that origin loading cannot be the driver of RT change, as MCM2-7 double hexamer loading is strictly uncoupled from origin firing. Hence, any mechanism responsible for RT advance must happen at the G1/S phase transition or during S-phase, most likely due to the regulated activity of DDK/CDK or the limitation and preferred recruitment of firing factors to early origins. This could be tested through overexpression of said factors.

NOT DONE: We agree that manipulating these factors would be a reasonable next approach to sort out mechanism. Due to limited resources and personnel, we will not be able to do this in a short period of time. We also argue that these are experiments for the next chapter of the story, likely requiring an entire PhD thesis (or multiple) to sort out.

ACTION DONE: We have added the following sentence to the Discussion section on mechanisms: The rapid nature of the RT change after induction of transcription suggests that RT changes can occur after the functional loading of inactive MCM helicases onto chromatin in telophase/early G1 (Dimitrova, JCB, 1999; Okuno, EMBO J. 2001; Dimitrova, J. Cell Sci, 2002), and possibly after S phase begins.

Reviewer #3 (Significance (Required)):

General: This manuscript presents a compelling study investigating the relationship between

Revision Plan

transcription and replication timing (RT) using a reductionist approach. The authors systematically manipulated transcriptional activity at the Ptn locus to dissect the elements of transcription that influence RT. The study's strengths lie in its rigorous experimental design, clear results, and the reconciliation of seemingly contradictory findings in the existing literature. However, some aspects could be improved, particularly in exploring the mechanistic details of transcription-independent RT regulation at the investigated domain, the generalisability of the findings to other cells/organisms, and enhancing the presentation of certain data (explanation of e.g. Figure 1c, dense figure arrangement, lack of a summary figure illustrating key findings (e.g., correlation between transcription rate, readthrough effects, and RT advancement)).

Advance: The manuscript directly addresses and reconciles contradictory findings in the literature regarding the effect of ectopic transcription on RT. Previous studies have reported varying effects, with some showing that transcription advances RT (Brueckner et al., 2020; Therizols et al., 2014), while others have shown no effect or only partial effects depending on the insertion site (Gilbert & Cohen, 1990; Goren et al., 2008). The current study conceptually advances the field by systematically testing different promoters and transcript lengths at a single locus (mechanistic insight), demonstrating that the length and strength of transcription, as well as promoter context, influence RT. This presents a unifying concept on how RT can be influenced. The authors also present a tunable system (technical advance) that allows rapid and reversible alterations of RT, which will certainly be useful for future studies and the field.

Audience: The primary audience will be specialised researchers in the fields of replication timing, epigenetics, and gene regulation. This study may be of interest beyond the specific field of replication timing, such as cancer biology, developmental biology, particularly if a more broader applicability of its tools and concepts can be shown.

Expertise: origin licensing, origin activation, MCM2-7, yeast and human cell lines

Revision Plan

3. Description of the revisions that have already been incorporated in the transferred manuscript

Please insert a point-by-point reply describing the revisions that were already carried out and included in the transferred manuscript. If no revisions have been carried out yet, please leave this section empty.

Reviewer 1 had only minor comments

Minor comment 1: Overall, the results are convincing, and the study appears to be well-conducted. In Figure 2, the authors use the hPGK promoter. However, it is unclear why they did not use the constructs from the previous experiments. Given that the hPGK promoter did not advance RT in Figure 1, the results in Figure 2 may not be entirely unexpected.

We took advantage of previously published cell lines using a PiggyBac Vector designed to pepper the reporter gene at random sites throughout the genome; the point of the experiment was to acquire supporting evidence for the hypothesis that any vector with its selectable marker driven by the hPGK promoter will not advance RT no matter where it is inserted. Since there are reports concluding that transcription per se is sufficient to advance RT, it was important to confirm that there was nothing unique about the particular vector or locus into which we inserted our panel of vectors.

ACTION DONE: We have now added the following sentence to the results describing this experiment: “By analyzing RT in these lines, we could evaluate the effect of a different hPGK vector on RT when integrated at many different chromosomal sites. “

Minor comment 3: Regarding the mechanism, if transcription across longer genomic regions contributes to RT changes, transcription-induced could DNA supercoiling play a role. For instance, could negative supercoiling generated by active transcription influence replication timing?

Yes, many mechanisms are possible.

ACTION DONE: We have added the following sentence to the discussion, referencing a seminal paper on that topic by Nick Gilbert: “ For example, long transcripts could remodel a large segment of chromatin, possibly by creating domains of DNA supercoiling (Naughton et. al., 2013).”

Minor comment 4: It remains puzzling why Pln transcription does not contribute to replication timing during differentiation. Is there any evidence of chromatin opening during this process? For example, are ATAC-seq profiles available that could provide insights into chromatin accessibility changes during differentiation?

Revision Plan

We thank the reviewer for asking this as we should have mentioned something very important here. Lack of necessity for transcription implies that independent mechanisms are functioning to elicit the RT switch. In other work (Turner et. al., bioRxiv, provisionally accepted to EMBO J.), we have shown that specific cis elements (ERCs) can function to maintain early replication in the absence of transcription.

ACTION DONE: We now explicitly state in the Discussion: “This is not surprising, given that ERCs can maintain early RT in the absence of transcription (Turner, bioRxiv).”

ACTION TO BE DONE SOON: We will provide a new Figure 6D showing ATAC-seq changes upon differentiation of mESCs to mNPCs and their location relative to the promoter/enhancer deletion. As you can see, there is an ATAC-seq site that appears during differentiation, upstream of the deletion. We will hypothesize in the revised manuscript that these are the elements that drive the RT switch and that future studies need to investigate that hypothesis. We have also added the following sentences to the discussion after the sentence above, stating: “In fact, new sites of open chromatin, consistent with ERCs appear outside of the deleted Ptn transcription control elements after differentiation (soon to be revised Figure 6D). The necessity and sufficiency of these sites to advance RT independent of transcription will be important to follow up.”

Figure for referee with unpublished data has been removed upon request by the authors.

We also provide the figure below for the reviewer only. These are preliminary data that are part of a separate project in the lab so they are not ready for publication, but are directly relevant to the reviewer’s question. This figure shows preliminary evidence for a region upstream of the Ptn promoter/enhancer deletion described in Figure 6 that, when deleted, DOES have an effect on the RT switch during differentiation. This deletion overlaps an ATAC-seq site we will show in the figure above.

Figure for referee with unpublished data has been removed upon request by the authors.

Revision Plan

Reviewer 2

Major comment 1: Figure 4 shows a titration of different DOX concentrations and provides clear evidence that the degree of RT advancement tracks well with the level of transcription. As the doses of DOX are quite high in this experiment, have the authors checked on a global scale to what extent transcription might be deregulated in neighbouring genes or genome-wide?

The DOX concentration that we use for all experiments other than the titration is 2 µg/ml, which is quite standard. The high concentrations (up to 16µg/ml) are only used in the titration experiments shown in Figure 4 to demonstrate that we have reached a plateau. In fact, we stated in Materials and Methods that high doses of Dox led to cell toxicity. Looking at the transcription datasets, there are no significant changes in transcription below 8µg/ml, a few dozen significant changes at 8 and more such changes at 16µg/ml of DOX. The tables of genome wide RT and transcription are provided in the manuscript for anyone wishing to investigate the effects of Dox on cellular physiology but at the concentration used in all other experiments (2µg/ml) there are no effects on transcription.

ACTION DONE: We have now modified the statement in the Materials and Methods to read: “ Mild toxicity and changes in genome-wide transcription were observed at 8µg/ml and more so at 16µg/ml”.

Major comment 2: One general aspect is that the whole study is only focused on the one single Ptn replication domain. Could the authors extend this rather narrow view a bit and also show RT data in the neighbouring domains. This would be particularly important for the DOX titration experiment that has the potential to induce transcriptional deregulation (see comment above).

ACTION DONE: We have now added to revised Supplemental Figure 4 a zoom out of 10 Mb surrounding the Ptn gene showing no detectable effects on RT at any of the titration concentrations.

ACTION TO BE DONE SOON: To address the generalization of the findings (length and strength matter), we have repeated the ESC to NPC differentiation and performed both Repli-seq and BrU-seq to evaluate RT changes relative to total genomic nascent transcriptional changes. The sequencing reads for this experiment are in our analyst's hands so we expect this to be ready within a few weeks. We will provide a new Figure 7 comparing genome-wide changes in RT vs. transcription to determine the significance of length and strength of transcription induction to RT advances and the necessity of transcriptional induction for RT advances. We and other laboratories have performed many integrative analyses of RNA-microarray/RNA-seq data vs. RT changes, but not total genomic nascent transcription and not with a focus on the effect of length and strength of transcription. For example, outcomes that would be consistent with our reductionist findings at the Ptn locus would be if we find domains that are advanced for RT with no induction of transcription (transcription not necessary) and little to no regions showing significant induction of transcription without RT advances.

Revision Plan

Major comment 3: Figure 5 shows that the full capacity to advance RT upon DOX induction of the Ptn gene is achieved after 3h to 6h of DOX induction, so substantially less than a full cell cycle in mESCs (12h). This result suggests that origin licensing/MCM loading cannot be the critical mechanism to drive the RT change because only a small fraction of the cells has undergone M/G1-phase where origins are starting to get loaded. As a large fraction of mESCs (60-70%) are S-phase cells in an asynchronous population, the mechanism is likely taking place directly in S-phase. Could the authors try to synchronize cells in G1/S using double-thymidine block, then induce DOX for 3h before allowing cells to reenter S-phase and then check replication timing of the domain? This can be compared to an alternative experiment where transcription is only induced for 3h upon release into S-phase. This could provide more mechanistic insights as to whether transcription is sufficient to drive RT changes in G1 versus S-phase cells.

We agree that the timing of induction is such that it is very likely that alterations in RT can occur during S phase. The reviewer proposes a reasonable experiment that could be done, but it would require a long delay of this publication to develop and validate those synchronization protocols and we do not have personnel at this time to carry out the experiment. This would be a great initiating experiment for someone to pursue the mechanisms by which transcription can advance RT.

ACTION DONE: We have added the following sentence to the Discussion section on mechanisms: The rapid nature of the RT change after induction of transcription suggests that RT changes can occur after the functional loading of inactive MCM helicases onto chromatin in telophase/early G1 (Dimitrova, JCB, 1999; Okuno, EMBO J. 2001; Dimitrova, J. Cell Sci, 2002), and possibly after S phase begins.

Minor comment 1: Figure 1B and Figure 6A. Quality of the genome browser snapshots could be improved and certain cryptic labelling such as "only Basic displayed by default" could be removed

ACTION DONE: We have modified these figures.

Minor comment 2: The genome browser tracks appear a bit small across the figures and could be visually improved.

ACTION DONE: We have modified the genome browser tracks to improve their presentation

Minor comment 3: In figure 1E we see an advancement in RT in Ptn gene caused by nearby enhanced Hyg-TK gene expression induced by mPGK promoter. However, in figure 3D we see mPGK promoter has reduced ability to advance RT of Ptn gene. It would be nice to address this discrepancy in the results.

Revision Plan

The reviewer's point is well taken. We are not sure of the answer. You can see that the transcription is very low in both cases, while the RT shift is greater in one replicate vs. the other.

ACTION DONE: We have, rather unsatisfactorily, added the following sentence to the results section describing Figure 3. "We do not know why the mPGK promoter was so poor at driving transcription in this context."

Reviewer 3

Major comment 1: A schematic or conceptual model summarising the major findings of transcription-dependent and independent mechanisms of RT advancement should be included in the discussion to add to the conceptual framework

NOT DONE: We discussed this at length between the two senior authors and the first author and we do not feel ready to draw a summary model. We do not know what is advancing RT when transcription is induced or not induced, and we are not comfortable choosing one possible model of many. We hope that the added speculations on mechanism in the Discussion will sufficiently convey the future research that we feel needs to be done.

ACTIONS DONE: In addition to the speculation on mechanism that already was in our Discussion section, we have added: On mechanisms of rapid induction of RT change, we have added to the Discussion: "The rapid nature of the RT change after induction of transcription suggests that RT changes can occur after the functional loading of inactive MCM helicases onto chromatin in telophase/early G1 (Dimitrova, JCB, 1999; Okuno, EMBO J. 2001; Dimitrova, J. Cell Sci, 2002), and possibly after S phase begins." And "For example, long transcripts could remodel a large segment of chromatin, possibly by creating domains of DNA supercoiling (Naughton et. al., 2013, PMID 23416946). "On mechanisms of RT advance in the absence of transcription, we have added the following to the Discussion: "This is not surprising, given that ERCEs can maintain early RT in the absence of transcription (Turner, bioRxiv). In fact, chromatin features with the properties of ERCEs do appear outside of the deleted Ptn transcription control elements after differentiation (soon to be revised Figure 6C). The necessity and sufficiency of these new chromatin features to advance RT independent of transcription will be important to follow up."

Major comment 2: Vouzas et al. spend a substantial part of the manuscript to delve into the requirements to advance RT and even use a Doxycycline-based titration for temporal advancement of RT. Yet, all conclusions come from the use of hybrid-genome mouse embryonic stem cells (mESCs). Therefore, it remains speculative if and whether findings can be generalized to other cell types or organisms. The authors could include another organism/ cell type to strengthen the relevance of their findings to a broader audience, particular as they identified promoters that drive ectopic gene expression without affecting RT. Showcasing this in other model organisms would be of great interest.

NOT DONE: To set this system up in another cell type or species would take a very long time. We also do not have personnel to carry that approach.

Revision Plan

ACTION TO BE DONE SOON: As an alternative approach that partially addresses this reviewer's concern, we will provide a new Figure 7 with an analysis of RT changes vs. transcriptional changes when mESCs are differentiated to neural precursor cells. As described above in response to Reviewer #2's criticism #2, we have repeated the ESC to NPC differentiation and performed both Repli-seq and BrU-seq to evaluate RT changes relative to total genomic nascent transcriptional changes. The sequencing reads for this experiment are in our analyst's hands so we expect this to be ready within a few weeks. We will compare genome-wide changes in RT vs. transcription to determine the significance of length and strength of transcription induction to RT advances and the necessity of transcriptional induction for RT advances. We and other laboratories have performed many integrative analyses of RNA-microarray/RNA-seq data vs. RT changes, but not total genomic nascent transcription and not with a focus on the effect of length and strength of transcription. For example, outcomes that would be consistent with our reductionist findings at the Ptn locus would be if we find domains that are advanced for RT with no induction of transcription (transcription not necessary) and little to no regions showing significant induction of transcription without RT advances.

Major comment 3: OPTIONAL: as with the previous point, the authors went to great depth and length to show how ectopic manipulations affect RT changes on a single locus using genome-wide methods. In addition, the manuscript would benefit from the inclusion of other loci, particularly as transcription of the Ptn locus wasn't needed during differentiation to advance RT at all.

NOT DONE: This rigorous reductionist approach is laborious and to set it up at one gene at a time at additional loci would be a huge effort taking quite a long time.

ACTION TO BE DONE SOON: (same as response above) As an alternative approach that partially addresses this reviewer's concern, we will provide a new Figure 7 with an analysis of RT changes vs. transcriptional changes when mESCs are differentiated to neural precursor cells. As described above in response to Reviewer #2's criticism #2, we have repeated the ESC to NPC differentiation and performed both Repli-seq and BrU-seq to evaluate RT changes relative to total genomic nascent transcriptional changes. The sequencing reads for this experiment are in our analyst's hands so we expect this to be ready within a few weeks. We will compare genome-wide changes in RT vs. transcription to determine the significance of length and strength of transcription induction to RT advances and the necessity of transcriptional induction for RT advances. We and other laboratories have performed many integrative analyses of RNA-microarray/RNA-seq data vs. RT changes, but not total genomic nascent transcription and not with a focus on the effect of length and strength of transcription. For example, outcomes that would be consistent with our reductionist findings at the Ptn locus would be if we find domains that are advanced for RT with no induction of transcription (transcription not necessary) and little to no regions showing significant induction of transcription without RT advances.

Major comment 4: The same point of Ptn not needing to be transcribed to advance RT of the respective domain, albeit being a very interesting observation, disturbs the flow of the manuscript, as the whole case was built around transcription and this particular locus-containing domain. Maybe one can adapt the storytelling to fit better within the overall framework.

Revision Plan

We would argue that demonstrating induction of Ptn, the only gene in this domain, is sufficient to induce early RT is a logical segway to asking whether, in the natural situation, induction is correlated with advance in RT. Our results show that transcription is sufficient but not necessary, which is expected if there are other mechanisms that regulate RT.

ACTION DONE: To make this transition more smooth, we have added the following sentence to the beginning of the results section describing Figure 6: “ This raises the question as to whether the natural RT advance that accompanies Ptn induction during differentiation requires Ptn transcription, or whether other mechanisms, such as ERCES (Sima / Turner) can advance RT independent of transcription. “

ACTION TO BE DONE SOON: To finish the work flow in a way that ties length and strength and sufficiency but not necessity in to the theme of natural cellular differentiation, we will provide a new Figure 7 with an analysis of RT changes vs. transcriptional changes when mESCs are differentiated to neural precursor cells, as described above.

Minor comment 1: While citations are thorough, some references (e.g., "need to add Wang, Klein, Mol. Cell 2021") are incomplete.

ACTION TO BE DONE SOON: We apologize that some references seemed to not be incorporated into the reference manager Mendely. Since we are still planning to add one more figure soon and we will need to add some references for the datasets that will be shown in future Figure 6D, after that draft is ready, we will comb the manuscript for any references that were not entered and correct them.

Minor comment 2: The text corresponding to Figure 1C could use more explanation for readers not familiar with the depiction of Repli-Seq data.

ACTION DONE: “Repli-seq labels nascent DNA with BrdU, followed by flow cytometry to purify cells in early vs. late S phase based on their DNA content, then BrdU-substituted DNA from each of these fractions is immunoprecipitated, sequenced and expressed as a log₂ ration of early to late synthesized DNA (log₂E/L). BrU-seq labels total nascent RNA, which is then immunoprecipitated an expressed as reads per million per kilobase (RPMK).”

Minor comment 3: Figure 1C needs labelling of the x-axes.

ACTION DONE: We have now labeled the X axes.

Minor comment 4: Statistical analyses should be used consistently throughout the manuscript and explained in more detail, i.e. significance levels, tests, instead of "Significant differences....calculated using x".

Revision Plan

We used the same analysis for all the Repliseq data and the same analysis for all the Bruseq data. We agree that we did not present this consistently in the figure legends and methods.

ACTION DONE: To correct the confusion we have clarified the statistical methods in the methods section and referred to methods in the figure legends as follows:

The methods description of statistical significance for RT now reads: “Statistical significance of RT changes for all windows in each sample, relative to WT, were calculated using RepliPrint (Ryba et al., 2011), with a p-value of 0.01 used as the cut-off for windows with statistically significant differences.”

The methods description of statistical significance for transcription now reads: “Differential expression analysis, including the calculation of statistically significant differences in expression, was conducted using the R package DESeq2. In Figure 1, statistical significance was calculated relative to HTK expression in the parental cell line, which is expected to be zero, since the parental line does not have an HTK insertion. In all other Figures significance was calculated relative to Ptn expression in the parental line, which is expected to be zero, since the parental line does not express Ptn.”

The legend to Figure 1C now reads: The red shading indicates 50kb windows with statistically significant differences in RT between WT casteneus and modified 129 alleles, determined as described in Methods.

The legend to Figure 1E now reads: “The asterisks indicate a significant difference in the levels of HTK expression relative to HTK expression in the parental cell line as described in Methods. There are no asterisks for the RT data, as statistical significance was calculated for individual 50kb windows as shown in panel (C).”

Each time significance is measured in the subsequent legends, it is followed by the phrase “, determined as described in Methods” or “presented as in Figure 1C” or “presented as in Figure 1E” as appropriate.

****Referees cross-commenting****

Comment on Reviewer#2's review, major comment 3: I do agree with their statement that origin loading cannot be the driver of RT change, as MCM2-7 double hexamer loading is strictly uncoupled from origin firing. Hence, any mechanism responsible for RT advance must happen at the G1/S phase transition or during S-phase, most likely due to the regulated activity of DDK/CDK or the limitation and preferred recruitment of firing factors to early origins. This could be tested through overexpression of said factors.

NOT DONE: We agree that manipulating these factors would be a reasonable next approach to sort out mechanism. Due to limited resources and personnel, we will not be able to do this in a short period of time. We also argue that these are experiments for the next chapter of the story, likely requiring an entire PhD thesis (or multiple) to sort out.

ACTION DONE: We have added the following sentence to the Discussion section on mechanisms: The rapid nature of the RT change after induction of transcription suggests that RT changes can occur after the functional loading of inactive MCM helicases onto chromatin in telophase/early G1 (Dimitrova, JCB, 1999; Okuno, EMBO J. 2001; Dimitrova, J. Cell Sci, 2002), and possibly after S phase begins.

Revision Plan

4. Description of analyses that authors prefer not to carry out

Please include a point-by-point response explaining why some of the requested data or additional analyses might not be necessary or cannot be provided within the scope of a revision. This can be due to time or resource limitations or in case of disagreement about the necessity of such additional data given the scope of the study. Please leave empty if not applicable.

Reviewer 1

Minor comment 2: Additionally, the study does not formally exclude the possibility that PIn protein expression itself influences RT. In Figure 1, readthrough transcription at the PIn locus could potentially drive protein expression. It would be useful to know whether the authors address this point in the discussion.

NOT DONE FOR NEED OF CLARIFICATION: It is unclear why a secreted neural growth factor would have a direct effect on replication timing in embryonic stem cells and, in particular, only in cis (remember there is a control allele that is unaffected). We would be happy to address this in the Discussion if we understood the reviewers' hypothesis. **We cannot respond to this comment without understanding the hypothesis being tested as we do not know how a secreted protein could affect the RT of one allele without affecting the other.**

Reviewer 2 – we have already or will address all of Reviewer 2s criticisms.

Reviewer 3

Major comment 3: OPTIONAL: as with the previous point, the authors went to great depth and length to show how ectopic manipulations affect RT changes on a single locus using genome-wide methods. In addition, the manuscript would benefit from the inclusion of other loci, particularly as transcription of the Ptn locus wasn't needed during differentiation to advance RT at all.

NOT DONE: This rigorous reductionist approach is laborious and to set it up at one gene at a time at additional loci would be a huge effort taking quite a long time.

ACTION TO BE DONE SOON: (same as response above) As an alternative approach that partially addresses this reviewer's concern, we will provide a new Figure 7 with an analysis of RT changes vs. transcriptional changes when mESCs are differentiated to neural precursor cells. As described above in response to Reviewer #2s criticism #2, we have repeated the ESC to NPC differentiation and performed both Repli-seq and BrU-seq to evaluate RT changes relative to total genomic nascent transcriptional changes. The sequencing reads for this experiment are in our analyst's hands so we expect this to be ready within a few weeks. We will compare genome-wide changes in RT vs. transcription to determine the significance of length and strength of transcription induction to RT advances and the necessity of transcriptional induction for RT advances. We and other laboratories have performed many integrative analyses of RNA-microarray/RNA-seq data

Revision Plan

vs. RT changes, but not total genomic nascent transcription and not with a focus on the effect of length and strength of transcription. For example, outcomes that would be consistent with our reductionist findings at the Ptn locus would be if we find domains that are advanced for RT with no induction of transcription (transcription not necessary) and little to no regions showing significant induction of transcription without RT advances.

Comment on Reviewer#1's review, comment mentioning ATAC-Seq: Another way to look at this could be to investigate for origin usage changes (BrdU-Seq or GLOE-Seq) of chromosome 6 during differentiation.

NOT DONE: Unfortunately we could not find any studies comparing origin mapping in mESCs and mNPCs.

Comment on Reviewer#2's review, major comment 3: I do agree with their statement that origin loading cannot be the driver of RT change, as MCM2-7 double hexamer loading is strictly uncoupled from origin firing. Hence, any mechanism responsible for RT advance must happen at the G1/S phase transition or during S-phase, most likely due to the regulated activity of DDK/CDK or the limitation and preferred recruitment of firing factors to early origins. This could be tested through overexpression of said factors.

NOT DONE: We agree that manipulating these factors would be a reasonable next approach to sort out mechanism. Due to limited resources and personnel, we will not be able to do this in a short period of time. We also argue that these are experiments for the next chapter of the story, likely requiring an entire PhD thesis (or multiple) to sort out.

ACTION DONE: We have added the following sentence to the Discussion section on mechanisms: The rapid nature of the RT change after induction of transcription suggests that RT changes can occur after the functional loading of inactive MCM helicases onto chromatin in telophase/early G1 (Dimitrova, JCB, 1999; Okuno, EMBO J. 2001; Dimitrova, J. Cell Sci, 2002), and possibly after S phase begins.

Dear David,

Thank you for the transfer of your manuscript to EMBO reports and for your revision plan. We think that your study is potentially a good fit for us, especially if you can add the genome-wide comparisons of length and strength of transcription induction to RT.

I asked referee 2 for feedback who overall agrees with your revision plan and only adds that "some of the suggested experiments e.g the cell synchronization and release experiment to check the effect of transcription activation in G1 versus S-phase could be done in a reasonable timeframe and would likely provide further insight". If these experiments could be added, it would be an asset.

I would thus like to invite you to revise your manuscript with the understanding that the referee concerns must be fully addressed and their suggestions taken on board. Please address all referee concerns in a complete point-by-point response. Acceptance of the manuscript will depend on a positive outcome of a second round of review. It is EMBO reports policy to allow a single round of major revision only and acceptance or rejection of the manuscript will therefore depend on the completeness of your responses included in the next, final version of the manuscript.

We realize that it is difficult to revise to a specific deadline. In the interest of protecting the conceptual advance provided by the work, we recommend a revision within 3 months (12th Aug 2025). Please discuss the revision progress ahead of this time with the editor if you require more time to complete the revisions.

- 1) A data availability section providing access to data deposited in public databases is missing. If you have not deposited any data, please add a sentence to the data availability section that explains that.
- 2) Your manuscript contains statistics and error bars based on $n=2$. Please use scatter blots in these cases. No statistics should be calculated if $n=2$.

3) We replaced Supplementary Information with Expanded View (EV) Figures and Tables that are collapsible/expandable online. A maximum of 5 EV Figures can be typeset. EV Figures should be cited as 'Figure EV1, Figure EV2' etc... in the text and their respective legends should be included in the main text after the legends of regular figures.

5) a complete author checklist, which you can download from our author guidelines <<https://www.embopress.org/page/journal/14693178/authorguide>>. Please insert information in the checklist that is also reflected in the manuscript. The completed author checklist will also be part of the RPF.

6) Please note that all corresponding authors are required to supply an ORCID ID for their name upon submission of a revised manuscript (<<https://orcid.org/>>). Please find instructions on how to link your ORCID ID to your account in our manuscript tracking system in our Author guidelines <<https://www.embopress.org/page/journal/14693178/authorguide#authorshipguidelines>>

12) All Materials and Methods need to be described in the main text using our 'Structured Methods' format, which is required for all research articles. According to this format, the Methods section includes a Reagents and Tools Table (listing key reagents, experimental models, software and relevant equipment and including their sources and relevant identifiers) followed by a Methods and Protocols section describing the methods using a step-by-step protocol format. The aim is to facilitate adoption of the methodologies across labs. More information on how to adhere to this format as well as a downloadable template (.docx) for the Reagents and Tools Table can be found in our author guidelines:
<https://www.embopress.org/page/journal/14693178/authorguide#structuredmethods>.

An example of a Method paper with Structured Methods can be found here: <https://www.embopress.org/doi/full/10.1038/s44320-024-00037-6#sec-4>

You are able to opt out of this by letting the editorial office know (emboreports@embo.org). If you do opt out, the Review

Process File link will point to the following statement: "No Review Process File is available with this article, as the authors have chosen not to make the review process public in this case."

I look forward to seeing a revised form of your manuscript when it is ready.

Best,
Esther

Response to Reviewers

Manuscript number: RC202502929

Corresponding author(s): David M. Gilbert

We are pleased with the quality of the reviews and the review process. What follows is a complete point by point response, with our responses in **blue font** and the changes made in the revised manuscript in **bold blue font**.

Reviewer #1 (Evidence, reproducibility and clarity (Required)):

The study investigates the relationship between replication timing (RT) and transcription. While there is evidence that transcription can influence RT, the underlying mechanisms remain unclear. To address this, the authors examined a single genomic locus that undergoes transcriptional activation during differentiation. They engineered the PIn locus by inserting promoters of varying strengths to modulate transcription levels and assessed the impact on replication timing using Repli-seq.

Key Findings:

- Figure 1C and 1D: The data show that higher transcription levels correlate with an advanced RT, suggesting that transcriptional activity influences replication timing.
- Figure 2: To determine whether transcription alone is sufficient to alter RT, the authors inserted an hPGK reporter at different genomic locations. However, given the findings in Figure 1, which suggest that this is not the primary mechanism,
- Figure 3: The authors removed the marker to examine whether the observed effects were due to the promoter-driven PIn locus, which has significantly larger than the marker.
- Figure 4: The study explores the effect of increased doxycycline (Dox) treatment at the TRE (tetracycline response element), further supporting the role of transcription in RT modulation.
- Figure 5: The findings demonstrate that Dox-induced RT advancement occurs rapidly, is reversible, and correlates with transcription levels, reinforcing the hypothesis that transcription plays a direct role in influencing replication timing.
- Figure 6. Shows that during differentiation transcription of PIn is not required for RT advancement.

Overall, the study presents a compelling link between transcription and replication timing, though some experimental choices warrant further clarification. **I have no major comments.**

Minor Comments:

Overall, the results are convincing, and the study appears to be well-conducted.

In Figure 2, the authors use the hPGK promoter. However, it is unclear why they did not use the constructs from the previous experiments. Given that the hPGK promoter did not advance RT in Figure 1, the results in Figure 2 may not be entirely unexpected.

We took advantage of previously published cell lines using a PiggyBac Vector designed to pepper the reporter gene at random sites throughout the genome; the point of the experiment was to acquire supporting evidence for the hypothesis that any vector with its selectable marker driven by the hPGK promoter will not advance RT no matter where it is inserted. Since there are reports concluding that transcription per se is sufficient to advance RT, it was important to confirm that there was nothing unique about the particular vector or locus into which we inserted our panel of vectors.

ACTION DONE:

Response to Reviewers

We have now added the following sentence to the results describing this experiment: “By analyzing RT in these lines, we could evaluate the effect of a different hPGK vector on RT when integrated at many different chromosomal sites.”

Additionally, the study does not formally exclude the possibility that PIn protein expression itself influences RT. In Figure 1, readthrough transcription at the PIn locus could potentially drive protein expression. It would be useful to know whether the authors address this point in the discussion.

It is unclear why a secreted neural growth factor would have a direct effect on replication timing in self-renewing embryonic stem cells and, in particular, only *in cis* (remember there is a control allele that is unaffected). We would be happy to address this in the Discussion if we understood the reviewers' hypothesis. **We cannot respond to this comment without understanding the hypothesis being tested as we do not know how a secreted protein could affect the RT of one allele without affecting the other in a non-competent cell line.**

Regarding the mechanism, if transcription across longer genomic regions contributes to RT changes, transcription-induced could DNA supercoiling play a role. For instance, could negative supercoiling generated by active transcription influence replication timing?

Yes, many mechanisms are possible.

ACTION DONE:

We have added the following to the discussion, referencing a seminal paper on DNA supercoiling by Nick Gilbert: “ One hypothesis worth investigating is that longer transcripts may remodel large segments of chromatin, possibly by creating domains of DNA supercoiling (Naughton et al., 2013)”

It remains puzzling why PIn transcription does not contribute to replication timing during differentiation. Is there any evidence of chromatin opening during this process? For example, are ATAC-seq profiles available that could provide insights into chromatin accessibility changes during differentiation?

We thank the reviewer for asking this as we should have mentioned something very important here. Lack of necessity for transcription implies that independent mechanisms are functioning to elicit the RT switch. In other work (Turner et. al., EMBO J. 2025), we have shown that specific cis elements (ERCEs) can function to maintain early replication in the absence of transcription.

ACTIONS DONE:

We now explicitly state in the Discussion: “This is not surprising, given that ERCEs can maintain early RT in the absence of transcription (Turner et. al., EMBO J. 2025).”

In addition, we have added a new Figure 6D showing ATAC-seq changes upon differentiation of mESCs to mNPCs and their location relative to the promoter/enhancer deletion. As you can see, there is an ATAC-seq site that appears during differentiation, upstream of the deletion, as well as several that appear in the gene body. We added the following sentences to the Discussion after the sentence above, stating: “In fact, new sites of open chromatin, consistent with the properties of ERCEs, do appear outside of the deleted Ptn transcription control elements after differentiation (Figure 6C). The necessity and sufficiency of these sites to advance RT independent of transcription will be important to follow up.”

Response to Reviewers

Figure for referee with unpublished data has been removed upon request by the authors.

We also provide the figure below for the reviewer only. These are preliminary data that are part of a separate project in the lab so they are not ready for publication, but are directly relevant to the reviewer's question. This figure shows preliminary evidence for a region upstream of the Ptn promoter/enhancer deletion described in Figure 6 that, when deleted, DOES have an effect on the RT switch during differentiation. This deletion overlaps an ATAC-seq site we will show in the figure above.

Figure for referee with unpublished data has been removed upon request by the authors.

Reviewer #1 (Significance (Required)):

This is a compelling basic single-locus study that systematically compares replication timing (RT) and transcription dynamics while measuring several key parameters of transcription.

My relevant expertise lies in transcriptional regulation and understanding how noncoding transcription influences local chromatin and gene expression.

Reviewer #2 (Evidence, reproducibility and clarity (Required)):

In the manuscript entitled: Transcription can be sufficient, but is not necessary, to advance replication timing", the authors use as they state a "reductionist approach" to address a long-standing question in the replication field on what level the process of transcription within a replication domain can alter the underlying replication timing of this domain. The authors use an elegant hybrid mouse embryonic stem cell line to discriminate the two allelic copies and focus on a specific replication domain harboring the neuronal Ptn gene that is only expressed upon differentiation. The authors first introduce four different promoters in the locus upstream of Ptn gene that drive expression of small transgenes. Only the promoters with highest transcriptional induction could advance RT. If the promoters are placed in such a way that they drive expression of the 96kb Ptn gene, then also some the weaker promoters can drive RT advancement, suggesting that it is a combination of transcriptional strength and size of the transcribed domain important for RT changes. Using a DOX-inducible promoter, the authors show that this happens very fast (3-6h after transcription induction) and is reversible as removal

Response to Reviewers

of DOX leads to slower RT again. Finally, deleting the promoter of Ptn gene and driving cells into differentiation still advances RT, allowing the authors to conclude that "transcription can be sufficient but not necessary to advance replication timing."

Major comments:

Overall, this is a well designed study that includes all necessary controls to support the author's conclusions. I think it is a very interesting system that the authors developed. The weakness of the manuscript is that there is no mechanistic explanation how such RT changes are achieved on a molecular basis. But I'm confident that the system could be indeed used to further dissect the mechanistic basis for the transcription dependence of RT advancements.

Therefore, I support publication of this manuscript if a few comments below can be addressed.

1) Figure 4 shows a titration of different DOX concentrations and provides clear evidence that the degree of RT advancement tracks well with the level of transcription. As the doses of DOX are quite high in this experiment, have the authors checked on a global scale to what extent transcription might be deregulated in neighbouring genes or genome-wide?

The DOX concentration that we use for all experiments other than the titration is 2 µg/ml, which is standard. The high concentrations (up to 16µg/ml) are only used in the titration experiments shown in Figure 4 to demonstrate that we have reached a plateau. In fact, we stated in Materials and Methods that high doses of Dox led to cell toxicity. Looking at the transcription datasets, there are no significant changes in transcription below 8µg/ml, a few dozen significant changes at 8 µg/ml and more such changes at 16µg/ml of DOX. The tables of genome wide RT and transcription are provided in the manuscript for anyone wishing to investigate the effects of Dox on cellular physiology but at the concentration used in all other experiments (2µg/ml) there are no effects on transcription.

ACTIONS DONE:

We have now modified the statement in the Materials and Methods to read: "Mild toxicity and changes in genome-wide transcription were observed at 8µg/ml and more so at 16µg/ml".

In addition, we have now added to revised Expanded View Figure 3 a zoom out of 10 Mb surrounding the Ptn gene showing no detectable effects on transcription or RT at any of the titration concentrations. We also have added the following statement to the Results text discussing this experiment: "The effects of increasing concentrations of Dox were restricted to the Ptn locus (Fig. EV3B-C)."

2) One general aspect is that the whole study is only focused on the one single Ptn replication domain. Could the authors extend this rather narrow view a bit and also show RT data in the neighbouring domains. This would be particularly important for the DOX titration experiment that has the potential to induce transcriptional deregulation (see comment above).

ACTIONS DONE:

As discussed above, we have now added to revised Expanded View Figure 3 a zoom out of 10 Mb surrounding the Ptn gene showing no detectable effects on transcription or RT at any of the titration concentrations. We also have added the following statement to the Results text discussing this experiment: "The effects of increasing concentrations of Dox were restricted to the Ptn locus (Fig. EV3B-C)."

Response to Reviewers

In addition, to address the generalization of the findings, we have repeated the ESC to NPC differentiation and performed both Repli-seq and BrU-seq to evaluate RT changes relative to total genomic nascent transcriptional changes. We found, as expected for a cell fate change, transcriptional induction is correlated with advances in RT, but some of the advances were so small as to require high resolution Repli-seq. However, to the reviewer's point, we did find ample examples of domains that advance RT with no detectable transcriptional activity, as well as small and large genes that are induced with no advance or even a delay in RT. These observations, described in new Figure 7, Expanded View Figure 5 and Supplementary Figures 3 and 4, extend our observations with the Ptn replication domain to other domains throughout the genome.

3) Figure 5 shows that the full capacity to advance RT upon DOX induction of the Ptn gene is achieved after 3h to 6h of DOX induction, so substantially less than a full cell cycle in mESCs (12h). This result suggests that origin licensing/MCM loading cannot be the critical mechanism to drive the RT change because only a small fraction of the cells has undergone M/G1-phase where origins are starting to get loaded. As a large fraction of mESCs (60-70%) are S-phase cells in an asynchronous population, the mechanism is likely taking place directly in S-phase. Could the authors try to synchronize cells in G1/S using double-thymidine block, then induce DOX for 3h before allowing cells to reenter S-phase and then check replication timing of the domain? This can be compared to an alternative experiment where transcription is only induced for 3h upon release into S-phase. This could provide more mechanistic insights as to whether transcription is sufficient to drive RT changes in G1 versus S-phase cells.

We agree that the timing of induction is such that it is very likely that alterations in RT can occur during S phase. The reviewer proposes a reasonable experiment that could be done, but it would require a long delay of this publication to develop and validate those synchronization protocols and we do not have personnel at this time to carry out the experiment. This would be a great initiating experiment for someone to pursue the mechanisms by which transcription can advance RT.

ACTION DONE:

We have added the following sentence to the Discussion section on mechanisms: "The rapid nature of the RT change after induction of transcription suggests that RT changes can occur after the functional loading of inactive MCM helicases onto chromatin in telophase/early G1 (Dimitrova, JCB, 1999; Okuno, EMBO J. 2001; Dimitrova, J. Cell Sci, 2002), and possibly after S phase begins."

Minor comments:

- Figure 1B and Figure 6A. Quality of the genome browser snapshots could be improved and certain cryptic labelling such as "only Basic displayed by default" could be removed

ACTION DONE:

We have modified these figures.

- The genome browser tracks appear a bit small across the figures and could be visually improved.

ACTION DONE:

We have modified the genome browser tracks to improve their presentation

- In figure 1E we see an advancement in RT in Ptn gene caused by nearby enhanced Hyg-TK gene expression induced by mPGK promoter. However, in figure 3D we see mPGK promoter

Response to Reviewers

has reduced ability to advance RT of Ptn gene. It would be nice to address this discrepancy in the results.

The reviewer's point is well taken. We are not sure of the answer. You can see that the transcription is very low in both cases, while the RT shift is greater in one replicate vs. the other.

ACTION DONE: We have, rather unsatisfactorily, added the following sentence to the results section describing Figure 3. "Surprisingly, the mPGK promoter was very weak in this context (Fig. 3B and Fig. EV2), and advanced RT to a lesser extent."

Reviewer #2 (Significance (Required)):

In my point of view, this is an important study that unifies a large amount of literature into a conceptual framework that will be interesting to a broad audience working on the intertwined fields of gene regulation, transcription and DNA replication, as well as cell fate switching and development.

Reviewer #3 (Evidence, reproducibility and clarity (Required)):

In their manuscript, "Transcription can be sufficient, but is not necessary, to advance replication timing," Vouzas et al. take a systematic and reductionist approach to investigate a late-replicating domain on chromosome VI. Here, they examine the effect of transcribing a single gene locus, Pleiotrophin, on replication timing. When inserting or manipulating promoters or transcript lengths using CRISPR-Cas9, replication timing was altered in mESCs as judged by a combination of Repli-Seq, Bru-Seq, and RNA-Seq. Importantly, they found that transcription can be sufficient to advance replication timing depending on the length and strength of the expression of an ectopically transcribed gene. Taken together, the manuscript presents a compelling argument that transcription can advance replication timing but is not necessary for it.

Major comments

- A schematic or conceptual model summarising the major findings of transcription-dependent and independent mechanisms of RT advancement should be included in the discussion to add to the conceptual framework

NOT DONE:

We discussed this at length between the two senior authors and the first author and we do not feel ready to draw a summary model. Other than the act of elongation, we do not know what is advancing RT when transcription is induced or not induced, and we are not comfortable choosing one possible model of many. We hope that the added speculations on mechanism in the Discussion will sufficiently convey the future research that we feel needs to be done.

ACTIONS DONE:

In addition to the speculation on mechanism that already was in our Discussion section, we have added:

On mechanisms of rapid induction of RT change, we have added to the Discussion: "The rapid nature of the RT change after induction of transcription suggests that RT changes can occur after the functional loading of inactive MCM helicases onto chromatin in telophase/early G1 (Dimitrova, JCB, 1999; Okuno, EMBO J. 2001; Dimitrova, J. Cell Sci, 2002), and possibly after S phase begins." And "One hypothesis worth investigating is that longer transcripts may remodel large segments of chromatin, possibly by creating

Response to Reviewers

domains of DNA supercoiling (Naughton et al., 2013) or reducing interactions with the nuclear lamina. In fact, small hPGK-driven reporter genes were found to have only moderate and localized (~20kb) effects on interactions of flanking chromatin with the lamina while longer genes had more extensive effects (Brueckner et al., 2020).”

On mechanisms of RT advance in the absence of transcription, we have added the following to the Discussion: “This is not surprising, given that ERCs can maintain early RT in the absence of transcription (Turner et. al., 2025). In fact, new sites of open chromatin, consistent with the properties of ERCs, do appear outside of the deleted Ptn transcription control elements after differentiation (Figure 6C). The necessity and sufficiency of these sites to advance RT independent of transcription will be important to follow up.”

• Vouzas et al. spend a substantial part of the manuscript to delve into the requirements to advance RT and even use a Doxycycline-based titration for temporal advancement of RT. Yet, all conclusions come from the use of hybrid-genome mouse embryonic stem cells (mESCs). Therefore, it remains speculative if and whether findings can be generalized to other cell types or organisms. The authors could include another organism/ cell type to strengthen the relevance of their findings to a broader audience, particular as they identified promoters that drive ectopic gene expression without affecting RT. Showcasing this in other model organisms would be of great interest.

NOT DONE:

To set this system up in another cell type or species would take a very long time. We also do not have personnel with the expertise to carry set up such systems.

ACTION DONE:

We provide a new Figure 7, along with Expanded View Figure 5 and Supplemental Figures 3 and 4 with an analysis of RT changes vs. transcriptional changes when mESCs are differentiated to neural precursor cells. As described above in response to Review #2s criticism #2, we have repeated the ESC to NPC differentiation and performed both Repli-seq and BrU-seq to evaluate RT changes relative to total genomic nascent transcriptional changes. We found, as expected for a cell fate change, transcriptional induction is almost always correlated with advances in RT, but some of the advances were so small as to require high resolution Repli-seq. However, to the reviewer’s point, we did find examples of domains that advance RT with no detectable transcriptional activity, as well as small and large genes that are induced with no advance or even a delay in RT. These observations extend our observations with the Ptn replication domain to other domains throughout the genome.

• OPTIONAL: as with the previous point, the authors went to great depth and length to show how ectopic manipulations affect RT changes on a single locus using genome-wide methods. In addition, the manuscript would benefit from the inclusion of other loci, particularly as transcription of the Ptn locus wasn’t needed during differentiation to advance RT at all.

NOT DONE:

This rigorous reductionist approach is laborious and to set it up at one gene at a time at additional loci would be a huge effort taking quite a long time.

ACTION DONE:

Response to Reviewers

We believe that the approach described above, examining natural changes in RT relative to changes in transcription, at least partially addresses this concern.

- The same point of Ptn not needing to be transcribed to advance RT of the respective domain, albeit being a very interesting observation, disturbs the flow of the manuscript, as the whole case was built around transcription and this particular locus-containing domain. Maybe one can adapt the storytelling to fit better within the overall framework.

ACTIONS DONE:

To make this transition more smooth, we have added the following sentence to the beginning of the results section describing Figure 6: “ This raises the question as to whether the natural RT advance that accompanies Ptn induction during differentiation requires Ptn transcription, or whether other mechanisms, such as ERCs, can advance RT independent of transcription (Sima et al., 2019; Turner et al., 2025). “

In addition, we believe that the new analysis of RT changes vs. transcriptional changes when mESCs are differentiated to mNPCs (new Figures 7, Expanded View Figure 5, and Supplementary Figures 3-4), as described above, extends the theme of “usually sufficient but not necessary”, to smooth the transition.

Minor comments

- While citations are thorough, some references (e.g., "need to add Wang, Klein, Mol. Cell 2021") are incomplete.

ACTIONS DONE:

We apologize that some references seemed to not be incorporated into the reference manager Mendely. We have combed the manuscript for any references that were not entered and corrected them.

- The text corresponding to Figure 1C could use more explanation for readers not familiar with the depiction of Repli-Seq data.

ACTION DONE:

We have added the following sentences to the Results text: “In Repli-seq, nascent DNA is labeled with BrdU, followed by flow cytometry to purify cells in early vs. late S phase based on their DNA content, immunoprecipitation of BrdU-substituted DNA and sequencing of the nascent DNA. Results are expressed as a log2 ratio of early to late S nascent DNA (log2E/L). BrU-seq labels total nascent RNA, which is then immunoprecipitated, sequenced and expressed as reads per million per kilobase (RPMK).”

- Figure 1C needs labelling of the x-axes.

ACTION DONE:

We have now labeled the X axes.

- Statistical analyses should be used consistently throughout the manuscript and explained in more detail, i.e. significance levels, tests, instead of "Significant differences....calculated using x".

We used the same analysis for all the Repliseq data and the same analysis for all the Bruseq data. We agree that we did not present this consistently in the figure legends and methods.

ACTION DONE:

Response to Reviewers

To correct the confusion we have clarified the statistical methods in the methods section and referred to methods in the figure legends as follows:

The methods description of statistical significance for RT now reads: “Statistical significance of RT changes for all windows in each sample, relative to WT, were calculated using RepliPrint (Ryba et al., 2011), with a p-value of 0.01 used as the cut-off for windows with statistically significant differences.”

The methods description of statistical significance for transcription now reads: “Differential expression analysis, including the calculation of statistically significant differences in expression, was conducted using the R package DESeq2. In Figure 1, statistical significance was calculated relative to HTK expression in the parental cell line, which is expected to be zero, since the parental line does not have an HTK insertion. In all other Figures significance was calculated relative to Ptn expression in the parental line, which is expected to be zero, since the parental line does not express Ptn.”

The legend to Figure 1C now reads: The red shading indicates 50kb windows with statistically significant differences in RT between WT casteneus and modified 129 alleles, determined as described in Methods.

The legend to Figure 1E now reads: “The asterisks indicate a significant difference in the levels of HTK expression relative to HTK expression in the parental cell line as described in Methods. There are no asterisks for the RT data, as statistical significance was calculated for individual 50kb windows as shown in panel (C).”

Each time significance is measured in the subsequent legends, it is followed by the phrase “, determined as described in Methods” or “presented as in Figure 1C” or “presented as in Figure 1E” as appropriate.

****Referees cross-commenting****

Comment on Reviewer#1's review, comment mentioning ATAC-Seq: Another way to look at this could be to investigate for origin usage changes (BrdU-Seq or GLOE-Seq) of chromosome 6 during differentiation.

NOT DONE:

This would also be a huge undertaking, more appropriate for a follow up study. Unfortunately, we also could not find any studies comparing BrdU-seq of GLOE-Seq mapping in mESCs and mNPCs to do preliminary studies of these data sets. We do provide some examples of high resolution Repli-seq in the new Figure 7 and new Supplementary Figures 3 and 4.

Comment on Reviewer#2's review, major comment 3: I do agree with their statement that origin loading cannot be the driver of RT change, as MCM2-7 double hexamer loading is strictly uncoupled from origin firing. Hence, any mechanism responsible for RT advance must happen at the G1/S phase transition or during S-phase, most likely due to the regulated activity of DDK/CDK or the limitation and preferred recruitment of firing factors to early origins. This could be tested through overexpression of said factors.

NOT DONE:

We agree that manipulating these factors would be a reasonable next approach to sort out mechanism. Due to limited resources and personnel, we will not be able to do this in a short period of time. We also argue that these are experiments for the next chapter of the story, likely requiring an entire PhD thesis (or multiple) to sort out.

Response to Reviewers

ACTION DONE:

We have added the following sentence to the Discussion section on mechanisms: “The rapid nature of the RT change after induction of transcription suggests that RT changes can occur after the functional loading of inactive MCM helicases onto chromatin in telophase/early G1 (Dimitrova, JCB, 1999; Okuno, EMBO J. 2001; Dimitrova, J. Cell Sci, 2002), and possibly after S phase begins.”

Reviewer #3 (Significance (Required)):

General: This manuscript presents a compelling study investigating the relationship between transcription and replication timing (RT) using a reductionist approach. The authors systematically manipulated transcriptional activity at the Ptn locus to dissect the elements of transcription that influence RT. The study's strengths lie in its rigorous experimental design, clear results, and the reconciliation of seemingly contradictory findings in the existing literature. However, some aspects could be improved, particularly in exploring the mechanistic details of transcription-independent RT regulation at the investigated domain, the generalisability of the findings to other cells/organisms, and enhancing the presentation of certain data (explanation of e.g. Figure 1c, dense figure arrangement, lack of a summary figure illustrating key findings (e.g., correlation between transcription rate, readthrough effects, and RT advancement)).

Advance: The manuscript directly addresses and reconciles contradictory findings in the literature regarding the effect of ectopic transcription on RT. Previous studies have reported varying effects, with some showing that transcription advances RT (Brueckner et al., 2020; Therizols et al., 2014), while others have shown no effect or only partial effects depending on the insertion site (Gilbert & Cohen, 1990; Goren et al., 2008). The current study conceptually advances the field by systematically testing different promoters and transcript lengths at a single locus (mechanistic insight), demonstrating that the length and strength of transcription, as well as promoter context, influence RT. This presents a unifying concept on how RT can be influenced. The authors also present a tunable system (technical advance) that allows rapid and reversible alterations of RT, which will certainly be useful for future studies and the field.

Audience: The primary audience will be specialised researchers in the fields of replication timing, epigenetics, and gene regulation. This study may be of interest beyond the specific field of replication timing, such as cancer biology, developmental biology, particularly if a more broader applicability of its tools and concepts can be shown.

Expertise: origin licensing, origin activation, MCM2-7, yeast and human cell lines

Dear David,

Thank you for the submission of your revised manuscript. We have now received the enclosed reports from the referees and I am happy to say that all support its publication now. Only a few editorial requests will need to be addressed before we can proceed with the official acceptance:

- Please add up to 5 keywords to your ms file.
- Please add a "Disclosure and Competing Interest Statement"
- The corr. author's email is missing on the title page of the ms, please add.
- The REFERENCE format is not correct: et al needs to be used after 10 author names; DOIs should only be used for preprints and datasets that have not been published yet. Please correct to the EMBO reports reference style.
- Please submit with your final ms a completed author checklist, which you can download from our author guidelines <<https://www.embopress.org/page/journal/14693178/authorguide>>. The completed author checklist will also be part of the transparent peer-review file (RPF).
- If "KWF Dutch Cancer Society" is a separate funder, please enter this as a separate funder during online ms submission.
- Figure 1 and Figure 5 are not centered on the page, please correct. Figures 1,3 and 5 are formatted in landscape, please provide these figures in portrait format.
- The callouts for Fig. 7G and Appendix Figures S2-S4 are missing, please add; Supplemental Figure 1 is not a correct callout.
- The APPENDIX FILE needs to be a PDF file, the table of content needs to have page numbers and the correct nomenclature should be Appendix Figure S1-S4 throughout the file (callouts in the ms also need updating); the individual Appendix figures need to be removed. Alternatively, you could also add 4 more EV figures and delete the Appendix file.
- The Nomenclature of the EV figure legends needs correction: it should be Figure EV1, etc. instead of Expanded View Figure 1, etc.
- The manuscript sections should be in the following order: Title page - Abstract & Keywords - Introduction - Results - Discussion - Methods - Data Availability - Acknowledgments - Disclosure Statement & Competing Interests - References - Figure Legends - (Main Tables with legends if applicable) - Expanded View Figure Legends.

* Figure Legends - Comments *

- Please define the annotated p values ****/****/**/* as well as provide the exact p-values for the same in the legend of figure 1E, 3D, 4C, 5E as appropriate and reasonable.
- Please note that the exact p values are not provided in the legends of figures 7B, G, please provide exact values as reasonable.
- Please indicate the statistical test used for data analysis in the legends of figures 1E, 3D, 4C, 5E, 7B
- Please note that information related to n is missing in the legends of figures 3D, 4C, 5E, 7G, EV1 C, D
- Please note that the error bars are not defined in the legends of figures 1E, 3D, 4C, 5E, EV1 C, D

I would like to suggest some minor changes to the abstract that needs to be written in present tense. Please let me know whether you agree with this and please address my comment:

DNA replication timing (RT) correlates with transcription during cell fate changes but there are exceptions, underscoring a need for reductionist approaches. Here, we manipulate length and strength of transcription at a single site upstream of the silent, late-replicating, Pleiotrophin (Ptn) gene in mouse embryonic stem cells (mESCs). Reporter genes driven by two of four promoters elicit an RT advance while all four promoters advance RT when driving the endogenous Ptn gene. Inducible transcription of the Ptn gene, but not the reporter gene, elicits a rapid and reversible RT advance. Deletion of the Ptn promoter and enhancer, followed by differentiation to neural precursors, eliminates transcription without attenuating the natural Ptn domain RT advance [should this rather be "delay" given that the gene is late replicating?]. Together with new genome-wide analyses, our results provide a solid empirical base with which to re-evaluate decades of seemingly contradictory literature. We also identify vectors that do not disturb RT and provide a robust system to induce RT changes, permitting mechanistic studies of transcription's role in RT and the consequences of RT changes to epigenomic remodeling.

EMBO press papers are accompanied online by A) a short (1-2 sentences) summary of the findings and their significance, B) 2-3 bullet points highlighting key results and C) a synopsis image that is exactly 550 pixels wide and 200-600 pixels high (the height is variable). The synopsis image should provide a sketch of the major findings, like a graphical abstract. Please note that text needs to be readable at the final size. Please send us this information along with the final manuscript.

Best,
Esther

Referee #1:

In the revised version of their manuscript, "Transcription can be sufficient, but is not necessary, to advance replication timing," Vouzas et al. have provided further explanation and new data on their systematic and reductionist approach to investigate a late-replicating domain on chromosome VI. By incorporating existing genome-wide data, for example, the production of a new Figure 6D (ATAC-Seq) and 7 (BrdU- and BrU-Seq) the authors provide clarification. Improvements span the material& methods section, description of statistical methods, unification of analyses, and presentation of data. Finally, the discussion has improved to reflect the intricate and non-linear relation of replication timing and transcription. Most of the reviewer's questions, suggestions, and inquiries have been addressed sufficiently to justify publication in EMBO Reports.

Referee #2:

The authors have adequately addressed my previous concerns and critiques. I would have loved to see the RT measurement after cell cycle synchronization to obtain more mechanistic insights but understand the necessary balance between timely publication of the manuscript and required workload. In the end, the experiment will not change the major conclusion that transcription can be sufficient but is not necessary to advance RT. It is a well conducted study and I support timely publication in its current state.

Referee #3:

I reviewed the first submission and now read the revised manuscript. The first submission was already of very high quality. The authors have addressed my comments. It is an excellent study.

The authors have addressed all minor editorial requests.

Prof. David Gilbert
San Diego Biomedical Research Institute
Chromosome Replication and Epigenome Regulation
United States

Dear David,

I am very pleased to accept your manuscript for publication in the next available issue of EMBO reports. Thank you for your contribution to our journal.

You may qualify for financial assistance for your publication charges - either via a Springer Nature fully open access agreement or an EMBO initiative. Check your eligibility: <https://link.springer.com/journal/44319/how-to-publish-with-us>

>>> Please note that it is EMBO Reports policy for the transcript of the editorial process (containing referee reports and your response letter) to be published as an online supplement to each paper. If you do NOT want this, you will need to inform the Editorial Office via email immediately. More information is available here: <https://link.springer.com/partners/embo-press/editorial-policies#Peer%20review>